# Lipopolysaccharide confinement in the bacterial outer membrane is governed by interactions within the conserved Lipid A anchor

Joe Nabarro [1,2,7], Rosalyn M Leaman[1,7], Samuel Lenton [1,5], Leonore Mantion[1,6], Richard J Spears[2], Mark C Coles [3], Dmitri O Pushkin[4], Martin A Fascione [2✉] & Christoph G Baumann [1✉]

## Abstract

**The outer membrane of Gram-negative bacteria is an asymmetric bilayer in which lipopolysaccharide (LPS), the principal component of the outer leaflet, promotes tight packing and ordering of the membrane components that are essential for the barrier and load-bearing functions of this membrane. Lipopolysaccharide mobility is known to be restricted in the outer membrane, but this confinement and the underlying biophysical interactions responsible remain to be fully characterized. Here, we apply a bio-orthogonal strategy for in situ site-specific fluorescent labeling of LPS. Using fluorescence microscopy, we quantify LPS lateral confinement in the outer membrane of *Escherichia coli* and demonstrate that this confinement is independent of oligosaccharide domain structure. We show that lipopolysaccharide assembles into discrete supramolecular structures, and that restricted lateral mobility arises from a combination of divalent cation-mediated electrostatic interactions in the anionic Lipid A headgroup, and hydrophobic interactions between acyl chains within the lipid milieu. Magnesium cations exert a greater influence than calcium cations on lipopolysaccharide lateral mobility. These traits are conserved across multiple pathogenic bacterial species, irrespective of O-antigen serotype, showing that lipopolysaccharide confinement is a ubiquitous feature of Gram-negative bacteria.**

**Keywords** Lipopolysaccharide; Gram-negative Bacteria; Outer Membrane Diffusion; Bio-orthogonal Labeling; Fluorescence Microscopy
**Subject Categories** Membranes & Trafficking; Microbiology, Virology & Host Pathogen Interaction

## Introduction

A distinct feature of the Gram-negative bacterial cell envelope is the outer membrane (OM) with its asymmetric distribution of phospholipid and lipopolysaccharide in the inner and outer leaflets, respectively. This OM asymmetry is critical to cellular integrity and antibiotic resistance in Gram-negative bacteria (Delcour, 2009; Silhavy et al, 2010), with several trans-envelope machines involved in assembling this bilayer and then maintaining its lipid homeostasis and asymmetry (Tan and Chng, 2024). The amphipathic glycolipid lipopolysaccharide (LPS) is a major component of the OM, alongside β-barrel outer membrane proteins (OMPs) (Koebnik et al, 2000; Nikaido, 2003), and as such its role in governing OM morphology and function is well established (Henderson et al, 2016; Lithgow et al, 2023). Lipid A anchors LPS in the membrane and has a chemical structure conserved across most Gram-negative bacteria (Whitfield and Trent, 2014), comprising a bis-phosphorylated β-1,6-glucosamine disaccharide bearing six acyl chains (Figs. 1A,D and EV1). The hydrophobic elements result in the tight packing of LPS promoted by intermolecular van der Waals interactions that maintain the lipid component in the outer leaflet of the OM in a gel-like state with low fluidity (Sun et al, 2022), in sharp contrast to the liquid-disordered state present in most phospholipid-rich biological membranes. Despite the significant functional role of LPS, the degree to which intermolecular interactions in this tightly packed state dictate its mobility in the OM remains underexplored.

Divalent cation-mediated electrostatic interactions between conserved anionic groups exposed at the base of LPS have long been hypothesized to influence its diffusion and therefore OM organization (Herrmann et al, 2015; Jeworrek et al, 2011). In addition, intermolecular hydrogen bonds mediated by the more distal regions of the oligosaccharide domain may also play a role (Snyder et al, 1999). These intermolecular interactions are predicted to restrict LPS lateral mobility in the OM, contributing to the tight packing which underpins its barrier function (Rahnamoun et al, 2020). A limited number of landmark in vivo studies have attempted to characterize the lateral mobility of LPS in

[1]Department of Biology, University of York, York YO10 5DD, UK. [2]Department of Chemistry, University of York, York YO10 5DD, UK. [3]Kennedy Institute of Rheumatology, Nuffield Department of Orthopaedics, Rheumatology and Musculoskeletal Science, University of Oxford, Oxford OX3 7FY, UK. [4]Department of Mathematics, University of York, York YO10 5DD, UK. [5]Present address: Department of Pharmacy, University of Copenhagen, Copenhagen 2100, Denmark. [6]Present address: ADNucleis, Parc Cap Quest, 24B Rue du Stade, Grézieu-la-Varenne 69290, France. [7]These authors contributed equally: Joe Nabarro, Rosalyn M Leaman. ✉E-mail: martin.fascione@york.ac.uk; christoph.baumann@york.ac.uk

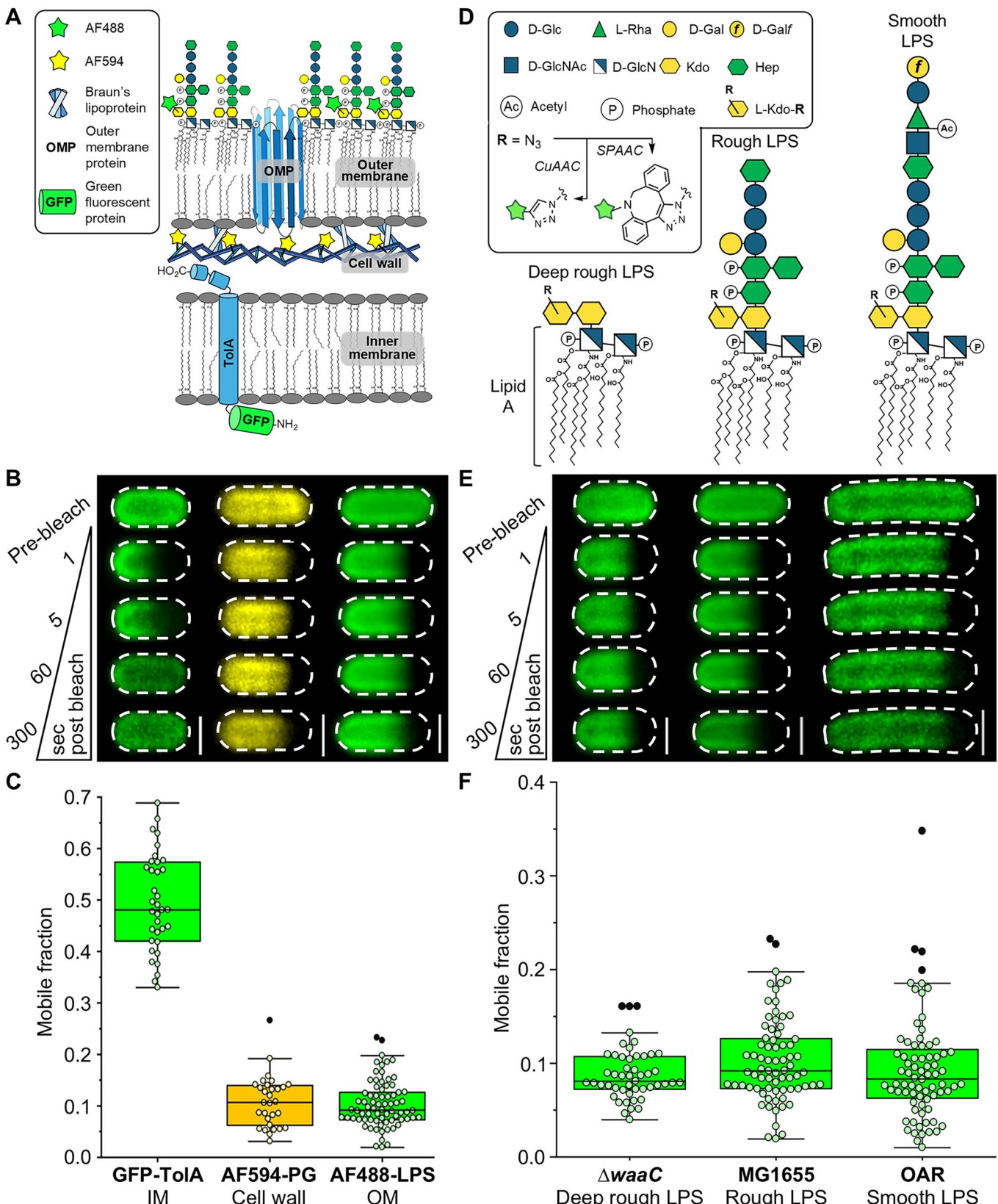

◀  **Figure 1.   LPS lateral mobility in the OM is very restricted and influenced by intermolecular interactions involving the conserved Lipid A-(Kdo)$_2$ core.**

(A) Schematic of bacterial cell envelope showing specific targets which were fluorescently labeled with an organic dye or GFP to assess lateral mobility by confocal fluorescence recovery after photobleaching (FRAP) microscopy. Cell wall peptidoglycan (PG) and the transmembrane TolA complex are characterized by very restricted and mostly unrestricted lateral mobilities, respectively. Fluorescent labeling of LPS via a 2-step bio-orthogonal approach enabled its lateral mobility to be assessed via FRAP. (B) Representative FRAP sequences of GFP-TolA in the IM (left), AF594-PG in the periplasm (middle), and AF488-labeled LPS in the OM outer leaflet (right) of *E. coli* cells. Dashed border denotes the outline of each bacterial cell. Scale bars: 1.0 µm. (C) Comparison of mobile fractions for GFP-TolA (median = 0.476, n = 36), AF594-PG (median = 0.107, n = 30) and AF488-LPS (median = 0.090, n = 69) demonstrate that LPS lateral mobility is very restricted. All FRAP experiments were done in triplicate using independent biological samples. (D) Schematics of deep rough, rough, and smooth LPS glycoforms present in Δ*waaC*, MG1655, and O-antigen restored (OAR) *E. coli* strains, respectively, showing the composition of the oligosaccharide domain displayed at the cell surface. In situ metabolic labeling of the chemically conserved LPS inner core oligosaccharide domain was done with an azide-functionalized Kdo-analog, which enabled specific conjugation of alkyne-functionalized organic fluorescent dyes via Cu(I)-catalyzed (CuAAC) or Cu(I)-free strain-promoted azide-alkyne cycloaddition (SPAAC). (E) Representative FRAP sequences for AF488-LPS in the OM of deep rough LPS-producing (left), rough LPS-producing (middle), and smooth LPS-producing (right) *E. coli* cells. Dashed border denotes the outline of each bacterial cell. Scale bars: 1.0 µm. (F) Comparison of FRAP-derived mobile fractions shows that restriction of LPS lateral mobility is independent of the different glycoforms present in Δ*waaC* (median = 0.081, n = 49), MG1655 (median = 0.091, n = 69), and OAR (median = 0.085, n = 74) *E. coli* strains. All FRAP experiments were done in triplicate using independent biological samples. For each box plot, the center line is the median, while the box defines the upper and lower quartiles of the data, and the whiskers enclose the interquartile range (i.e., middle 50% of the data). Each symbol represents an individual measurement with black-filled symbols classified as outliers (i.e., outside the interquartile range). Source data are available online for this figure.

the OM of intact cells (Ghosh and Young, 2005; Mühlradt et al, 1974; Schindler et al, 1980), with contrasting findings. Two of these studies suggested endogenous smooth LPS was immobile, but the large (Mühlradt et al, 1974) or multivalent (Ghosh and Young, 2005) exogenous fluorescent probes used to label the LPS may be unsuitable for accurate in vivo characterization, as their binding potentially influences LPS mobility and organization in the OM.

Metabolic LPS engineering (Dumont et al, 2012; Nilsson et al, 2017) enables direct, in situ fluorescent labeling of the Lipid A-(Kdo)$_2$ conserved within the LPS inner core of Gram-negative bacterial strains (Figs. 1D and EV1). This specific bio-orthogonal conjugation strategy can be used to overcome the limitations of the aforementioned LPS labeling methods by allowing covalent coupling of organic dyes to surface-exposed LPS molecules in intact *Escherichia coli* cells (Ziylan et al, 2023), and was recently applied to demonstrate low mobility of rough LPS in the OM of *E. coli* (Kumar et al, 2025). Herein, we have employed this bio-orthogonal labeling strategy with diffraction-limited and single-molecule fluorescence microscopy to confirm this recent work (Kumar et al, 2025), and demonstrate that LPS is laterally confined in the OM of Gram-negative bacterial strains, irrespective of the carbohydrate domain size and structure in the LPS headgroup. By characterizing the molecular basis for this restriction, we provide new insight into the mechanisms that underpin the vital barrier and load-bearing functions of the OM in Gram-negative bacteria (Rojas et al, 2018; Sun et al, 2022).

## Results

### LPS is tightly confined in the OM irrespective of oligosaccharide domain size

In a two-step approach, bacteria were initially cultured with 3-deoxy-D-manno-oct-2-ulosonic acid (Kdo) monosaccharide modified with a bio-orthogonal azide group (Kdo-N$_3$). Kdo-N$_3$ was taken up via a salvage pathway, biosynthetically incorporated into the inner core oligosaccharide domain of LPS (Figs. 1D and EV1) (Dumont et al, 2012; Nilsson et al, 2017) and then "click"-labeled at the cell surface via Cu(I)-catalyzed (CuAAC) or Cu(I)-free strain-promoted azide-alkyne cycloaddition (SPAAC) (Mbua et al, 2011;

Wang et al, 2003) with an exogenously delivered alkyne-functionalized, small (< 1000 Da) photostable fluorophore. The water-soluble fluorophores used here for LPS labeling (i.e., Alexa Fluor 488 (AF488), AZDye 568 (AZ568), and AZDye 647 (AZ647)) had a very low propensity to interact with membranes (Hughes et al, 2014). The bio-orthogonal CuAAC approach enabled the high-level, site-specific fluorescent labeling of LPS required for FRAP, super-resolution imaging, and biochemical analysis by gel electrophoresis. Using the CuAAC approach and FRAP, we confirm that LPS is laterally confined in the outer leaflet of the OM in the *E. coli* K-12 cell envelope (Fig. 1B,E), in agreement with previous work done using the SPAAC approach (Kumar et al, 2025). Furthermore, cells "click"-labeled with fluorophore using the CuAAC and SPAAC approaches yielded identical levels of LPS confinement by FRAP (Fig. EV2), indicating that the loss of cell viability associated with copper exposure during the CuAAC reaction does not disrupt native LPS confinement.

Next, we compared the mobile fraction distributions of fluorescently labeled rough LPS (AF488-LPS) in the OM of intact *E. coli* K-12 MG1655 cells with those of fluorescently labeled polymeric peptidoglycan (PG) in the cell wall (Kuru et al, 2012; Siegrist et al, 2013). Both species exhibited very restricted average lateral mobility (Fig. 1A–C), and despite the difference in their molecular weights (average $M_{w,PG} \approx 3.9 \times 10^9$ Da vs. $M_{w,rough\ LPS} \approx 3000$ Da) no statistically significant difference was observed when FRAP-derived mobile fraction distributions were compared ($P = 0.63$, Appendix Table S1). Furthermore, the median mobile fraction measured by FRAP for the cell wall ($\approx 0.107$) represents a threshold for identifying a completely immobile species in the bacterial cell envelope. The immobility of LPS in the OM and PG in the cell wall sharply contrasted with that of a recombinant N-terminal GFP-tagged inner membrane (IM) protein (GFP-TolA), shown previously to undergo Brownian diffusion (Fig. 1A–C) (Rassam et al, 2018), with the lateral mobility of AF488-LPS significantly restricted compared to that of GFP-TolA ($P = 4.4 \times 10^{-7}$, Appendix Table S1).

Having confirmed LPS confinement in the context of the Gram-negative OM, we sought to identify and characterize the underlying intermolecular interactions. We began by investigating the influence of LPS oligosaccharide domain structure (Fig. 1D) on lateral mobility, conducting FRAP experiments on labeled LPS in

the OM of an *E. coli* K-12 Δ*waaC* mutant strain capable of only producing deep rough LPS, *E. coli* MG1655 where rough LPS predominates, and an O-antigen restored (OAR) *E. coli* strain producing smooth LPS (Fig. 1E) (Baba et al, 2006; Browning et al, 2013). Efficient fluorescent labeling using the CuAAC approach was confirmed for these bacterial strains with different LPS glycoforms (Fig. EV3; Appendix Fig. S1). Despite pronounced differences in the predominant LPS oligosaccharide domain structures (Fig. 1D), inter-strain variations in LPS mobile fraction distributions were statistically insignificant (Appendix Table S1). Any small differences in LPS lateral mobility between these strains and the tightly confined cell wall (Fig. 1E,F) are undetectable in our FRAP experiments.

## LPS assembles into supramolecular structures in the OM

We then employed two-color direct stochastic optical reconstruction microscopy (dSTORM) to map the distributions of the fluorescently labeled LPS relative to an abundant OMP (OmpA, $10^5$ copies in the OM (Koebnik et al, 2000)), which is distributed throughout the beta-barrel protein milieu at the *E. coli* cell surface (Benn et al, 2021) and is used here as a marker for these porin-rich regions (Fig. 2A; Appendix Fig. S2). Simultaneous OMP fluorescent-labeling was achieved using amber codon suppression (Plass et al, 2011), incorporating an alkyne functional group within an extracellular loop of OmpA (OmpA*), enabling azide-functionalized dye conjugation (Appendix Fig. S2). OmpA* production was performed in a Δ*ompA E. coli* strain for which growth was inhibited by SDS and EDTA (Appendix Fig. S3). This SDS and EDTA induced growth inhibition was not observed in the *ompA** expressing Δ*ompA E. coli* strain upon site-specific insertion of the non-canonical amino acid (Appendix Fig. S3), therefore the recombinant OmpA* protein is able to substitute for wild-type OmpA. Interrogation of the resulting dSTORM data revealed distinct LPS-rich and OMP-rich regions within OMs that were independent of the dye selected to label each OM component (Appendix Fig. S2), with median diameters of ~0.26 μm and ~0.18 μm, respectively (Fig. 2B). Subsequent co-localization analysis confirmed that LPS can form supramolecular structures or patches largely devoid of OmpA (Fig. 2C), as recently shown (Benn et al, 2021). This discrete clustering of fluorescently labeled LPS allowed us to explore the factors influencing lateral mobility within these glycolipid-rich patches, thus facilitating the dissection of dominant intermolecular LPS–LPS interactions over LPS–OMP interactions.

## LPS lateral confinement is influenced by OM asymmetry and OM-PG covalent linkages

Accordingly, we investigated whether OM composition, asymmetry, and covalent OM-PG associations influence average LPS lateral mobility. We compared the mobility of LPS in the MG1655 strain with that of LPS in the *imp4213* strain (Ruiz et al, 2005), in which wild-type levels of OMPs are maintained, but OM asymmetry is compromised. In *imp4213* cells, a solitary in-frame deletion in *lptD* disrupts LptD/LptE lipoprotein complex formation, reducing the rate of LPS insertion in the OM and increasing OM permeability to antibiotics (Ruiz et al, 2005). The resulting decrease in LPS density

in the OM outer leaflet prevents tight packing of adjacent LPS molecules, reducing the capacity for LPS–LPS interactions and creating spaces in the outer leaflet that are spontaneously filled with phospholipid (PL) from the inner leaflet. Our bio-orthogonal SPAAC approach enabled the low-level, site-specific fluorescent labeling of LPS required for single-molecule detection and localization. Therefore, single-particle tracking (SPT) was used to directly probe whether sub-populations of LPS molecules with different lateral mobilities existed in the *imp4213* strain.

SPT of video data for AF488-LPS in the *imp4213* strain revealed two populations of rough LPS molecules with distinct diffusive characteristics (Fig. 3A,C). One-third of the LPS molecules were observed to diffuse freely in the OM ($D_{2D} = 0.0550 \pm 0.00744$ μm²/s), while two-thirds of the LPS molecules displayed restricted lateral diffusion ($D_{2D} = 0.0347 \pm 0.00237$ μm²/s) (Fig. 3B; Table EV1). The lateral diffusion coefficients obtained for confined rough LPS (MG1655, $0.0182 \pm 0.000645$ μm²/s) and confined deep rough LPS (Δ*waaC*, $0.0181 \pm 0.000727$ μm²/s) were much slower in comparison and nearly identical to each other in magnitude (Table EV1). The confinement diameter estimated from the asymptotic mean-squared displacement (MSD) value was greater for the *imp4213* strain ($0.723 \pm 0.0207$ μm) relative to the MG1655 ($0.543 \pm 0.00666$ μm) and Δ*waaC* ($0.543 \pm 0.00619$ μm) strains (Table EV1). These observations indicate that LPS mobility is globally increased in the *imp4213* strain due to an overall reduction in corralling within the supramolecular LPS-rich patches (Fig. 2), and a parallel increase in the lateral diffusion rate, perhaps due to increased fluidity when more PL is present in the outer leaflet of the OM in *imp4213* cells. Consistent with these single-molecule level observations (i.e., increase in $D_{2D}$ and confinement diameter, and presence of a freely diffusing sub-population), the LPS mobile fraction in the *imp4213* strain determined by FRAP (≈ 0.20) was also significantly higher compared to the MG1655 strain (≈ 0.09, Fig. 3D; Appendix Table S1). These complementary data suggest that strict maintenance of OM asymmetry and tight, ordered packing of adjacent molecules to maximize the number and strength of LPS–LPS interactions is a prerequisite for restricted lateral mobility of LPS in the OM. To reinforce this hypothesis, we demonstrated that restoration of OM asymmetry in the *imp4213* strain, through expression of a functional recombinant LptD protein capable of LptD/LptE complex formation (Appendix Fig. S4), resulted in a statistically significant decrease in the average LPS lateral mobility to near wild-type levels (Fig. 3D; Appendix Table S1). Rescued *imp4213* cells producing functional LptD also showed reduced sensitivity to SDS, EDTA, and bile salts (Appendix Fig. S4), implying links exist between OM asymmetry, LPS confinement, and the barrier function of the OM.

The periplasmic surface of the OM is held near the bacterial cell wall by the very abundant Braun's lipoprotein (Lpp), which is both covalently coupled to peptidoglycan and anchored in the membrane (Asmar and Collet, 2018). We also measured the average lateral mobility of LPS in an *E. coli* Δ*lpp* strain, shown previously to have decreased OM stiffness and stability (Fitzmaurice et al, 2025; Rojas et al, 2018; Sonntag et al, 1978). A statistically significant increase in the lateral mobility of LPS was observed in the Δ*lpp* strain ($P = 3.4 \times 10^{-7}$, Appendix Table S1) relative to the MG1655 strain (Fig. 3D). Previously, we showed that OMP lateral mobility was not affected by the deletion of *lpp* (Rassam et al,

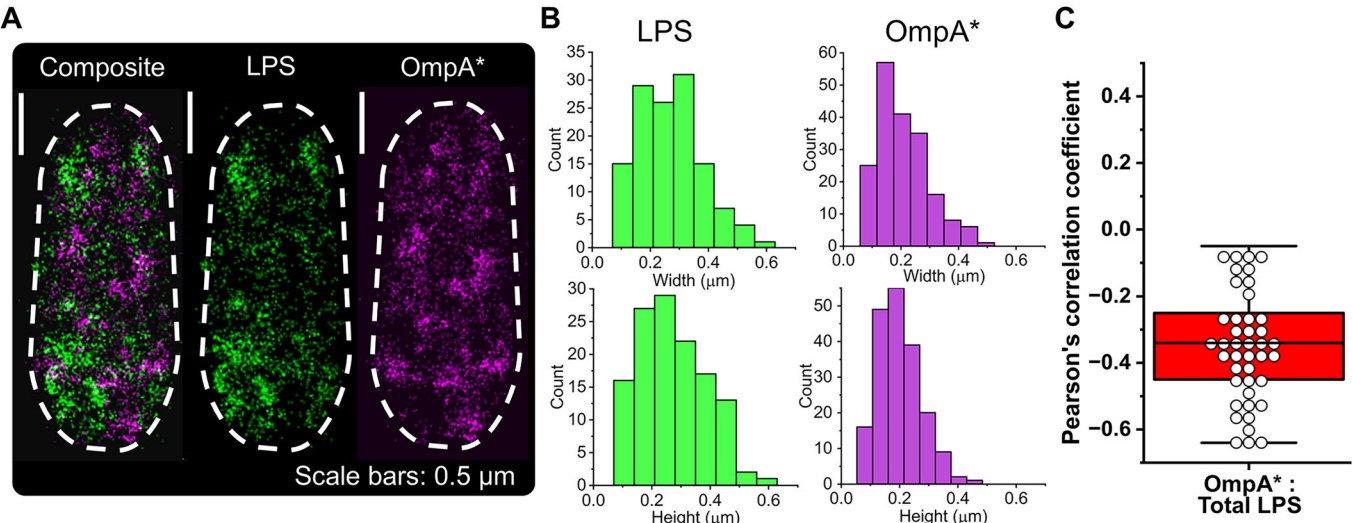

**Figure 2. Two-color direct stochastic optical reconstruction microscopy (dSTORM) reveals distinct LPS- and OMP-rich regions across the *E. coli* cell surface.**

(A) Representative two-color dSTORM images acquired using total internal reflection illumination showing the discrete clustering of AZ488-labeled LPS and AZ647-labeled recombinant non-canonical amino acid (ncAA) containing OmpA* in the OM of an *E. coli* Δ*ompA* cell. The discrete clustering of these abundant OM components, which is apparent in the composite image, indicates that local enrichment of LPS molecules and OMPs can occur in the OM, resulting in their own respective supramolecular islands. Dashed border denotes the outline of the bacterial cell. Scale bars: 0.5 μm. (B) An automated thresholding algorithm was used to identify LPS- and OMP-rich regions or islands in the respective fluorescence channels, and the height (longest axis) and width (shortest axis perpendicular to long axis) were measured manually for all patches >50 nm in size. Histograms of the width and height measurements for individual LPS islands ($n = 127$ islands from 31 cells) and OMP islands ($n = 190$ islands from 30 cells) indicated that a broad range of island sizes existed in the OM. The median size of the LPS islands (median width: $0.260 \pm 0.010$ μm, median height: $0.267 \pm 0.009$ μm) was slightly greater than the OMP islands (median width: $0.189 \pm 0.006$ μm, median height: $0.180 \pm 0.006$ μm). All experiments were done in triplicate using independent biological samples. (C) Co-localization analysis of two-color dSTORM data confirmed the strong negative correlation (median Pearson correlation coefficient = $-0.35$) between the surface location of most fluorescently labeled LPS and OmpA* in *E. coli* Δ*ompA* cells. All experiments were done in triplicate using independent biological samples ($n = 42$). In the box plot, the center line is the median while the box defines the upper and lower quartiles of the data and the whiskers enclose the interquartile range (i.e., middle 50% of the data). Source data are available online for this figure.

2015). This implies that the separate LPS-rich and OMP-rich regions of the OM may respond differently to the decrease in OM stiffness and stability present in the Δ*lpp* strain. The Δ*lpp* strain also has a hypervesiculation phenotype where patches of OM are lost as OM vesicles (Sonntag et al, 1978), with OM vesicle (OMV) biogenesis biased toward LPS-rich bilayer regions lacking Braun's lipoprotein (Guerrero-Mandujano et al, 2017). Comparable increases in the lateral mobility of LPS were observed in the Δ*lpp* and *imp4213* strains (Fig. 3D), even though the origin of the OM perturbations is very different in these strains. Since the majority of LPS remained immobile in these mutants, as determined by FRAP (Fig. 3D), it is possible that the LPS-rich supramolecular assemblies in the OM incorporate redundant biophysical interactions that organize and stabilize them.

## Divalent cation-mediated interactions influence LPS mobility

LPS is an anionic glycolipid with a formal charge density of up to 1.0 per hydrocarbon chain (Nikaido, 2003), which is greater than PL where formal negative charge density per hydrocarbon chain is 0.5 (Nikaido, 2003). Divalent cation ($Ca^{2+}$ and $Mg^{2+}$) coordination by negatively charged substituents ($PO_4^{2-}$ and $CO_2^-$) within the conserved Lipid A-$(Kdo)_2$ core of LPS have been proposed to neutralize these potentially repulsive charges enabling tighter LPS packing in the OM (Kucerka et al, 2008; Rahnamoun et al, 2020) and are therefore considered critical for maintenance of OM

asymmetry, stability, and integrity (Clifton et al, 2015). However, our current understanding of the affinity of LPS molecules for specific divalent cations and the resulting influence on ionic bridge formation between adjacent anionic oligosaccharide head groups in the OM relies heavily on in silico modeling (Brandner et al, 2024; Clifton et al, 2015; López et al, 2020; Vaiwala and Ayappa, 2024). For instance, a recent computational study concluded that $Ca^{2+}$ (rather than $Mg^{2+}$) mediates interactions between the $PO_4^{2-}$ groups of β-1,6-glucosamine disaccharides in Lipid A and exerts the greatest stabilizing influence on LPS in the OM (Clifton et al, 2015). But current models are limited by the complexities of LPS, particularly the chemical heterogeneity of smooth phenotypes consisting of O-antigen-containing LPS of different lengths. This is significant because most of the Gram-negative bacteria that exist outside the laboratory—such as those that occupy pathological niches—produce O-antigen-containing smooth LPS as a key virulence factor (Raetz and Whitfield, 2002). Since FRAP results (Fig. 1E,F) demonstrated deep rough, rough, and smooth LPS were all tightly confined in the OM, we reasoned that mobility-defining interactions must involve groups common to all these LPS glycoforms. This potentially includes divalent cation-mediated interactions within the conserved, negatively charged Lipid A-$(Kdo)_2$ motif at the base of LPS.

To explore this hypothesis, we briefly incubated cells presenting deep rough, rough, and smooth LPS with divalent cation chelating agents EDTA or EGTA at ambient temperature (20 °C) after fluorescent labeling of the LPS by the CuAAC approach, and just

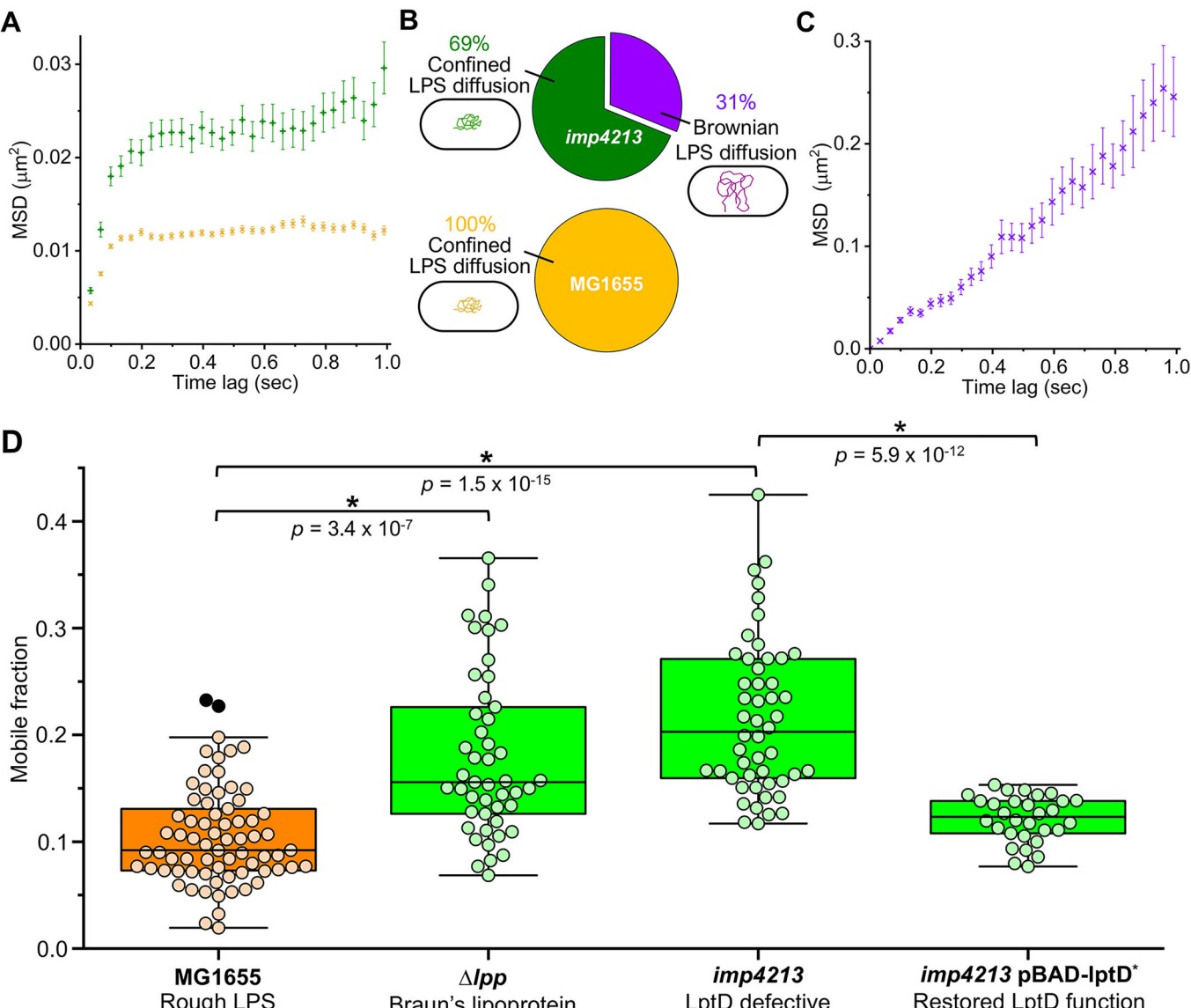

**Figure 3. OM asymmetry, LPS–LPS intermolecular interactions, and OM-PG associations contribute to LPS confinement in the OM of *E. coli*.**

(A) Mean-squared displacement (MSD) was calculated for single AF488-labeled LPS molecules that could be tracked in the OM for at least 1 s before photobleaching (error is reported as S.E.M.). TIRFM video data were collected at 30 Hz from a minimum of three independent biological replicates. The MSD for all rough LPS in *E. coli* MG1655 cells (orange ×, $n = 156$ tracks) and most rough LPS in *E. coli imp4213* cells (green +, $n = 104$ tracks) rapidly approached an asymptotic value which was consistent with confined lateral diffusion. Linear regression of the MSD for the first four time delays yielded lateral diffusion coefficients ($D_{2D}$) of ~0.0182 μm²/s and ~0.0347 μm²/s, respectively, for rough LPS in these cells. (B) The majority of rough LPS molecules (69%) tracked on *E. coli imp4213* cells displayed confined lateral diffusion. The remainder of the tracked LPS molecules (31%) displayed free Brownian lateral diffusion. All LPS molecules tracked on wild-type *E. coli* MG1655 cells displayed confined lateral diffusion. SPT of AF488-labeled LPS molecules was not possible with the Δ*lpp* strain due to a high background fluorescence in these samples. (C) The MSD ( ± S.E.M.) for freely diffusing rough LPS in *E. coli imp4213* cells (violet ×, $n = 49$ tracks) increased linearly with time, and linear regression of the first 13 time delays yielded $D_{2D} \approx 0.0550$ μm²/s. TIRFM video data were collected at 30 Hz from a minimum of three independent biological replicates. (D) FRAP-derived mobile fractions for rough LPS in the OM of *E. coli* MG1655, Δ*lpp*, and LptD-defective *imp4213* cells were compared to *E. coli imp4213* pBAD-lptD* cells 2 h post-initiation of functional LptD* production. Statistically significant increases in LPS mobile fractions (by Mann–Whitney test) were observed in Δ*lpp* (median = 0.16, $n = 47$) and *imp4213* (median = 0.21, $n = 49$) cells relative to wild-type *E. coli* MG1655 cells (median = 0.09, $n = 74$). Restoration of OM asymmetry via production of functional recombinant LptD* in *imp4213* pBAD-lptD* cells (median = 0.12, $n = 30$) triggered a reduction in LPS lateral mobility to near wild-type levels. For each box plot, the center line is the median, while the box defines the upper and lower quartiles of the data, and the whiskers enclose the interquartile range (i.e., middle 50% of the data). Each symbol represents an individual measurement with black-filled symbols classified as outliers (i.e., outside the interquartile range). Source data are available online for this figure.

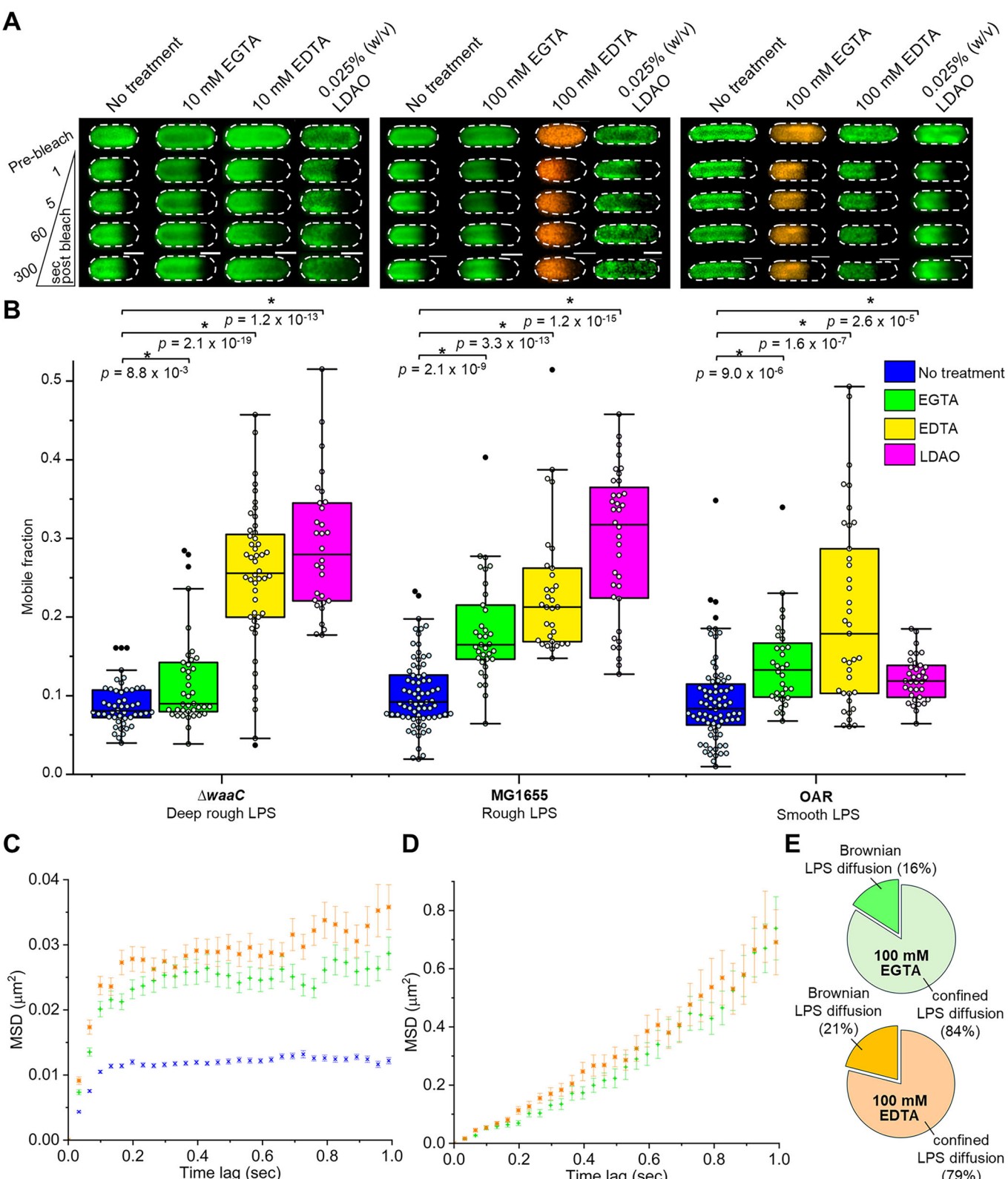

◀ **Figure 4. Confined lateral diffusion of LPS in the OM of *E. coli* is influenced by divalent cation-mediated interactions and hydrophobic interactions within the membrane.**

(A) Representative FRAP sequences for AF488-LPS (green) or AF568-LPS (orange) in the OM of deep rough LPS-producing *ΔwaaC* (left), rough LPS-producing MG1655 (middle), and smooth LPS-producing OAR (right) *E. coli* cells without treatment, and after treatment with chelator (EGTA or EDTA) or detergent (0.025% (w/v) LDAO). Dashed border denotes the outline of each bacterial cell. Scale bars: 1.0 μm. (B) Comparison of FRAP-derived LPS mobile fractions in *E. coli* cells producing different LPS glycoforms (statistical analysis done by Mann–Whitney test): *ΔwaaC* (untreated, $n = 49$; 10 mM EGTA, $n = 39$; 10 mM EDTA, $n = 48$; LDAO, $n = 30$), MG1655 (untreated, $n = 74$; 100 mM EGTA, $n = 35$; 100 mM EDTA, $n = 31$; LDAO, $n = 37$), and OAR (untreated, $n = 74$; 100 mM EGTA, $n = 32$; 100 mM EDTA, $n = 37$; LDAO, $n = 35$). The FRAP sequences and mobile fraction data for the untreated *ΔwaaC*, MG1655, and OAR *E. coli* cells were reused from Fig. 1E,F. For each box plot, the center line is the median, while the box defines the upper and lower quartiles of the data, and the whiskers enclose the interquartile range (i.e., middle 50% of the data). Each symbol represents an individual measurement with black-filled symbols classified as outliers (i.e., outside the interquartile range). (C) Collection of TIRFM video data, and the calculation and analysis of the MSD (± S.E.M.) for single AF488-labeled LPS molecules in the OM of *E. coli* MG1655 cells was done as described in Fig. 3. The MSD for all rough LPS in untreated cells (blue ×, $n = 156$ tracks) rapidly approached an asymptotic value, which was consistent with confined lateral diffusion. The majority of rough LPS molecules tracked on 100 mM EDTA (orange ✳, $n = 118$ tracks) and 100 mM EGTA (green +, $n = 108$ tracks) treated cells also had an asymptotic MSD value (≈0.03 μm²), which was consistent with confined lateral diffusion, but the degree of confinement was reduced relative to untreated cells (≈ 0.01 μm²). Linear regression of the MSD yielded $D_{2D} ≈ 0.0377$ μm²/s and $D_{2D} ≈ 0.0373$ μm²/s for confined rough LPS in EDTA- and EGTA-treated cells, respectively. These values are increased relative to untreated cells ($D_{2D} ≈ 0.0182$ μm²/s). LPS confinement was globally reduced in MG1655 cells after these chelator treatments due to a two-fold increase in $D_{2D}$ and a three-fold increase in the confinement diameter. (D) The MSD (± S.E.M.) for rough LPS molecules tracked on 100 mM EDTA-treated (orange ✳, $n = 31$ tracks) and 100 mM EGTA-treated (green +, $n = 21$ tracks) cells increased linearly with time, which indicated free Brownian diffusion. Linear regression yielded $D_{2D} ≈ 0.153$ μm²/s and $D_{2D} ≈ 0.113$ μm²/s, respectively, for these cells. (E) Percentage of rough LPS molecules observed to undergo confined and free Brownian lateral diffusion in the OM of 100 mM EDTA-treated and 100 mM EGTA-treated cells. All FRAP and SPT experiments were done in triplicate using independent biological samples. Source data are available online for this figure.

prior to FRAP analysis. EDTA and heat (37 °C) treatment is known to disrupt LPS in the OM and increase the permeability of bacterial cells (Leive, 1968; Leive et al, 1968), nevertheless chelator treatment is an easy way to assess the relative importance of divalent cations on overall LPS confinement. Our chelator concentrations (i.e., 10 mM for deep rough LPS strain, 100 mM for rough and smooth LPS strains) and treatment conditions were carefully selected for each strain to ensure the cells retained their normal rod-like morphology (Appendix Fig. S5). This was confirmed for each cell by DIC microscopy before and after the FRAP experiment. Although both chelating agents coordinate $Mg^{2+}$ and $Ca^{2+}$, EGTA has a five-fold higher affinity for $Ca^{2+}$ compared to $Mg^{2+}$, while EDTA has approximately equivalent affinities for both $Mg^{2+}$ and $Ca^{2+}$ (Moeschler et al, 1980; Qin et al, 1999). Comparison of the effects of EDTA versus EGTA on median LPS lateral mobility therefore provided insights into the relative influence of $Mg^{2+}$ (EDTA treatment) and $Ca^{2+}$ (EGTA treatment) mediated interactions on LPS confinement. Repeating experiments with strains producing deep rough (*ΔwaaC*), rough (MG1655), or smooth LPS (OAR) provided insight into the relative importance of divalent cation-mediated interactions compared to other potential forms of intermolecular interactions (such as hydrogen bonds) likely to exist between the oligosaccharide domains of adjacent LPS molecules.

These FRAP experiments confirmed that divalent cation-mediated interactions between adjacent LPS molecules do contribute toward LPS lateral restriction in *E. coli*, with statistically significant positive shifts in LPS mobile distributions observed for all *E. coli* strains after EDTA or EGTA treatments (Fig. 4A,B; Appendix Table S1). The increase in LPS mobility observed by FRAP for the rough LPS strain after treatment with these chelators is larger for EGTA when compared to the deep rough strain (Fig. 4B). This suggests the phosphate groups in the inner core oligosaccharide domain (Fig. 1D) contribute to LPS lateral restriction by binding divalent cations, with a potential preference for $Ca^{2+}$, which bridge the anionic groups in neighboring rough LPS molecules. It is also easier to remove $Mg^{2+}$ and $Ca^{2+}$ from the OM with chelators for the rough LPS strain compared to the

smooth LPS strain, likely due to steric exclusion by the extended oligosaccharide in the LPS O-antigen repeats. Interestingly, the chelator treatment does not further exacerbate the increased LPS mobility in *imp4213* cells measured by FRAP (Fig. EV4; Appendix Table S1). The apparent insensitivity of LPS mobility in *imp4213* cells to chelator treatments suggests that loss of OM asymmetry and tight packing of adjacent LPS molecules also contribute to disruption of divalent cation-mediated LPS–LPS interactions.

In addition, SPT of video data for fluorescently labeled LPS in the MG1655 strain after treatment with EDTA or EGTA revealed two populations of LPS molecules with distinct diffusive characteristics (Fig. 4C,E). The majority of the LPS species remained confined when treated with chelator (79% for EDTA, 84% for EGTA); however, a new population of freely diffusing LPS molecules was also observed (21% for EDTA, 16% for EGTA) (Fig. 4E). The lateral diffusion coefficient of the confined LPS species was similar for both chelators (EDTA: $D_{2D} = 0.0377$ μm²/s; EGTA: $D_{2D} = 0.0373$ μm²/s) and slightly increased relative to untreated cells (Fig. 4D; Table EV1). The confinement diameter for LPS also increased in the MG1655 strain upon chelator treatment (EDTA: $0.815 ± 0.0238$ μm; EGTA: $0.748 ± 0.0200$ μm), yielding values equivalent to what was observed for confined LPS species in the *imp4213* strain (Table EV1). In our previous study, a fluorescent OMP-labeling strategy was used to follow the lateral diffusion of single CirA receptors (a monomeric OMP) in the bacterial OM by SPT (Rassam et al, 2015), and it was used here to probe how chelator treatment affected lateral diffusion in OMP-rich regions. In striking contrast to LPS, all of the CirA receptors displayed confined diffusion in the presence of these chelators (Appendix Fig. S6) with the lateral diffusion coefficient (EDTA: $D_{2D} = 0.0454 ± 0.00340$ μm²/s; EGTA: $D_{2D} = 0.0402 ± 0.00471$ μm²/s) and confinement diameter (EDTA: $0.846 ± 0.0229$ μm; EGTA: $0.740 ± 0.0238$ μm) increased marginally relative to untreated cells ($D_{2D} = 0.0178 ± 0.000623$ μm²/s and $0.540 ± 0.00665$ μm, respectively)(Table EV1). These SPT results are again consistent with divalent cation depletion in the OM disrupting LPS–LPS interactions predominantly, while LPS–OMP interactions in OMP-rich regions remain mostly intact.

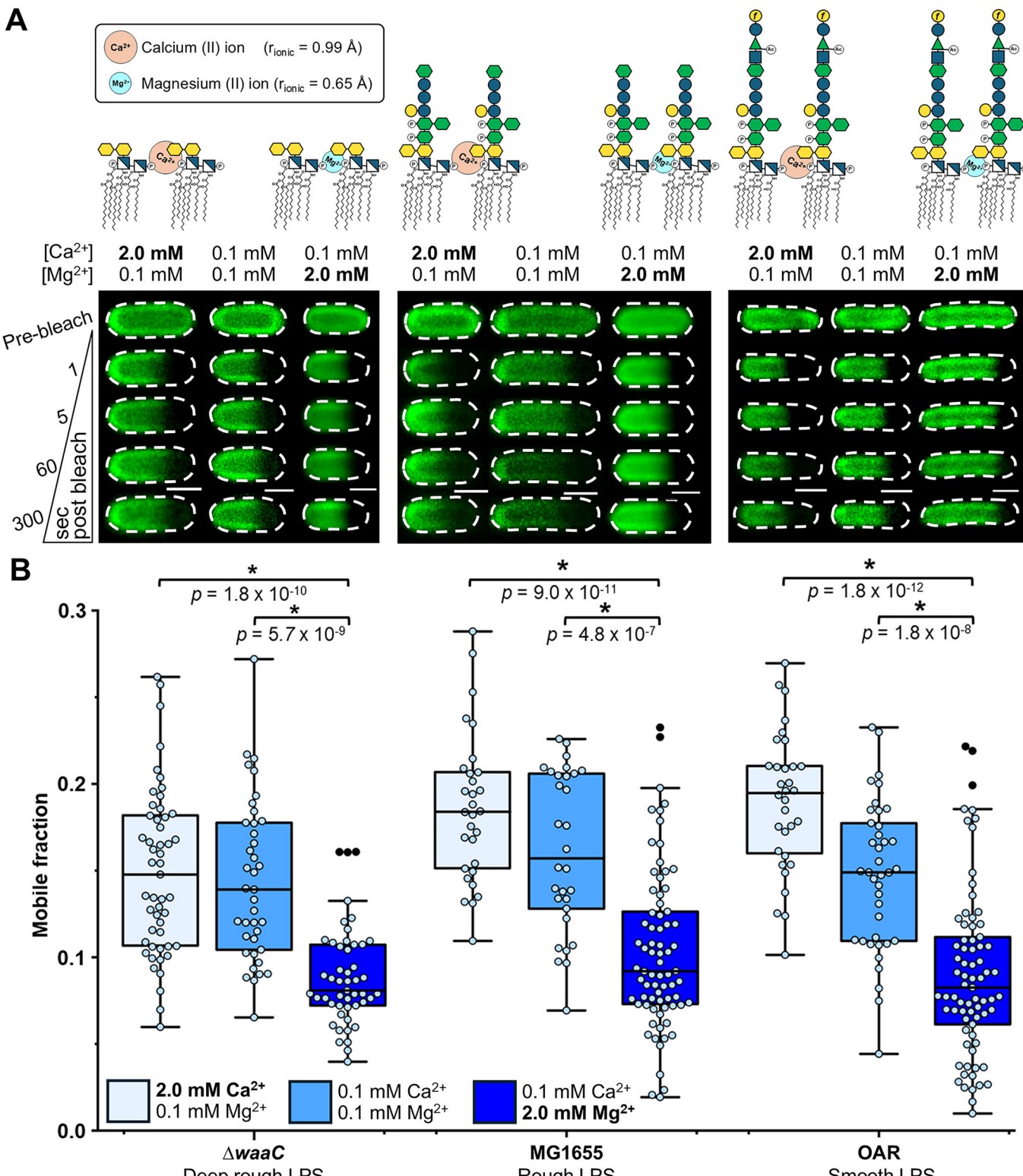

◀ **Figure 5.  A reduction in the Mg²⁺ ion concentration triggers an increase in LPS lateral mobility in the OM of *E. coli* cells.**

(A) Schematics of deep rough, rough, and smooth LPS glycoforms present in Δ*waaC*, MG1655, and O-antigen restored (OAR) *E. coli* strains, respectively, showing how calcium (II) and magnesium (II) ions can bridge adjacent LPS molecules by binding to phosphate groups within their conserved Lipid A cores. The smaller physical size (ionic radius = 0.65 Å) and higher charge density of the Mg²⁺ ion will permit tighter packing and stronger interactions between adjacent LPS molecules in the OM compared to the physically larger Ca²⁺ ion (ionic radius = 0.99 Å) with a lower charge density. Representative FRAP sequences for AF488-LPS in the OM of deep rough LPS-producing Δ*waaC* (left), rough LPS-producing MG1655 (middle), and smooth LPS-producing OAR (right) *E. coli* cells after culturing in chemically defined medium (CDM) with varied Mg²⁺ and Ca²⁺ ion concentrations at constant total divalent cation concentration. Dashed border denotes the outline of each bacterial cell. Scale bars: 1.0 µm. (B) Comparison of FRAP-derived LPS mobile fractions in *E. coli* cells producing different LPS glycoforms after culturing in CDM with varied Mg²⁺ and Ca²⁺ ion concentrations. The FRAP sequences and mobile fraction data for the Δ*waaC*, MG1655, and OAR *E. coli* cells cultured in low Ca²⁺, high Mg²⁺ CDM were reused from Fig. 1E,F. All FRAP experiments were done in triplicate using independent biological samples. For each box plot, the center line is the median, while the box defines the upper and lower quartiles of the data, and the whiskers enclose the interquartile range (i.e., middle 50% of the data). Each symbol represents an individual measurement with black-filled symbols classified as outliers (i.e., outside the interquartile range). Bacterial cells cultured in high Ca²⁺ (2.0 mM), low Mg²⁺ (0.1 mM) CDM or low Ca²⁺ (0.1 mM), low Mg²⁺ (0.1 mM) CDM displayed a statistically significant increase in the median LPS mobile fraction (by Mann–Whitney test) compared to cells cultured in low Ca²⁺, high Mg²⁺ CDM, irrespective of the LPS glycoform produced by the *E. coli* strain. This demonstrates that Mg²⁺-mediated LPS–LPS interactions exert a greater restrictive influence on LPS lateral mobility in the OM compared to Ca²⁺-mediated interactions. Source data are available online for this figure.

## Mg²⁺ rather than Ca²⁺ ions exert a greater influence on LPS lateral mobility

Notably, EDTA treatment triggered larger shifts in LPS mobile fractions compared to EGTA treatment (Fig. 4; Appendix Table S1), implying that Mg²⁺ mediated interactions exert a greater restrictive influence on LPS diffusion compared to Ca²⁺. Alternatively, it may be that EDTA is more effective at stripping OM-coordinated divalent cations compared to EGTA. Categorical determination of the relative influence of Mg²⁺ versus Ca²⁺-mediated OM interactions on LPS average mobility from these FRAP experiments alone is difficult because both EGTA and EDTA will chelate Ca²⁺ and Mg²⁺, and these chelators will affect OM permeability. Analyses are further complicated by the composition of the standard chemically defined medium (CDM) used in this work to facilitate metabolic labeling with Kdo-azide, which contains 2.0 mM Mg²⁺ and 0.1 mM Ca²⁺. Therefore, we conducted a series of experiments growing cells in CDM with altered Mg²⁺ and Ca²⁺ ion ratios (Fig. 5A) after verification that these changes did not limit bacterial growth or alter cell morphology (Appendix Fig. S7). The lowest Mg²⁺ ion concentrations employed in these experiments were not growth-limiting (Rao and Igoshin, 2021). We observed that LPS mobile fraction distributions in the OM of cells cultured in low Mg²⁺ (0.1 mM), low Ca²⁺ (0.1 mM) CDM were significantly elevated compared with those measured in cells cultured in standard, high Mg²⁺, low Ca²⁺ CDM (Fig. 5; Appendix Tables S1 and S2). Restoration to original levels of LPS restriction in standard CDM (with high Mg²⁺ and low Ca²⁺) were not observed when the Ca²⁺ concentration was increased to 2.0 mM. The slightly higher LPS mobility observed by FRAP in all high Ca²⁺, low Mg²⁺ CDM conditions relative to low Mg²⁺, low Ca²⁺ CDM (Fig. 5B) suggests that preferentially bound Mg²⁺ can be displaced by a higher concentration of Ca²⁺. These results support our initial hypothesis that Mg²⁺ mediated interactions exert a greater restrictive influence on LPS lateral diffusion compared to Ca²⁺ interactions, and reinforce the conclusions drawn from chelator treatment experiments (Fig. 4).

## Hydrophobic forces within the OM contribute to restricted LPS lateral mobility

Even with chelator treatment or reduced Mg²⁺ concentration the majority of LPS molecules in the OM outer leaflet remain immobile

as determined by FRAP irrespective of LPS oligosaccharide domain size (Figs. 4 and 5). Under standard conditions, most species of Gram-negative bacteria produce hexa-acylated Lipid A in LPS (Fig. EV1), with all six acyl chains being saturated, facilitating tight packing of adjacent LPS molecules whilst providing a large potential area for hydrophobic interactions (e.g., van der Waals contacts)(Simpson and Trent, 2019). We therefore theorized these interactions might also contribute to our observed high levels of LPS restriction. To investigate, we subjected cells to detergent treatment at a sub-lytic concentration (Fig. 4A,B), which did not inhibit bacterial growth or alter cell morphology (Appendix Fig. S8). *N,N*-dimethyldodecylamine N-oxide (LDAO) is a zwitterionic, OM-targeting detergent used for selective isolation of OMPs with the ability to trigger changes in LPS monolayer structure (Heinz and Niederweis, 2000; Newstead et al, 2008). We observed that pre-FRAP treatment with 0.025% (w/v) LDAO triggered positive shifts in the LPS mobile fraction distributions of all three *E. coli* strains (Fig. 4A,B), confirming that interactions within the hydrophobic environment of the OM do contribute to LPS restriction. As observed in FRAP experiments after chelator treatment, the size of positive mobility shifts was influenced by LPS oligosaccharide domain structure, with larger increases observed in cells with deep rough or rough LPS, highlighting the steric protective effect the extended O-antigen affords to the conserved Lipid A-(Kdo)₂ motif of LPS. The restrictive intermolecular O-antigen-mediated hydrogen-bonding network, which likely scales with core oligosaccharide length (Fitzmaurice et al, 2025), may also offset the destabilizing impact of LDAO molecules that do penetrate through the O-antigen. We tested whether treatment with a sub-lytic concentration of chaotropic agent (300 mM urea is a physiological concentration (Shaykhutdinov et al, 2009)) disrupted the LPS oligosaccharide domain structure and increased lateral mobility, but only a minor effect on mobility was observed (Appendix Fig. S9). Our experimentally derived relationship between LPS oligosaccharide domain size and OM detergent susceptibility reinforces previous studies that have emphasized the importance of LPS O-antigens in conveying enhanced resistance to bile salts in the context of GI tract infections (Crawford et al, 2012).

The increase in lateral mobility of LPS in the deep rough (Δ*waaC*) strain induced by 0.025% (w/v) LDAO treatment is consistent with the detergent molecule disrupting the optimal hydrophobic packing between LPS molecules. As LDAO is used to solubilize OMPs (Heinz and Niederweis, 2000; Newstead et al,

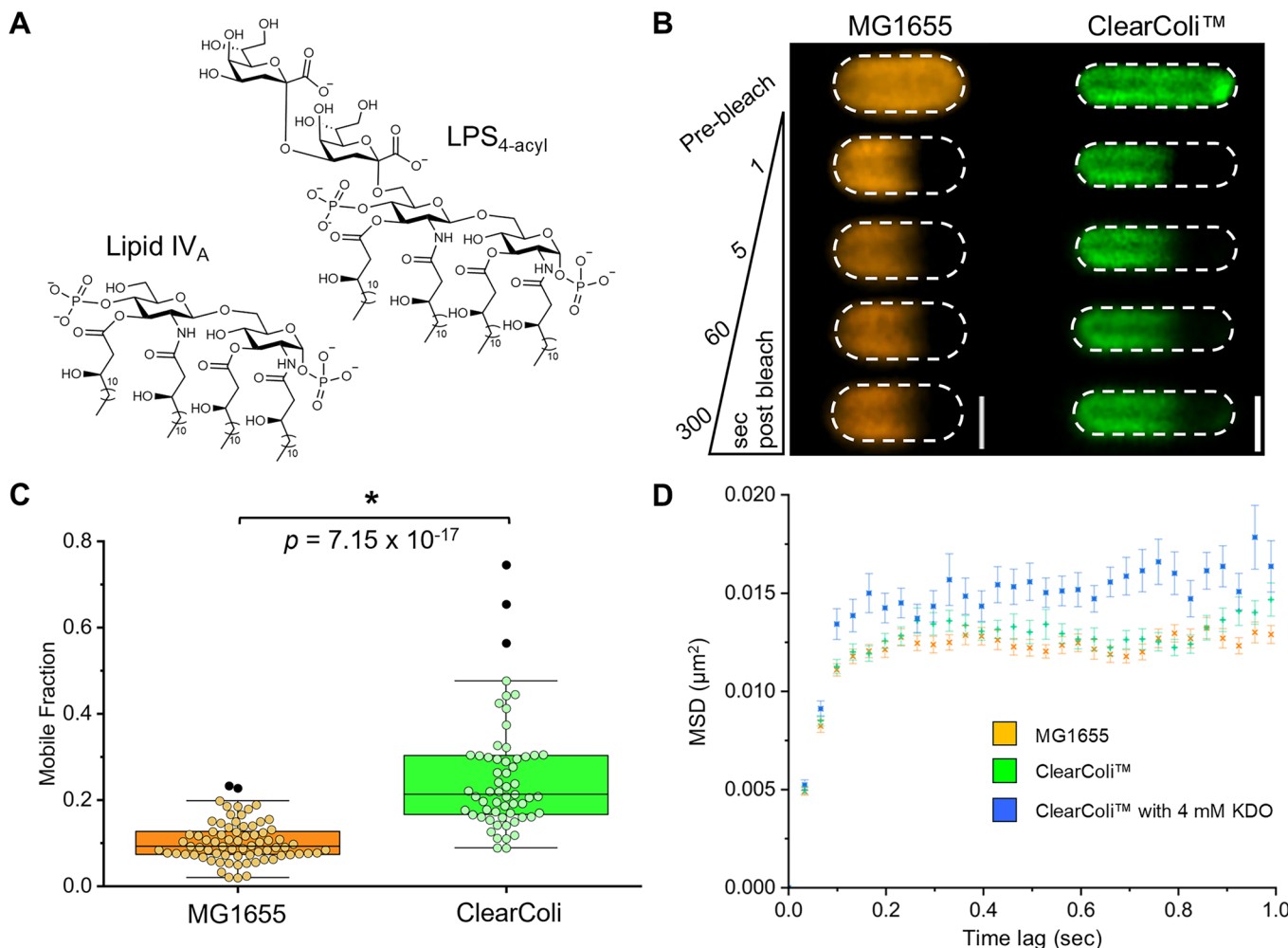

**Figure 6. Reduction in acyl chain-mediated intermolecular contacts between LPS molecules causes an increase in lateral mobility as detected by FRAP and SPT-TIRFM.**

(A) ClearColi™ strain inserts an LPS-like molecule in the outer leaflet of the OM, which is tetra-acylated Lipid IV$_A$. Metabolic labeling with Kdo or Kdo-azide produces a deep rough-like LPS with 4 acyl chains (LPS$_{4-acyl}$). The chemical structures of Lipid IV$_A$ and deep rough-like LPS (LPS$_{4-acyl}$) produced by ClearColi™ in the absence (Lipid IV$_A$) or presence (LPS$_{4-acyl}$) of metabolic labeling with Kdo sugar are presented here. (B) Representative time-lapse FRAP images monitoring fluorescence recovery in the photobleached region of the OM for *E. coli* MG1655 cells with AF568-labeled rough LPS and ClearColi™ cells with AF488-labeled LPS$_{4-acyl}$ (cells grown with 4 mM Kdo-azide). Dashed border denotes the outline of each bacterial cell. Scale bars: 1.0 μm. (C) Lateral mobility of AF488- and AF568-labeled LPS measured by FRAP. LPS mobile fractions in the OM of *E. coli* MG1655 (rough LPS) and ClearColi™ grown with Kdo-azide (deep rough-like, tetra-acylated Lipid IV$_A$). A statistically significant increase (by Mann–Whitney test) in the LPS mobile fraction was observed in the ClearColi™ strain (median = 0.212, $n = 58$) relative to the wild-type *E. coli* MG1655 strain (median = 0.092, $n = 74$). All FRAP experiments were done in triplicate using independent biological samples. For each box plot, the center line is the median, while the box defines the upper and lower quartiles of the data, and the whiskers enclose the interquartile range (i.e., middle 50% of the data). Each symbol represents an individual measurement, with black-filled symbols classified as outliers (i.e., outside the interquartile range). (D) Mean-squared displacement (MSD) was calculated for single AF488-labeled colicin Ia/CirA receptor complexes that could be tracked for at least 1 s before photobleaching (error reported as S.E.M.). All total internal reflection fluorescence microscopy video data was collected at 30 Hz from a minimum of three independent biological replicates, except for ClearColi™ cells grown with 4 mM Kdo, where two independent biological replicates were used. The MSD for all OMP complexes on *E. coli* MG1655 cells (orange ×, $n = 84$ tracks), ClearColi™ cells grown without Kdo sugar (green +, $n = 90$ tracks), and ClearColi™ cells grown with 4 mM Kdo sugar (blue ×, $n = 78$ tracks) approached an asymptotic value, which was consistent with confined lateral diffusion. Linear regression of the MSD for the first 4 time delays yielded $D_{2D} \approx 0.0178\ \mu m^2/s$, $D_{2D} \approx 0.0181\ \mu m^2/s$ and $D_{2D} \approx 0.0229\ \mu m^2/s$, respectively, for these cells. These results demonstrate that changes in acyl chain-mediated interactions between Lipid IV$_A$ molecules in the outer leaflet of the OM primarily affect LPS–LPS interactions, with minimal effect on LPS–OMP interactions. Source data are available online for this figure.

2008; Young et al, 2023), the increases in LPS mobility may also result from the preferential disruption of LPS–OMP interactions rather than LPS–LPS interactions. We therefore utilized an endotoxin-free *E. coli* strain (ClearColi™)(Mamat et al, 2015), which incorporated tetra-acylated Lipid IV$_A$ (Fig. 6A), rather than hexa-acylated Lipid IV$_A$, to dissect whether the detergent was

indeed disrupting hydrophobic LPS–LPS interactions. Once metabolically labeled with Kdo-azide, the tetra-acylated Lipid IV$_A$ (termed LPS$_{4-acyl}$) has an oligosaccharide headgroup (two Kdo sugars) equivalent to the native deep rough LPS (Fig. 6A). However, we hypothesized that the LPS$_{4-acyl}$ would prevent optimal acyl chain packing and increase lateral mobility. We observed exactly

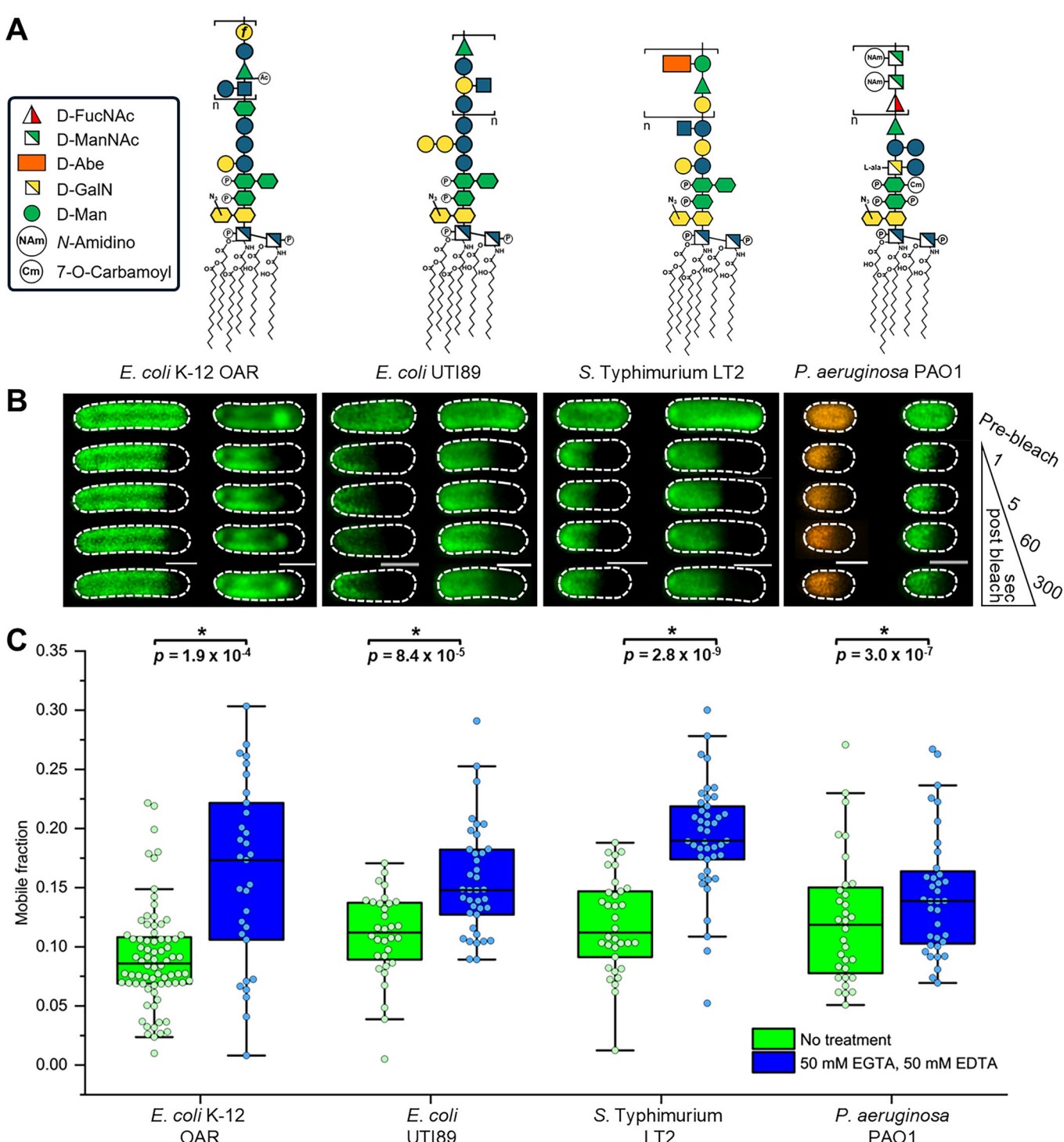

this result for the LPS$_{4\text{-acyl}}$ strain in the standard CDM by FRAP (Fig. 6B,C). The observed LPS$_{4\text{-acyl}}$ lateral mobility in the ClearColi™ strain was like that observed in rough (MG1655) and deep rough (ΔwaaC) strains after 0.025% (w/v) LDAO treatment (Fig. 4A,B; Appendix Table S1). These results are consistent with an increase in the lateral mobility of LPS due to sub-optimal acyl chain packing. To ascertain whether the OMP-rich regions of the OM were also perturbed in the ClearColi™ strain, we used fluorescent

labeling to track the lateral diffusion of the CirA receptor with SPT in the ClearColi™ strain, before and after metabolic labeling with Kdo sugar (Fig. 6D). The observed lateral diffusion coefficients ($D_{2D} = 0.0181 \pm 0.000820\ \mu m^2/s$ and $D_{2D} = 0.0229 \pm 0.00206\ \mu m^2/s$) and confinement diameters ($0.546 \pm 0.00760\ \mu m$ and $0.600 \pm 0.0124\ \mu m$) of this OMP in the ClearColi™ strain before and after metabolic labeling with Kdo sugar were nearly indistinguishable from that observed for the MG1655 strain

**Figure 7. Divalent cation-dependent lateral confinement of LPS is observed in pathogenic Gram-negative bacteria, irrespective of O-antigen and capsule-forming serotypes.**

(A) Schematics of smooth LPS glycoforms present in *E. coli* K-12 O-antigen restored (OAR), *E. coli* UTI89, *S.* Typhimurium LT2, and *P. aeruginosa* PAO1 strains showing the composition of the oligosaccharide domain displayed at the cell surface. The *E. coli* UTI89 and *P. aeruginosa* PAO1 strains produce a capsular polysaccharide, which was removed prior to fluorescent-labeling of the LPS. (B) Representative FRAP sequences for AF488-LPS (green) or AF568-LPS (orange) in the OM of smooth LPS-producing strains before (left-hand image sequence) and after (right-hand image sequence) combined 50 mM EDTA and 50 mM EGTA treatment. The combined chelator treatment was found to be as effective as separate treatments with 100 mM EDTA and 100 mM EGTA at removing coordinated $Mg^{2+}$ and $Ca^{2+}$ ions from the OM. Dashed border denotes the outline of each bacterial cell. Scale bars: 1.0 μm. (C) Comparison of FRAP-derived LPS mobile fractions for fluorescently labeled-LPS in the OM of *E. coli* K-12 OAR ($n_{pre-treatment} = 74$, $n_{post-treatment} = 31$), *E. coli* UTI89 ($n_{pre-treatment} = 32$, $n_{post-treatment} = 38$), *S.* Typhimurium LT2 ($n_{pre-treatment} = 33$, $n_{post-treatment} = 43$), and *P. aeruginosa* PAO1 ($n_{pre-treatment} = 30$, $n_{post-treatment} = 30$) cells pre- and post-treatment with the combined chelators. The pre-treatment FRAP sequences and mobile fraction data for the *E. coli* K-12 OAR strain were reused from Fig. 1E,F. All FRAP experiments were done in triplicate using independent biological samples. For each box plot, the center line is the median while the box defines the upper and lower quartiles of the data and the whiskers enclose the interquartile range (i.e., middle 50% of the data). Each symbol represents an individual measurement with black-filled symbols classified as outliers (i.e., outside the interquartile range). LPS lateral mobility is tightly confined in the *E. coli* UTI89, *S.* Typhimurium LT2, and *P. aeruginosa* PAO1 pathogenic strains. The combined chelator treatment triggered a statistically significant increase in the lateral mobility of the fluorescently labeled LPS (by Mann–Whitney test) in the OM of each O-antigen producing strain, irrespective of the oligosaccharide domain composition and the degree of acylation in the Lipid A core. Source data are available online for this figure.

(Fig. 6D; Table EV1). This emphasizes the importance of Lipid A acyl chain number in determining the fluidity of the LPS-rich OM regions, while intimating that the packing of LPS molecules within OMP-rich regions may be different and highly influenced by LPS–OMP interactions.

## LPS confinement is conserved in pathogenic Gram-negative bacteria

Although the general structure of Lipid A and inner core oligosaccharides is highly conserved across Gram-negative bacteria, significant interspecies differences in outer core and O-antigen repeat structure exist (Amor et al, 2000; Bertani and Ruiz, 2018; Lerouge and Vanderleyden, 2002). Having demonstrated that, under chemical challenge, differences in *E. coli* K-12 LPS oligosaccharide domain structure influence average LPS mobility, we elected to carry out selected FRAP experiments on other Gram-negative bacteria. We characterized LPS mobility in *Salmonella enterica* serovar Typhimurium LT2, *Pseudomonas aeruginosa* PAO1, and *E. coli* UTI89 strains to ascertain the extent to which observations for *E. coli* K-12 could be extrapolated across important pathogens within the Gram-negative clade. *P. aeruginosa* PAO1 and *E. coli* UTI89 exhibit a capsular serotype, a proven virulence factor that we found prevented CuAAC and SPAAC "click"-labeling of LPS with water-soluble fluorophores. To enable fluorophore bioconjugation to Kdo-N3 in the LPS of these capsule-forming strains, we therefore developed an adapted labeling protocol. This involved capsular polysaccharide stripping using EDTA, followed by the restoration of original OM-coordinated divalent cation concentrations via "bathing" of "stripped" cells in freshly supplemented CDM, prior to fluorescent labeling by the CuAAC approach. This method enabled comparison of average LPS mobilities and the impact of chelator treatments in bacteria with and without capsular serotypes.

The level of restricted lateral mobility for LPS was roughly equivalent in all strains, irrespective of whether cells had capsule-forming or non-capsule-forming serotypes (Fig. 7A–C; Appendix Tables S1 and S3). These experiments reinforce the hypothesis that LPS confinement is a universal trait conserved across Gram-negative bacterial species with conventional OM physiology, including those that occupy pathological niches. Similarly, the restrictive influence of divalent cation-mediated interactions in the

OM on lateral LPS mobility is not a trait exclusive to *E. coli* K-12 laboratory strains, as pre-FRAP treatment with 50 mM EDTA and 50 mM EGTA triggered a statistically significant increase in LPS mobility in all bacteria (Fig. 7B; Appendix Table S1). This further supports our assertion that the molecular basis of LPS confinement resides in the structure of Lipid A and the inner core oligosaccharide domain, and explains why these structural motifs are highly conserved across the Gram-negative bacterial clade (Raetz and Whitfield, 2002).

## Discussion

Our characterization of LPS lateral mobility using in vivo ensemble-averaged (FRAP) and single-molecule (SPT) fluorescence microscopy confirms that this essential glycolipid is tightly restricted in the native OM of *E. coli*, in agreement with recent work (Kumar et al, 2025), but also in other pathogenic and non-pathogenic Gram-negative bacteria. Remarkably, the observed confinement is not dependent on oligosaccharide headgroup size or structure (Fig. 2), and this includes LPS from the K and O serotype groups (Fig. 7). Rather, we identify saturated acyl chains and bridging divalent cations at the base of LPS as principally responsible for LPS lateral confinement in the OM. We measured the lateral diffusion coefficient for confined LPS molecules ($D_{2D} \approx 0.02$ μm²/s) by performing SPT measurements on this glycolipid in the native *E. coli* OM for the first time (Table EV1). Our $D_{2D}$ values for confined LPS were very similar to those measured by SPT for confined BODIPY-labeled LPS in reconstituted polymer-supported proteolipid membrane bilayers (Rassam et al, 2015), though we did not observe freely diffusing LPS molecules in the native OM as was observed in these artificial bilayers.

The confinement diameters estimated from SPT data for deep rough and rough LPS, and for the CirA receptor with these glycoforms, were very similar ($\approx 0.54$ μm) (Table EV1) and are consistent with the formation of LPS-rich and OMP-rich regions as observed in our dSTORM data (Fig. 2B). FRAP measurements of LPS lateral mobility were performed by consistently photobleaching a region (1.37 μm × 0.74 μm) equivalent to a circular spot with a diameter of ~0.58 μm; thus, the bleach region diameter is equivalent to the measured confinement diameter. Therefore, the

low mobility measured by FRAP in the native OM (median mobile fraction ≈0.1, Fig. 1E,F) is consistent with the degree of confinement measured by SPT (~ 0.54 µm) (Table EV1). Furthermore, the measurements of LPS lateral mobility made for chemically challenged cells also yielded consistent increases in both the lateral diffusion coefficient ($D_{2D}$) and confinement diameter measured by SPT (Table EV1) and the mobile fraction measured by FRAP (Figs. 3 and 4).

Our FRAP, SPT, and super-resolution imaging data support the view that the native OM is a mosaic of OMP-rich and LPS-rich regions (Benn et al, 2021). OMP-rich regions in the OM are in supramolecular structures (Rassam et al, 2015), where each OMP is surrounded by a ring-like shell of asymmetric lipid, which mediates strong interactions between neighboring OMPs (Webby et al, 2022), thus stabilizing these regions in the OM relative to the OMP-devoid LPS-rich patches. Our data also clearly demonstrate that increases in OM permeability, loss of OM lipid asymmetry or deletion of Braun's lipoprotein OM-PG associations manifest as a reduction in the level of LPS confinement in the LPS-rich patches. The abundant Braun's lipoprotein may act as a static membrane lipid in the inner leaflet of the native OM bilayer, thus exerting drag on other membrane lipids and contributing indirectly to LPS immobility. The hypervesiculation observed with the Δ*lpp* strain (Sonntag et al, 1978) could somewhat compensate for the loss of this membrane lipid anchoring effect by removing LPS-rich regions lacking OM-PG linkages via OMV formation. Further work is required to fully understand how vesiculation might influence global LPS mobility in the native OM.

We showed that divalent cation-mediated interactions involving charged groups at the base of the LPS inner core oligosaccharide domain exerted a significant influence on LPS lateral diffusion. Most computational studies to date have identified $Ca^{2+}$-mediated interactions as being solely responsible for bridging anionic head groups in neighboring LPS molecules (Brandner et al, 2024; Dias et al, 2014; Kirschner et al, 2012; Santos et al, 2017; Vaiwala and Ayappa, 2024; Wu et al, 2013; Wu et al, 2014). However, in contrast to these studies, we show that $Mg^{2+}$-mediated interactions exert a greater restrictive influence on LPS diffusion compared to $Ca^{2+}$. Notably, the importance of $Mg^{2+}$ ions in defining the lateral packing and permeability of model membranes incorporating LPS has been observed previously (Garidel et al, 2005; Snyder et al, 1999). Furthermore, divalent cation depletion in the OM generates a sub-population of LPS molecules undergoing Brownian free diffusion, which is not normally present, while OMPs only experience a moderate increase in the lateral diffusion coefficient and confinement diameter determined using SPT. This suggests that the LPS–OMP interactions that exist in OMP islands are more resilient to chemical challenges compared to the LPS–LPS interactions that dominate in the LPS-rich regions.

The chemical structure and conformation of LPS facilitate the formation of multiple types of intermolecular LPS–LPS interactions, contributing to the tight packing of LPS molecules in the outer leaflet of the OM required for its aforementioned barrier and load-bearing functions (Delcour, 2009; Rojas et al, 2018; Silhavy et al, 2010). It appears, based on this experimental work, that the optimal packing of acyl chains and the binding of $Mg^{2+}$ are both critical to maintaining native van der Waals contacts and electrostatic interactions between the Lipid A moieties of

neighboring LPS molecules. These observations also align with our understanding of how and why bacteria make chemical modifications to the Lipid A anchor of LPS (e.g., alter acyl chain characteristics or phosphate modification with positively charged constituents) that is already at the cell surface in order to alter the properties of the OM (Simpson and Trent, 2019). The extent to which lateral diffusion in the OM is altered in response to external challenges or changes in membrane composition appears dictated by the different biophysical properties of LPS- and OMP-rich regions (Sun et al, 2022). We have demonstrated that global increases in LPS mobility occur within LPS-rich patches when OM asymmetry and stability are perturbed. The new insight gained by revealing the molecular basis of LPS confinement in the OM can assist in the design and development of novel OM-disrupting antimicrobial therapies and will improve our ability to build more accurate in silico models of the Gram-negative cell envelope.

## Methods

### Reagents and tools table

| Reagent/resource | Reference or source | Identifier or catalog number |
|---|---|---|
| **Experimental models** | | |
| *Escherichia coli* K-12 MG1655 | Blattner et al (1997) | MG1655 |
| *Escherichia coli* K-12 BW25113 | Grenier et al (2014) | BW25113 |
| *Escherichia coli* K-12 Δ*waaC* | Baba et al (2006) | JW3596-KC |
| *Escherichia coli* K-12 Δ*lpp* | Baba et al (2006) | JW1667-KC |
| *Escherichia coli* K-12 Δ*ompA* | Baba et al (2006) | JW0940-KC |
| *Escherichia coli* K-12 BW25113 imp4213 | Gift from Emmanuele Severi | *imp4213* |
| *Escherichia coli* K-12 DFB1655 | Browning et al (2013) | O-antigen restored strain (OAR) |
| *Escherichia coli* O18:K1:H7 UTI89 | Chen et al (2006) | UTI89 |
| ClearColi™ K-12 MG1655 endotoxin-free | Research Corporation Technologies | ClearColi™ |
| *Pseudomonas aeruginosa* PAO1 | Stover et al (2000) | PAO1 |
| *Salmonella enterica enterica* serovar Typhimurium strain LT2 | McClelland et al (2001) | *Salmonella* Typhimurium LT2 |
| **Recombinant DNA** | | |
| pBADcLIC | Geertsma and Poolman (2007) | |
| pBAD-lptD* | This study | |
| pBAD-ompA* | This study | |
| pEVOL-pylRS | Gift from Edward Lemke | Plasmid encoding pyrrolysyl-tRNA synthetase and pyrrolysyl-tRNA from *Methanosarcina mazei* |
| pNP4 | Gerding et al (2007) | Plasmid encoding recombinant GFP-TolA |

| Reagent/resource | Reference or source | Identifier or catalog number |
|---|---|---|
| **Oligonucleotides (5′ to 3′)** | | |
| TTTGGGCTAACAGG AGGAATTACATATGA AAAAACGTATCCCCACTCTCC | This study | lptD^WT-for |
| GAGATGAGTTTTTGTTCTA GAAAGCTTACAAAGTGT TTTGATACGGCAGA | This study | lptD^WT-rev |
| TCTGCCGTATCAAA ACACTTTGTAAGCTT TCTAGAACAAAAACTCATCTC | This study | pBADcLIC-for |
| GGAGAGTGGGGATA CGTTTTTTCATATGTAATT CCTCCTGTTAGCCCAAA | This study | pBADcLIC-rev |
| ATGCCATAGCATTTTTATCC | This study | pBADcLIC_forS |
| GATTTAATCTGTATCAGG | This study | pBADcLIC-revS |
| TACTTTGAGTTCTACCTGCC | This study | lptDmid_forS |
| ATGCTGGAGTTACTGGTCGC | This study | lptDmid_revS |
| CGATGACAACATAAC ATGGGAGAATTAGGA CAAAACGGGTTCACTGGT | This study | lptD^[D600]-forM |
| ACCAGTGAACCCGT TTTGTCCTAATTCTC CCATGTTATGTTGTCATCG | This study | lptD^[D600]-revM |
| GCTAACAGGAGGAA TTAACCATGGATGAA AAAGACAGCTATCGCGATTG | This study | ompA^WT-for |
| GATGAGTTTTTGTTCT AGAAAGCTTCGTTAAGC CTGCGGCTGAGTTAC | This study | ompA^WT-rev |
| CGTTGTAACTCAG CCGCAGGCTTAACGA AGCTTTCTAGAACA AAAACTCATCTCAG | This study | pBAD-for-2 |
| CAATCGCGATAGCTGTCT TTTTCATCCATGGTTA ATTCCTCCTGTTAGCC | This study | pBAD-rev-2 |
| CGTATGCCGTACAAA GGCAGCGTTTAGAAC GGTGCATACAAAGCTCAGGGC | This study | ompA^[E89]-for |
| GCCCTGAGCTTTGTATGCA CCGTTCTAAACGCTGCC TTTGTACGGCATACG | This study | ompA^[E89]-rev |
| **Chemicals, enzymes, and other reagents** | | |
| Ampicillin | Melford | A40040 |
| Chloramphenicol | Sigma | C-0378 |
| Kanamycin | Sigma-Aldrich | K4000 |
| Tris(2-carboxyethyl)phosphine (TCEP) | Thermo Fisher Scientific | 20491 |
| 8-Azido-3,8-dideoxy-D-manno-octulosonic acid (Kdo-N_3) | Vector Laboratories, Inc. | CCT-1241 |
| N-acetylneuraminic acid (Neu5Ac) | Fluorochem | F047790 |
| 3-azido-D-alanine (D-Ala-azide) | Jena Bioscience GmbH | CLK-AA004 |
| Propargyl-D-Ala-D-Ala (prop-D-Ala-D-Ala) | Gift from Darshita Budhadev and Martin Fascione | |
| ε-(tert-butoxycarbonyl)-L-lysine (L-Lys(Boc)-OH) | Thermo Scientific Chemicals | 11440347 |

| Reagent/resource | Reference or source | Identifier or catalog number |
|---|---|---|
| N-propargyl-L-lysine | Sirius Fine Chemicals SiChem GmbH | SC-8002 |
| β-mercaptoethanol (β-ME) | Sigma | M7522 |
| Anhydrous dimethylsulfoxide (DMSO) | Sigma-Aldrich | 276855 |
| N,N-dimethyldodecyl-amine N-oxide (LDAO) | Sigma | D9775 |
| CyGEL™ | BioStatus | CY10500 |
| 40× Phosphate Buffered Saline pH 7.4 | BioStatus | PB40125 |
| 5 μm diameter silica beads (10% w/v) | Bang Laboratories | SS05N |
| Poly-D-lysine (MW 30-70 kDa) | Sigma | P7886 |
| Paraformaldehyde (ultrapure, methanol-free) | Polysciences Ltd. | 18814-20 |
| Glucose oxidase (GluOx) | Sigma-Aldrich | G-6641 |
| Catalase (Cat) | Sigma-Aldrich | C100 |
| β-mercaptoethylamine hydrochloride (β-MEA) | Sigma | 30078 |
| Liquified phenol ( ≥ 89.0%) | Sigma-Aldrich | P9346 |
| Alexa Fluor 488 C_5 maleimide | Thermo Fisher Scientific | A10254 |
| Alexa Fluor 488-alkyne | Thermo Fisher Scientific | A10267 |
| AZDye 568-alkyne | Vector Laboratories, Inc. | CCT-1293 |
| AZDye 594-alkyne | Vector Laboratories, Inc. | CCT-1297 |
| AZDye 647-alkyne | Vector Laboratories, Inc. | CCT-1301 |
| Alexa Fluor 488-azide | ThermoFisher Scientific | A10266 |
| AZDye 594-azide | Vector Laboratories, Inc. | CCT-1295 |
| AZDye 647-azide | Vector Laboratories, Inc. | CCT-1299 |
| AZDye 488-dibenzocyclooctyne (AZDye 488-DBCO) | Vector Laboratories, Inc. | CCT-1278 |
| AZDye 568-dibenzocyclooctyne (AZDye 568-DBCO) | Vector Laboratories, Inc. | CCT-1294 |
| TetraSpeck 0.2 μm fluorescent microsphere | Thermo Fisher Scientific | T7280 |
| **Software** | | |
| AlphaFold2 | Jumper et al (2021) | |
| MATLAB R2024b | MathWorks | |
| OriginPro 2024b | OriginLab | |
| Python 3 | Python Software Foundation | |
| ZEN Black 3.0 SR FP2 | Carl Zeiss AG | |
| FIJI | Schindelin et al (2012) | |

| Reagent/resource | Reference or source | Identifier or catalog number |
|---|---|---|
| ImageJ | Schneider et al (2012) | version 1.51j |
| PaTrack | Dosset et al (2016) | |
| TrackMate | Tinevez et al (2017) | |
| **Other** | | |
| NEBuilder® HiFi DNA Assembly master mix | New England Biolabs | E2621S |
| QIAGEN Plasmid Midi kit | Qiagen | 12143 |
| Click-iT™ Cell Reaction Buffer kit | Thermo Fisher Scientific | C10269 |
| Pro-Q Emerald 300 Lipopolysaccharide Gel Stain kit | Thermo Fisher Scientific | P30635 |

## Bacterial strains and plasmids

Details of strains used in this study are provided in Appendix Table S4. *Escherichia coli* K-12 BW25113 and MG1655 strains were both used in this study (Blattner et al, 1997; Grenier et al, 2014). Keio collection mutants of the *E. coli* K-12 BW25113 strain used in this work (Δ*waaC*, Δ*lpp*, Δ*ompA*) were validated previously via PCR, confirming the presence of a kanamycin resistance cassette within the open-reading frame of interest (Baba et al, 2006; Rassam et al, 2015). Plasmid pNP4 encoding GFP-TolA was kindly provided by Piet de Boer (Gerding et al, 2007) and transformed into the *E. coli* K-12 BW25113 strain. The solitary in-frame deletion in the *lptD* gene was introduced into the *E. coli* K-12 BW25113 strain using P1 transduction by Dr. Emmanuele Severi (Northumbria University) using a previously published donor MC4100 strain with the *imp4213* allele (Ruiz et al, 2005). Introduction of this allele was verified by Sanger sequencing (Eurofins Genomics LLC) after its amplification by PCR (using wild-type *lptD* amplification primers, Appendix Table S5). *Pseudomonas aeruginosa* PAO1 is a model pathogenic strain (Stover et al, 2000) and was obtained from Prof. Gavin Thomas (University of York). *Salmonella enterica enterica* serovar Typhimurium strain LT2 (McClelland et al, 2001) (abbreviated to *S.* Typhimurium LT2 in this work) and *E. coli* UTI89 (Chen et al, 2006) strains were provided by Prof. Marjan van der Woude (University of York). The endotoxin-free ClearColi™ K-12 strain was commercially available (Mamat et al, 2015). The O-antigen restored (OAR) *E. coli* DFB1655 strain was obtained from Prof. Ian Henderson (The University of Queensland) and Dr. Douglas Browning (Aston University) (Browning et al, 2013).

The pBAD-lptD* plasmid was produced via Gibson isothermal assembly (Gibson et al, 2009). PCR was used to amplify the *lptD* gene insert (see Appendix Table S5 for primers) from purified *E. coli* K-12 subs. MG1655 genomic DNA. The amplified gene was cloned into a pBADcLIC vector (Geertsma and Poolman, 2007) using a commercial Gibson isothermal assembly kit (NEBuilder® HiFi DNA Assembly). PCR site-directed mutagenesis (see Appendix Table S5 for primers) was then used to incorporate the amber stop codon TAG in place of the codon for the D600 residue (Appendix Fig. S4A). Successful cloning and mutagenesis were verified via DNA sequencing (Eurofins Genomics LLC). The

pEVOL-pylRS plasmid, encoding the pyrrolysyl-tRNA synthetase/pyrrolysyl-tRNA pair from *Methanosarcina mazei* was obtained from Dr. Edward Lemke (EMBL Heidelberg) under MTA (Brabham et al, 2020; Plass et al, 2011). pBAD-lptD* and pEVOL-pylRS were co-transformed into *imp4213* via electroporation (MicroPulser, BioRad Laboratories) (1.8 kV, 25 μF, time constant: 3.5 ms). *Imp4213* pBAD-lptD$^{D600}$ pEVOL-pylRS were selected by incubating co-transformed cells overnight at 37 °C on an Lysogeny Broth (Miller formulation, LB-Miller) agar plate containing 50 μg mL$^{-1}$ ampicillin and 17.5 μg mL$^{-1}$ chloramphenicol. Unless otherwise specified, the incubation of bacterial cultures was done at 37 °C with 220 rpm agitation in a 50 mL screw-cap polypropylene tube. Liquid pre-cultures were prepared by inoculating 5 mL supplemented M9 chemically defined medium (M9 CDM, Appendix Table S6) with appropriate antibiotics (Appendix Table S4) using a well-isolated single colony of the desired strain picked from freshly streaked LB-Miller agar plate. Post-inoculation cultures were incubated for 6–8 h or until cell densities reached a minimum optical density at 600 nm (OD$_{600}$) of 1.5.

The pBAD-ompA* plasmid was produced using an analogous method to that employed in the production of pBAD-lptD amber stop codon-containing plasmids. Briefly, the *ompA* gene from *E. coli* K-12 BW25113 was amplified via PCR from purified bacterial chromosomal DNA and integrated into the pBADcLIC vector via NEBuilder® HiFi DNA Assembly (see Appendix Table S5 for primers). PCR-based site-directed mutagenesis was then used to mutate the codon for E89 to an amber stop codon (TAG) in the pBAD-ompA plasmid. Following plasmid purification (Qiagen© Midi plasmid purification kit) and DNA sequencing (Eurofins Genomics LLC), the TAG-containing pBAD-ompA vector (pBAD-ompA*) and pEVOL-pyIRS were co-transformed into *E. coli* Δ*ompA* cells via electroporation as above. Co-transformed cells were selected using LB-Miller agar plates with 30 μg mL$^{-1}$ kanamycin, 100 μg mL$^{-1}$ ampicillin, and 35 μg mL$^{-1}$ chloramphenicol. A single colony from this plate was used to inoculate a volume of supplemented M9 CDM with kanamycin, ampicillin, and chloramphenicol antibiotics. Following prolonged pre-culture incubation, the appropriate volume of pre-culture was used to inoculate a fresh batch of supplemented M9 CDM with ampicillin and chloramphenicol antibiotics to an OD$_{600}$ of 0.05. The cultures were then incubated for 2–3 h.

## Colicin Ia purification and fluorophore-labeling

The colicin Ia probe was engineered with an inactivating disulfide (*top-lock*) in the coiled-coil receptor-binding domain (L257C, A411C), allowing high-affinity binding to the CirA receptor at the surface of *E. coli* cells but preventing translocation through the cell envelope and bacterial intoxication. The colicin Ia probe also had a solvent-exposed cysteine in the cytotoxic domain (Cys544) for maleimide-directed fluorescent labeling. The colicin Ia-based probe used in this work was purified and fluorescently labeled with the Alexa Fluor 488-maleimide (Invitrogen) as described previously (Rassam et al, 2015). To increase the labeling efficiency, tris(2-carboxyethyl)phosphine (TCEP, Thermo Scientific) was added to the purified colicin Ia to give a final concentration of 0.2 mM, and it was incubated at room temperature for 30 min before adding the Alexa Fluor 488-maleimide to initiate the labeling reaction. The labeling efficiency (~ 1 fluorophore per protein) was confirmed by

spectrophotometry (colicin Ia: $\varepsilon_{280\ nm} = 59{,}360\ M^{-1}\ cm^{-1}$; AF488: $\varepsilon_{495\ nm} = 71{,}000\ M^{-1}\ cm^{-1}$) after correcting for the absorption at 280 nm by the AF488 dye ($A_{280\ nm} = 0.11 \times A_{495\ nm}$). The photobleaching of top-locked AF488-labeled colicin Ia immobilized onto a quartz slide surface for TIRFM yielded a single-step drop in fluorescence intensity consistent with fluorescent labeling at a single position.

## Production of 8-azido-3,8-dideoxy-D-manno-octulosonic acid (Kdo-N$_3$)

Kdo-N$_3$ was synthesized and purified following established protocols (Dumont et al, 2012). The synthesized Kdo-N$_3$ was utilized for all ensemble and single-molecule fluorescence microscopy experiments and yielded the results as commercially available Kdo-N$_3$ (Vector Laboratories, Inc.) used in more recent work. Freeze-dried stocks of the synthesized compound and the purchased compound were resuspended to a concentration of 400 mM in sterile, deionized water and stored at −20 °C until required.

## LPS metabolic labeling with Kdo-N$_3$

Pre-cultures were used to inoculate fresh medium charged with 4 mM Kdo-N$_3$ to a starting theoretical OD$_{600}$ of 0.05. Post-inoculation metabolic labeling cultures were incubated for 14–16 h. Cultures were subsequently harvested upon reaching late stationary phase. Cells were isolated via centrifugation (8000×$g$, 3 min, 4 °C) and washed three times with fresh volumes of supplemented M9 CDM containing 5 mM arabinose to remove residual Kdo-N$_3$ prior to LPS fluorescent labeling.

## In situ fluorescent labeling of Kdo-N$_3$-containing LPS inner core domains via Cu(I) catalyzed azide-alkyne cycloaddition (CuAAC) in bacteria with non-capsular serotypes

Cells presenting Kdo-N$_3$-containing LPS were pelleted via centrifugation and resuspended to an OD$_{600}$ of 1.0 in "Click-iT" reaction mix prepared according to the manufacturer's protocol ("Click-iT" Cell Reaction Buffer Kit, Molecular Probes®, Invitrogen) supplemented with 4 mM $N$-acetylneuraminic acid (Neu5Ac) and charged with the alkyne-functionalized fluorescent dye (concentrations specified in Appendix Table S7). The suspension was transferred to a sterile 2-mL microcentrifuge tube and incubated on a rotary wheel (12 rpm, 80° incline relative to the bench top) for 30 min at room temperature. Post-labeling, the suspension was transferred to a fresh, sterile 2 mL microcentrifuge tube, and cells bearing fluorescent LPS were pelleted via centrifugation. The labeled cell pellet was then washed a further three times with supplemented M9 media to remove residual fluorophore and "Click-iT" mix components.

## In vivo fluorescent labeling of Kdo-N$_3$-containing LPS inner core domains via Cu(I) catalyzed azide-alkyne cycloaddition (CuAAC) in bacteria with capsular serotypes (E. coli UTI89 and P. aeruginosa PAO1)

After the three pre-labeling wash steps to remove residual Kdo-N$_3$, two additional wash steps were done with supplemented M9 CDM containing 100 mM EDTA to remove extracellular capsular matrices and LPS for fluorescent labeling purposes. To restore OM-coordinated divalent cations (Mg$^{2+}$ and Ca$^{2+}$) and remove residual EDTA prior to fluorescent labeling, the capsule-stripped cell pellets were then washed twice with supplemented M9 CDM. CuAAC fluorescent labeling of exposed, Kdo-N$_3$-containing LPS was done as detailed above.

## Live-cell Kdo-N$_3$-containing LPS labeling of E. coli BW25113 or MG1655 and BW25113 ΔwaaC via Cu(I) free strain promoted azide-alkyne cycloaddition (SPAAC)

Washed Kdo-N$_3$-labeled LPS liquid culture cell samples were diluted in fresh, supplemented M9 CDM to an OD$_{600}$ of 1.0 and transferred to a sterile 2-mL microcentrifuge tube. The appropriate volume of a dibenzocyclooctyne (DBCO)-functionalized fluorophore was then added directly to this suspension (Appendix Table S8). Labeling suspensions were then incubated for 1 h at 30 °C on a rotary wheel (12 rpm, 80° incline relative to the benchtop). Suspensions were then transferred to a fresh, sterile 2 mL microcentrifuge tube and pelleted via centrifugation. The labeled cell pellet was then washed three times in freshly supplemented M9 CDM to remove residual dye. Extra care was taken during gentle resuspension of cell pellets at each stage of this protocol to ensure cells were exposed to minimal levels of shear stress, thereby ensuring minimal loss of cell viability during this labeling protocol.

## In vivo peptidoglycan pentapeptide cross-link metabolic labeling using D-Ala-azide (D-Ala-N$_3$) amino acid or propargyl-D-Ala-D-Ala (prop-D-Ala-D-Ala) dipeptide followed by fluorescent labeling via CuAAC

M9 CDM was inoculated to an OD$_{600}$ of 0.05 via the addition of an appropriate volume of MG1655 starter culture to an OD$_{600}$ of 0.05 and charged with 4 mM D-Ala-azide or prop-D-Ala-D-Ala. Cultures were then incubated for 10–12 h and harvested via centrifugation upon reaching the late stationary phase. The cell pellet was then washed three times to remove residual functionalized D-Ala analog and fluorescently labeled via CuAAC via resuspension in "Click-iT" mix charged appropriately functionalized AZDye 594-dye (Vector Laboratories, Inc.), selected for its ability to penetrate the outer membranes of metabolically labeled cells. "Click-iT" mixes were incubated for 30 min on a rotary wheel (12 rpm, 80° incline relative to the bench top) after which labeled cells were pelleted via centrifugation. Labeled cell pellets were washed three times with supplemented M9 CDM to remove residual dye and "Click-iT" mix components.

## Noncanonical amino acid labeling of LptD

Sites for non-canonical amino acid (ncAA) incorporation into LptD in E. coli BW25113 were selected based on in silico modeling of E. coli LptD crystal structure [PDB: 4RHB] deposited in the PDB database (Appendix Fig. S10). Four sites for ncAA incorporation were selected based on criteria designed to minimize the impact of ncAA incorporation on LptD function and structure, whilst maximizing the probability of ncAA incorporation at an accessible site to enable efficient downstream bio-orthogonal fluorophore coupling and

minimize dynamic quenching of the coupled extrinsic probe by certain amino acids (e.g., tryptophan, histidine, methionine, and tyrosine) (Chen et al, 2010). We selected two lysine (K473 and K602) and two aspartic acid (D592 and D600) residues located on two unstructured, exposed extracellular loops of LptD. These selections were informed by results from published structural, computational, and biochemical studies of LptD (Dong et al, 2014; Freinkman et al, 2011). By disregarding the extracellular loops, residues, and regions implicated in LptD function by these previous studies, we were able to minimize the probability that ncAA incorporation would have a deleterious impact on LptD structure and function.

Introduction of the L-lysine ncAA to the culture medium resulted in leaky expression of the recombinant, amber-stop codon-containing lptD gene (lptD*), pylRS and tRNA$^{Pyl}$ and subsequent production of recombinant, full-length LptD with the ncAA integrated in one of four positions in its extracellular surface loops (LptD*). The accessible alkyne groups in the surface loops of LptD* enabled conjugation of an azide-functionalized, extrinsic fluorescent organic dye molecule via CuAAC (Huisgen, 1963; Wang et al, 2003). LptD* fluorescent labeling enabled in vitro characterization of the recombinant protein by fluorescence imaging of LptD bands during SDS-PAGE screens. We applied the same methodology to label LptD with equivalent efficiency with ε-(tert-Butoxycarbonyl)-L-lysine (L-Lys(Boc)-OH) ncAA, (Appendix Fig. S4C) (Sharma et al, 2018; Wan et al, 2010) when carrying out experiments that did not require LptD* fluorescent labeling. LptD is one of only two essential OMPs necessary for E. coli growth under standard culture conditions (Okuda et al, 2016; Wu et al, 2006). It was therefore impossible to produce the recombinant ncAA-containing LptD protein variants in a ΔlptD background. However, we were able to express recombinant ncAA-containing lptD variants in E. coli BW25113 imp4213 (Sampson et al, 1989). This mutant lptD strain produces a defective LptD protein (Braun and Silhavy, 2002). The resulting phenotype (e.g., increased LPS lateral mobility, increased sensitivity to antibiotics, detergents, and increased OM permeability) stems directly from the impaired function of the mutant LptD protein (Ruiz et al, 2005; Wu et al, 2005). We were able to demonstrate the functionality of the ncAA-containing LptD variants by rescuing a wild-type phenotype in the imp4213 strain (Fig. 3D; Appendix Fig. S4C), by replicating the results of previous studies in which an SDS-PAGE screen was used to show complexing of LptD with LptE (vital for LptD function and LPS insertion) (Appendix Fig. S4B) (Chng et al, 2010; Freinkman et al, 2011). Direct LptD* fluorophore conjugation via CuAAC enabled us to carry out these experiments without the use of immunoblotting for LptD and LptDE visualization. Validation of ncAA-LptD production and subsequent fluorescent labeling was done for all four ncAA-containing lptD variants, along with SDS-PAGE screens of recombinant protein production levels and ability to form a complex with LptE. Based on this screening process, the LptD$^{D600}$ mutant was selected and used to rescue a wild-type phenotype in the imp4213 strain (Fig. 3D; Appendix Fig. S4C).

## Rescuing wild-type phenotype in the LptD-defective E. coli BW25113 imp4213 strain by production of functional ncAA-containing LptD*

E. coli K-12 BW25113 imp4213 cells were co-transformed with pBAD-lptD* and pEVOL-pyIRS by electroporation, and selection on LB-Miller agar plates was done using reduced concentrations of ampicillin and chloramphenicol (50 μg mL$^{-1}$ and 17.5 μg mL$^{-1}$, respectively). Reduced antibiotic concentrations were required for selection of co-transformed cells because imp4213 cells exhibit elevated sensitivity to antibiotics (Eggert et al, 2001; Sampson et al, 1989). A single colony of co-transformed imp4213/pBAD-lptD*/pEVOL-pyIRS was used to inoculate a volume of LB-Miller with 50 μg mL$^{-1}$ ampicillin and 17.5 μg mL$^{-1}$ chloramphenicol. Post-inoculation, this culture was incubated for 6 h. The pre-culture was used to inoculate two fresh volumes of LB-Miller with ampicillin and chloramphenicol antibiotics to a starting OD$_{600}$ of 0.01. Post-inoculation cultures were incubated for ~3 h to an OD$_{600}$ of 0.5. One of the cultures was charged with ε-(tert-Butoxycarbonyl)-L-lysine (L-Lys(Boc)-OH) (Fisher Scientific, Appendix Fig. S4) at a final concentration of 1 mM, and both cultures were incubated for a further 20 min. In total, 100 μL volumes of the L-Lys(Boc)-OH supplemented culture were spread across a MacConkey agar plate and a LB-Miller agar plate with 0.5% (w/v) SDS and 1 mM EDTA, both supplemented with 1 mM L-Lys(Boc)-OH. Similarly, 100 μL volumes of the unsupplemented culture were spread across identical plates that had not been supplemented with L-Lys(Boc)-OH. After incubating the plates at room temperature for 5 min to allow the liquid culture to absorb into the agar, they were incubated overnight at 37 °C. Growth on L-Lys(Boc)-OH supplemented and unsupplemented plates was then compared.

## OmpA fluorescent labeling via ncAA incorporation followed by CuAAC

To investigate the distributions and spatial organizations of LPS and OMPs relative to one another in the outer leaflet of the OM, OmpA was selected for labeling via adaptation of the strategy used for LptD. OmpA is one of the most abundant OMPs in E. coli with ~100,000 molecules per cell (Koebnik, 1999; Koebnik et al, 2000), and lateral confinement of OmpA in the OM does not depend on its peptidoglycan-binding domain (Verhoeven et al, 2013). A three-dimensional model of the E. coli OmpA structure was produced using AlphaFold2 (Jumper et al, 2021). We then identified regions of the OmpA primary sequence predicted to form unstructured extracellular loops. We selected an accessible glutamic acid (E89) residue in extracellular loop 2 for N-propargyl-L-lysine incorporation via mutation of the respective codon in the ompA gene sequence to amber stop codons (TAG) (Appendix Fig. S2A) (Hancock et al, 2010; Milles et al, 2012; Nguyen et al, 2009).

After isolation and amplification of the E. coli BW25113 ompA gene sequence from the purified bacterial chromosome via PCR (Appendix Table S5), we incorporated the wild-type ompA gene into a pBADcLIC expression vector. The target codons in the ompA gene were mutated to TAG via PCR mutagenesis (Appendix Table S5). Under standard laboratory culture conditions, OmpA is a non-essential OMP. Therefore, we were able to co-transform TAG-containing ompA expression vectors into an E. coli ΔompA strain (Appendix Table S4) along with the aaRS/suppressor tRNA vector pEVOL-pyIRS (Baba et al, 2006). Full-length ncAA-containing OmpA (OmpA*) production was initiated by charging cell cultures with 1 mM N-propargyl-L-lysine (Sirius Fine Chemicals, SiChem GmbH). OmpA* fluorescent labeling with a functionalized small extrinsic organic dye molecule was done in situ via CuAAC.

## Rescuing wild-type phenotype in the *E. coli* BW25113 Δ*ompA* strain by production of functional ncAA-containing OmpA*

*E. coli* K-12 BW25113 Δ*ompA* cells were co-transformed with pBAD-ompA* and pEVOL-pyIRS by electroporation, and selection on LB-Miller agar plates was done using ampicillin and chloramphenicol (100 $\mu$g mL$^{-1}$ and 35 $\mu$g mL$^{-1}$, respectively). A single colony of co-transformed Δ*ompA*/pBAD-ompA*/pEVOL-pyIRS was used to inoculate a volume of LB-Miller with 100 $\mu$g mL$^{-1}$ ampicillin and 35 $\mu$g mL$^{-1}$ chloramphenicol. Post-inoculation, this culture was incubated for 6 h at 37 °C with 220 rpm agitation. The pre-culture was used to inoculate two fresh volumes of LB-Miller with ampicillin and chloramphenicol antibiotics to a starting OD$_{600}$ of 0.01. Post-inoculation cultures were incubated for ~3 h to an OD$_{600}$ of 0.5. One of the cultures was charged with *N*-propargyl-L-lysine at a final concentration of 1 mM, and both cultures were incubated for a further 20 min as above. In all, 5 $\mu$L dilutions (1:1 to 1:1000) of the ncAA-charged culture were streaked across LB-Miller agar plates containing 0.5% (w/v) SDS and 1 mM EDTA, with 1 mM *N*-propargyl-L-lysine added to the agar plates to ensure continued OmpA* production. Similarly, 5 $\mu$L dilutions (1:1 to 1:1000) of the culture without the ncAA added were spread across identical plates that had not been supplemented with *N*-propargyl-L-lysine. After incubating the plates at room temperature for 5 min to allow the liquid culture to absorb into the agar, they were incubated overnight at 37 °C. Growth on *N*-propargyl-L-lysine-supplemented and unsupplemented plates was then compared.

## Dual differential fluorescent labeling of OmpA* and newly inserted Kdo-N₃-containing LPS

*E. coli* K-12 BW25113 Δ*ompA* cells were co-transformed with pBAD-OmpA* and pEVOL-pyIRS via electroporation as detailed above. A 5 mL volume of supplemented M9 CDM with 100 $\mu$g mL$^{-1}$ ampicillin, 30 $\mu$g mL$^{-1}$ kanamycin and 35 $\mu$g mL$^{-1}$ chloramphenicol was inoculated with a single colony of freshly co-transformed cells and incubated at 37 °C with 220 rpm agitation for ~7 h (to an OD$_{600}$ of 1.5–2.0). This pre-culture was used to inoculate a fresh volume of supplemented M9 CDM plus selection antibiotics to a starting OD$_{600}$ of 0.05. The culture was incubated as before for approximately 3 h, to an OD$_{600}$ of ~0.5. Uninduced recombinant OmpA* production and LPS Kdo-N₃ metabolic labeling were done simultaneously via the addition of *N*-propargyl-L-lysine and Kdo-N₃ to final concentrations of 1.0 mM and 4.0 mM, respectively. The recombinant OmpA* production/LPS metabolic labeling culture was then re-incubated under standard conditions for a further 1 h. After 1 h, the culture was removed and placed immediately on ice to halt further cell division, LPS insertion, and OmpA* production. The culture was diluted with pre-chilled, supplemented M9 CDM plus selection antibiotics to an OD$_{600}$ of 1.0 and transferred to a pre-chilled 2.0 mL microcentrifuge tube.

Cells were then harvested via centrifugation (10,000×*g*, 3 min, 4 °C) and the supernatant was carefully removed. The cell pellet was then resuspended to an OD$_{600}$ of 1.0 in fresh, pre-chilled supplemented M9 CDM plus selection antibiotics via repeated, gentle pipetting. The resuspended culture was transferred to a fresh-pre-chilled 2.0 mL microcentrifuge tube on ice. This washing step was repeated three times in total to ensure the removal of

residual *N*-propargyl-L-lysine and Kdo-N₃. The washed cell pellet was then resuspended to an OD$_{600}$ of 1.0 in "Click-iT" reaction mix charged with AF488- or AZ647-alkyne, transferred to a 2-mL microcentrifuge tube, and incubated at room temperature for 30 min on a rotary wheel (12 rpm, 80° incline relative to benchtop). In replicate experiments, the LPS and OmpA$^{E89}$ labeling order was alternated, as were the dye combinations (Appendix Table S9). This alternative labeling strategy was implemented in the experimental replicates to ensure that the order in which species were labeled and the dye combinations used did not influence results (Fig. 2A; Appendix Fig. S2). Cells were then pelleted via centrifugation (10,000×*g*, 3 min, 4 °C). The spent 'Click-iT' reaction mix supernatant was carefully removed. The cell pellet was then resuspended to an OD$_{600}$ of 1.0 in 0.2-$\mu$m-filtered, pre-chilled phosphate-buffered saline pH 7.4 (PBS; 137 mM NaCl, 2.7 mM KCl, 4.3 mM Na₂HPO₄, Sigma-Aldrich) via gentle repeated pipetting.

The suspension was transferred to a fresh 2.0 mL microcentrifuge tube and cells were re-pelleted via centrifugation (10,000×*g*, 3 min, 4 °C). The PBS supernatant was carefully removed, and the pellet resuspended to an OD$_{600}$ of 1.0 in "Click-iT" reaction mix charged with AZ647- or AF488-N₃. The suspension was transferred to a 2.0-mL microcentrifuge tube and incubated on a rotary wheel (12 rpm, 80° incline relative to the bench top) at room temperature for 30 min. Cells were then pelleted via centrifugation (10,000×*g*, 3 min, 4 °C). The spent "Click-iT" reaction mix was carefully removed, and the pellet was resuspended in pre-chilled, filtered PBS to an OD$_{600}$ of 1.0. The suspension was transferred to a fresh 1.5-mL pre-chilled microcentrifuge tube and cells re-pelleted via centrifugation (10,000×*g*, 3 min, 4 °C). This washing step was repeated three times in total. Cells were then fixed via resuspension to an OD$_{600}$ of 1.0 in 4% (v/v) paraformaldehyde in PBS followed by incubation at room temperature for 30 min on a rotary wheel (12 rpm, 80° incline relative to the benchtop). The suspension was then transferred to a fresh, pre-chilled 1.5-mL microcentrifuge tube, and cells were pelleted via centrifugation (10,000×*g*, 3 min, 4 °C). The pellet consisting of dual-labeled, fixed cells was washed three more times in pre-chilled, 0.2-$\mu$m filtered PBS as before. After the final wash step, cell pellets were resuspended to an OD$_{600}$ of 1.0 in PBS and stored at 4 °C or on ice, protected from light until imaging was done.

### LPS extraction

LPS was extracted via adaptation of the hot phenol—aqueous method developed by Westphal and Jann (1965). The cell pellet was resuspended to a theoretical OD$_{600}$ of 0.5 in PBS plus 0.5 mM MgCl₂ and 0.15 mM CaCl₂, pH 7.4, and pelleted via centrifugation (10,000×*g*, 10 min, 4 °C). The washed cell pellets were then resuspended to the same OD$_{600}$ in ultrapure deionized water (resistivity = 18.2 MΩ cm) and transferred to a glass dram with a Teflon-coated lid. An equal volume of 90% (w/v) phenol, pre-warmed to 65 °C was added and the emulsion stirred vigorously at 65 °C for 15 min. Drams were then transferred to ice for 15 min to cool. Emulsions were then transferred to microcentrifuge tubes and centrifuged (8500×*g*, 10 min, 15 °C). The LPS-containing aqueous fraction was transferred to a 50-mL screw-cap polypropylene tube. A second aqueous extraction was then done on the remaining phenol layer, and the aqueous fractions were pooled in the 50 mL screw-cap polypropylene tube. Sodium acetate was added to the pooled aqueous fractions to a final concentration of 0.5 M. Ten volumes of 95% (v/v) ethanol, pre-chilled to −20 °C were then

added, and after mixing via repeated inversions the suspension was incubated overnight to maximize LPS precipitation. The following day LPS was pelleted via centrifugation (2000×*g*, 10 min., 4 °C). After supernatant aspiration, the LPS pellet was dried under a steady stream of $N_{2(g)}$. The dried LPS pellet was resuspended in 100 µL ultrapure deionized water and transferred to a 1.5 mL microcentrifuge tube. Sodium acetate was added to a final concentration of 0.5 M, and 1.1 mL pre-chilled 95% (v/v) ethanol was added. After tube content mixing via repeated inversion LPS was re-precipitated via overnight incubation at −20 °C. Precipitated LPS was pelleted via centrifugation. The pellet was dried again under $N_{2(g)}$ and finally resuspended in 50 µL PBS.

## LPS Tricine-sodium dodecyl sulfate-polyacrylamide gel electrophoresis (TSDS-PAGE)

A volume (15 µL) of the purified LPS was mixed with a volume (5 µL) of 4× loading buffer (Appendix Table S10) plus 2% (v/v) β-mercaptoethanol (β-ME). After vortexing and a brief centrifugation, the samples were incubated for 1 h at 40 °C. The samples were then re-vortexed and centrifuged briefly before loading the entire sample volume into the well of a 10 cm² 1 mm thick 15% (w/v) acrylamide resolving/6% (w/v) acrylamide stacking Tricine gel prepared according to the protocol published by Schägger (2006). Commercial smooth LPS standards (Thermo Scientific®) were analyzed alongside extracted LPS samples to facilitate the classification of sample LPS bands. O55:B5-AZ488 LPS conjugate samples were prepared for TSDS-PAGE by adding 1 µL of 1 mg mL⁻¹ LPS conjugate stock solution to 9 µL ultrapure deionized water followed by 3.3 µL of 4× loading buffer plus 2% (v/v) β-ME (Appendix Table S10). Samples were vortexed and briefly centrifuged before being incubated at 40 °C for 1 h, then re-vortexed and briefly centrifuged again before loading the entire sample volume into the well of a Tricine gel. Electrophoresis was done at 4 °C using pre-chilled buffers with the gel tank protected from light to prevent fluorophore (AF488) photobleaching. Electrophoresis was done in constant current mode for 45 min at 30 mA until a dye front formed as a fine horizontal line at the stacking/resolving gel interface. The current was then increased to 60 mA for ~3 h until the dye front was approximately 1 cm from the base of the gel.

## Processing and visualization of LPS TSDS-PAGE gels

Upon completion of electrophoresis, the gels were fixed via immersion in 150 mL 50% (v/v) methanol/3% (v/v) glacial acetic acid and incubated overnight at room temperature on a slow-moving rocker, protected from light. The following day, gels were washed via immersion in 150 mL 3% (v/v) glacial acetic acid and incubated for 20 min at room temperature on a slow-moving rocker. This step was repeated three times in total. AF488-LPS bands were then visualized using an Amersham Typhoon 5 gel and blot bioimaging system equipped with a 488 nm argon laser and 525 BP filter. To visualize total LPS bands, the gels were then stained using a Pro-Q Emerald 300 LPS gel stain kit according to the manufacturer's instructions (Thermo Fisher Scientific). Briefly, after fixation and washing steps, LPS carbohydrates were oxidized via immersion in 30 mL 1% (w/v) $H_5IO_6$ followed by incubation for 45 min on a slow-moving rocker at room temperature. Gels were then washed three times in 150 mL 3% (v/v) glacial acetic acid for 20 min. Gels were then stained via immersion in 25 mL dilute

(1:50) Pro-Q Emerald 300 stain and incubation for 90 min at room temperature on a slow-moving rocker. After post-staining washing steps, Pro-Q-stained LPS bands were visualized under UV illumination using a GeneGenius gel imaging system.

## Pre-FRAP and pre-TIRFM treatments with chelator, detergent, or urea

After in situ fluorescent labeling of the LPS and the post-labeling washing steps, the cell pellet was resuspended to an $OD_{600}$ of 1.0 in M9 CDM plus chelator, *N*,*N*-dimethyldodecylamine *N*-oxide (LDAO) or urea at pre-determined maximum concentrations cells were able to withstand whilst remaining viable and exhibiting standard growth rates (see Appendix Table S11) and transferred to a sterile 2-mL microcentrifuge tube. The suspension was then incubated on a rotary wheel (12 rpm, 80° incline relative to the benchtop) for 30 min at ambient temperature. Cells presenting deep rough and/or rough LPS (see Appendix Table S4) were mounted and imaged in M9 CDM plus chelator, detergent or urea and 5 µm silica beads at an $OD_{600}$ of 2.0. Cells presenting smooth LPS (see Appendix Table S4) were mounted in PBS-charged CyGEL™ (BioStatus) with 5 µm diameter silica beads added.

## Culturing bacterial cells in M9 chemically defined medium with altered Mg²⁺ and Ca²⁺ concentrations

Standard culturing, metabolic- and fluorescent labeling protocols were followed exchanging standard M9 CDM (see Appendix Table S6) for M9 CDM supplemented with altered Mg²⁺ and Ca²⁺ concentrations (see Appendix Table S12). Preliminary $OD_{600}$ growth curve control experiments done in triplicate showed that exchange of standard M9 CDM for the two modified M9 compositions had no statistically significant impact on rates of cell growth.

## Mounting of *E. coli* strains producing deep rough or rough LPS for fluorescence microscopy using poly-ᴅ-lysine-coated coverslips or quartz slides

Washed and labeled cell pellets were resuspended in freshly supplemented M9 medium (plus antibiotic(s) where required, Appendix Table S4) to an $OD_{600}$ of 2.0. In all, 48 µL of the sample was mixed with 2 µL 0.5% (w/v) 5 µm diameter silica bead slurry (Bang Laboratories, slurry prepared in unsupplemented M9 CDM, final bead concentration = 0.02% (w/v)). In total, 30 µL of this suspension was pipetted dropwise along the long axis centreline of a clean 1.0–1.2 mm glass slide and covered with a 22 mm × 64 mm No. 1.5 coverslip coated with 20 µg mL⁻¹ poly-ᴅ-Lysine (prepared in 50 mM MOPS, pH 8). The coverslip edges were sealed with clear nail varnish to prevent the sample drying out. The slide was then incubated in a light-excluding container for 10 min at room temperature to enable cells to adhere to the coverslip and allow the clear nail varnish to set.

## Mounting of *E. coli* strains producing smooth LPS for fluorescence microscopy using thermoreversible CyGEL

Chilled (on ice to ~4 °C), washed, and labeled smooth LPS cell pellets were resuspended to an $OD_{600}$ of 2.0 in pre-chilled CyGEL

charged with 1× PBS pH 7.4 and 5-µm silica beads on ice via gentle, repeated pipetting. In all, 30 µL of the suspension was then aliquoted along the long axis centreline of a pre-chilled 1.0–1.2 mm glass slide and covered with a pre-chilled clean no. 1.5 glass coverslip. The slide was sealed via the application of a clear nail varnish bead along the edges of the coverslip and placed in a pre-chilled, light-excluding container and incubated at 4 °C for 10 min allowing even suspension dispersal within the sealed cavity between the slide and coverslip. The container was then incubated at room temperature for a further 15 min resulting in the CyGEL solidification and the satisfactory immobilization of cells for imaging purposes.

## Fluorescence recovery after photobleaching (FRAP) and confocal fluorescence microscopy

FRAP experiments were conducted at room temperature (20–22 °C) using either an upright Zeiss LSM710 confocal fluorescence microscope or an inverted Zeiss LSM 780 multiphoton confocal fluorescence microscope (in Biosciences Technology Facility, University of York) (see Appendix Tables S13 and S14 for image acquisition settings). Data were collected using the FRAP program within the ZEN2011 software. Pre-bleach fluorescence and differential interference contrast (DIC) images were acquired. One pole was subjected to photobleaching (using a defined, standardized area of 50 pixels × 30 pixels) via application of 20 - 30 laser pulses at maximum laser power. Five post-bleach images were then collected at 1-s intervals. Subsequent post-bleach images were then collected at 1-, 2- and 5-minute time points post-bleaching. Some FRAP data were also collected by photobleaching the middle of cells, and analysis of the resulting image sequences yielded identical results. Post-FRAP DIC images were then acquired to verify that the position of the subject cell had not changed, and the integrity of its cell envelope had not been compromised.

### Quantitative analysis of FRAP data

Data from individual FRAP image sequences were converted into recovery curves by plotting normalized fluorescence intensity in the bleached region over time. Normalization of fluorescence recovery was necessary for comparison of individual FRAP analyses since levels of initial fluorescence and the extent of photobleaching varied from cell to cell. A double-normalization technique was then applied to normalize the fluorescence intensity in the bleached region of each image against both the initial cell fluorescence and the degree of sample photobleaching during image acquisitions (Phair et al, 2003).

*For each time series image*
Bleached region average fluorescence intensity ($B_t$) and whole cell average fluorescence intensity ($T_t$) values are normalized against corresponding background average fluorescence intensity ($BG_t$):

$$\hat{B}_t = B_t - BG_t$$

$$\hat{T}_t = T_t - BG_t$$

At each time point, $t$ the double normalized average fluorescence intensity in the bleached region ($\hat{I}_t$) is obtained using the formula below:

$$\hat{I}_t = \frac{\frac{\hat{B}_t}{\hat{B}_0}}{\frac{\hat{T}_t}{\hat{T}_0}} = \frac{\hat{T}_0 \hat{B}_t}{\hat{T}_t \hat{B}_0}$$

Recovery curves were constructed from FRAP data using a custom MATLAB code (available at https://github.com/RosieLeaman/FRAP). The FRAP image sequence data were processed in the following steps:

1. Identification of the cell being photobleached (if multiple cells are in the image).
2. Identification of the bleached area boundaries, cell outline, and identification of a region of background that does not contain any other cells for the purposes of double normalization.
3. Measurement of the average intensity in each of the three regions at each time point, applying any drift correction in the *xy* plane as required.
4. Construction of the corresponding recovery curve from the extracted average intensities at each time point (see Appendix Fig. S11).

Mobile fraction formula:

$$Mobile\,fraction = \frac{fluorescence\,recovered}{fluorescence\,photobleached} = \frac{I_{end} - I_{postbleach}}{1 - I_{postbleach}}$$

$$Immobile\,fraction = 1 - mobile\,fraction$$

$I$ = average fluorescence intensity in bleached region.
$I_{end}$ = average fluorescence intensity in the bleached region at 300 s.
$I_{postbleach}$ = average fluorescence intensity in bleached immediately after the photobleaching step.

### Statistical analysis of FRAP data

The mobile fraction was defined as the proportion of labeled species (in this context AF- or AZDye-labeled, Kdo-analog containing LPS) that undergo lateral diffusion over the course of the experiment. The immobile fraction was defined as the proportion of molecules that do not undergo diffusion over the course of the experiment. As detailed in Appendix Fig. S11, the mobile fraction of fluorescently labeled LPS molecules can be derived from the FRAP recovery curves. Prior double normalization enabled comparison between different experimental conditions. Individual experiment mobile fractions were calculated in Microsoft Excel using the following equation and data outputs from the FRAP analysis code implemented in MATLAB (Mathworks):

$$Mobile\,fraction = \frac{I_{end} - I_{postbleach}}{1 - I_{postbleach}}$$

For each condition, the labeled LPS mobile fractions were calculated in at least 30 individual cells (n ≥ 30) collected over at least three independent biological replicates ($N \geq 3$). Box plots were produced representing the mobile fraction distributions for given conditions using the seaborn Python library. In each box plot, the center line is the median, while the box defines the upper and lower

quartiles of the data, and the whiskers enclose the interquartile range (i.e., middle 50% of the data). Values outside of the whiskers were determined as outliers (i.e., outside the interquartile range) by the seaborn box plot function. To assess differences in the mobile fraction distributions obtained from two conditions, a two-sided Mann–Whitney $U$ test was applied since it does not assume that the underlying distribution is normal. The test was applied in MATLAB using the ranksum function. Differences in the mobile fraction distributions of two conditions were judged statistically insignificant if the $P$ value of the test >0.05.

## Preparation of labeled cell samples for direct stochastic optical reconstruction microscopy (dSTORM)

Immediately after the "Click-iT" labeling step, the cell suspension was transferred to a pre-chilled 2 mL microcentrifuge tube and pelleted via centrifugation (10,000–12,000×$g$, 2 min, 4 °C). Labeled cell pellets were resuspended to an $OD_{600}$ of 1.0 in 0.2 μm filtered PBS supplemented with 2 mM $MgSO_4$ and 0.5 mM $CaCl_2$, and transferred to a fresh, pre-chilled 2 mL microcentrifuge tube. Cells were re-pelleted via centrifugation. This washing step was done a total of three times. The washed and labeled cell pellet was fixed via resuspension to an $OD_{600}$ of 1.5 in 4% (v/v) ultrapure, methanol-free paraformaldehyde (PFA, Polysciences Ltd.) in PBS. This solution was incubated for 30 min on a rotary wheel (12 rpm, 80° incline relative to the benchtop) at room temperature. The fixed cell suspension was then transferred to a fresh 1.5 mL microcentrifuge tube and pelleted via centrifugation. The pellet of fixed and labeled cells was resuspended to an $OD_{600}$ of 1.0 in PBS with $MgSO_4$ and $CaCl_2$, transferred to a fresh microcentrifuge tube, and pelleted via centrifugation. This washing step was done three times in total.

## Mounting of fixed, washed, and fluorescently labeled cell samples using poly-D-lysine-coated coverslips for imaging by dSTORM

The washed, fixed, and labeled cell pellet in a microcentrifuge tube was placed on ice to cool to 4 °C and resuspended in degassed, pre-chilled Glucose oxidase (GluOx) (Sigma-Aldrich)/Catalase (Cat) (Sigma-Aldrich) $O_2$ scavenging buffer to an $OD_{600}$ of 1.5 via repeated gentle pipetting and vortexing (Appendix Table S15) (Van de Linde et al, 2011; Veigel et al, 1998). In all, 45.5 μL of the suspension was transferred to a 500 μL microcentrifuge tube on ice. 2.5 μL of 1 M β-mercaptoethylamine hydrochloride (β-MEA) (50 mM final concentration), 2 μL of 0.5% (w/v) 5 μm diameter silica bead slurry (0.02% (w/v) final bead concentration), and 0.5 μL TetraSpeck 0.2 μm microsphere standard solution ($1.5 \times 10^9$ particles mL$^{-1}$, Thermo Fisher Scientific) were added. TetraSpeck beads were added to enable accurate AF488 and AZ647 channel alignment during two-color dSTORM image processing. In all, 10 μL of the suspension was then pipetted onto the center of a pre-chilled (to minimize $O_2$ scavenging buffer enzyme activity), clean 1.0–1.2 mm thick glass slide. The slide was covered with a poly-D-lysine-coated, 18 mm² high precision (Zeiss), no. 1.5 glass coverslip ensuring no bubbles were present in the slide chamber. The chamber was sealed with a clear nail varnish. The slide was then incubated at 4 °C for 20 min while protected from light to allow the nail varnish to dry and the cells to adhere to the poly-D-lysine-coated coverslip surface.

## 2D-dSTORM data acquisition with AF488- and AZ647-labeled samples

dSTORM imaging experiments were done on a Zeiss Elyra 7 super-resolution imaging platform (Biosciences Technology Facility, University of York) in laser widefield beam path mode using ZEN Black 3.0 SR FP2 software (Carl Zeiss AG). Objective coupled total internal reflection fluorescence (TIRF) illumination was employed using a Plan-Apochromat ×63/1.46 NA Korr oil immersion objective Var 2 lens together with TIRF uHP (ultra-high power) laser power density setting. Image areas were set to 128 pixels × 128 pixels for two cells or 64 pixels × 64 pixels for a single cell and saved in 16-bit format. Initial excitation and intersystem crossing of all fluorophores in the target cell(s) from their ground singlet state to an excited triplet "dark" state was achieved via application of a short (200–300 frames, 50 ms exposure time), high-intensity laser pulse (488 nm: 20–25%, 642 nm: 10–15%) with the microscope in epifluorescence (EPI) mode. dSTORM time series image sequences were then collected using highly inclined and laminated optical sheet (HILO) illumination for 10,000 frames (50 ms exposure time) using reduced laser powers (488 nm: Starting at 2% increasing to 15% over the time series, 642 nm: Starting at 0.7% and increasing to 5% over the time series). To propagate fluorophore "blinking" in the latter 5000 frames, 405 nm light was introduced during the "transfer" phase of individual image collection to reduce fluorophore photobleaching resulting from simultaneous exposure to both 488 nm and 405 nm light (Van de Linde et al, 2011). HILO illumination mode was used to enable excitation of fluorophores in the outer membrane furthest from the coverslip thereby maximizing the signal to noise (S:N) ratio. Additional dSTORM image acquisition settings are summarized in Appendix Table S16.

Two-color dSTORM imaging of AF488- and AZ647-labeled species in the OM of in-frame, in-focus cells was done sequentially using the Zeiss Elyra 7 super-resolution imaging platform with 488 nm and 642 nm excitation lasers. The BP420-480 + BP490-550 emission filter (Appendix Table S16) was used to prevent cross-channel "bleeding" of fluorescence emission from the two dyes. dSTORM data for AZ647-labeled species was collected first using the second sCMOS camera due to the lower photostability of the dye compared to AF488. dSTORM data for AF488-labeled species was then collected using the first sCMOS camera. After the initial high-intensity burst to trigger fluorophore intersystem crossing from ground singlet states to a "dark" triplet state, the fluorescent emission 'blinking' events were collected over 10,000 frames using a 50 ms exposure for each channel.

## Processing of single color channel 2D-dSTORM images

dSTORM raw data were processed using the SMLM (single-molecule localization microscopy) processing facility found within the ZEN Black 3.0 SR software. Time series image sequences were converted to a crude single dSTORM image, discarding overlapping molecules using a x,y Gauss fit model, with a peak mask size of 9 pixels and a peak intensity to noise ratio of 6.5. Typical filtering settings are detailed in Appendix Table S17. SMLM-grouping settings were adjusted to minimize double-counting of individual dye molecules (5 frames for the maximum "ON" time, 50 frames for maximum "OFF" gap, 1.7 pixel for capture radius). Pixel

resolution was set at 10 nm pixel$^{-1}$ with localized peaks displayed in gauss mode reflecting the degree of localization precision. dSTORM image data was converted into images in .CZI format using the ZEN 3.0 SR software "Convert to Image" tool and exported to FIJI for final image production (Schindelin et al, 2012). Final images were saved in both .TIFF and .PNG formats.

## Co-localization analysis of fluorescently labeled newly inserted LPS and propargyl-OmpA$^{E89}$ in the bacterial OM

Initial two-color dSTORM image processing and subsequent production were done for each color channel in ZEN black Elyra software as detailed above. Channels were aligned using the "Channel Alignment" tool in ZEN black 3.0 SR software, using the positions of TetraSpeck microspheres as markers. TetraSpeck microspheres also served as fiducials for drift correction during SMLM image processing. Subsequent image processing and co-localization analyses were done in FIJI (Schindelin et al, 2012). Each two-color 2D-dSTORM cell image was processed individually using a standard co-localization method. A Gaussian blur was applied to both channels ($\sigma = 1.0$). The channels were separated into individual images and converted to TIFF format (8 bit) and converted to a greyscale color scheme.

Regions for co-localization analysis were selected via application of a minimal threshold based on the image with the lowest apparent signal (usually fluorescently labeled propargyl-OmpA$^{E89}$) to exclude extracellular regions and dark regions, thus preventing background fluorescence signal and its "absence" from artificially inflating correlation values. The Co-loc2 FIJI plug-In was then applied to obtain Pearson correlation coefficient (PCC) values describing the degree of co-localization of the labeled species in the 2 channels (Manders et al, 1992; Pearson, 1896). This method for co-localization analysis was selected because PCC is independent of signal and background fluorescence levels, removing the need for extended image pre-processing and thus minimizing the potential influence from unconscious user bias. Normalized output values ($+1$ to $-1$) enable results for individual cells to be grouped and PCC value distributions from different conditions compared without further processing. The Co-loc2 plug-in superimposed the fluorescent distributions of both channels, represented by scatterplots. The application of linear fits to these scatterplots enabled the calculation of PCC values for each two-color image describing the pixel-by-pixel covariance in both channels.

The PCC formula is given by:

$$PCC = \frac{\sum_i (AZ647_i - \overline{AZ647}) \times (AF488_i - \overline{AF488})}{\sqrt{\sum_i (AZ647_i - \overline{AZ647})^2 - \sum_i (AF488_i - \overline{AF488})^2}}$$

$AZ647_i$ = AZ647 fluorescent intensity in pixel $i$

$AF488_i$ = AZ488 fluorescent intensity in pixel $i$

$\overline{AZ647}$ = AZ647 mean fluorescent intensity

$\overline{AF488}$ = AZ488 mean fluorescent intensity.

Potential PCC values range from $+1$ where the fluorescent intensities of the two channels are perfectly linearly related (i.e., perfect co-localization) to $-1$ where fluorescent intensities are perfectly inversely related (i.e., a complete absence of co-localization). A value of 0 is expected if the distributions of the two differentially labeled fluorescent species were independent of one another.

## Identification and measurement of LPS and OmpA-containing regions in 2D two-color dSTORM images

After processing the two-color 2D-dSTORM data and conversion to .CZI images, the LPS and OmpA channels were split into separate images in ImageJ (version 1.54j) (Schneider et al, 2012). A global threshold was applied to each image using the ISODATA auto threshold algorithm to define discrete patches within each image. The image was then converted to a binary mask using the "Make binary" tool in ImageJ. Lateral and perpendicular longitudinal measurements were recorded for patches with median diameters > 50 μm. Histograms of the patch dimensions were prepared using OriginPro (OriginLab).

## In vivo TIRFM imaging and single-particle tracking of fluorescently labeled LPS or colicin Ia/CirA complex

For localization of LPS, the bacteria were metabolically labeled with Kdo-azide and fluorescently labeled by the SPAAC approach using AZDye 488-DBCO as described above (Appendix Table S8). For localization of CirA receptors, the bacterial cells were resuspended in CDM ($OD_{600} \approx 1$) containing the AF488-labeled colicin Ia probe at a concentration of 300 nM. The suspension was then incubated on a rotary wheel (12 rpm, 80° incline relative to the benchtop) for 30 min at room temperature, protected from light. The labeled cells were washed twice by centrifugation (10,000–12,000×$g$, 2 min) in a 2 mL microcentrifuge tube and resuspended in a reduced volume of fresh supplemented M9 CDM to yield a final theoretical $OD_{600}$ of 3–4. Bacteria were immobilized on poly-D-lysine-coated quartz sides in supplemented M9 CDM for TIRFM using an ultra-thin sample chamber as described previously (Rassam et al, 2015).

The prism-coupled TIRFM (including the 488 nm laser power, optical filters, and two-channel image splitter) used in this work was described previously (Rassam et al, 2015). All digital video was recorded with an Evolve 512 emCCD (Photometrics) at 30 Hz ($512 \times 512$ pixels$^2$) or at 67 Hz ($128 \times 128$ pixels$^2$) and collected at room temperature (20-22 °C). Video data was collected after moderate photobleaching to enable the detection and tracking of single fluorophores. All raw digital video was processed with a moving 3-frame median time filter implemented in MATLAB (MathWorks), then analyzed with PaTrack software to localize (with Gaussian fitting of fluorescence peak to achieve sub-pixel resolution) and track fluorophores (Dosset et al, 2016). This tracking software uses a back-propagation neural network trained on synthetic trajectories to assign free Brownian, confined, and directed diffusion modes in each single-particle trajectory. The algorithm was particularly useful in this work, where individual LPS molecules might display both free Brownian and confined diffusion characteristics in a single trajectory. The single-molecule trajectories analyzed in this work were all at least 0.9 s in duration (typically 0.9–2 s in duration) and displayed either confined or free Brownian diffusion (i.e., mixed trajectories were not observed). The type of diffusion assigned to the trajectories was confirmed by analyzing the categorized trajectories again using TrackMate (Tinevez et al, 2017). The following parameters were routinely used in PaTrack for tracking fluorophores in the video data (at 30 Hz and 67 Hz) for

AZDye 488-labeled rough or deep rough LPS and AF488-labeled colicin Ia/CirA receptor complex: resolution = 0.096 μm/pixel, dimension particle = 0.4 μm, maximum particle displacement = 0.4 μm, death frames = 5, short trajectory filter = 12 frames, and maximum eccentricity = 1.7. The fluorescence intensity of every tracked fluorophore was manually checked to confirm that photobleaching occurred in a single-step drop to the background intensity level, which indicated a single fluorophore was present.

Fluorophores with a multi-step drop in fluorescence intensity during photobleaching and those that did not photobleach during the video recording (thus precluding use of the aforementioned test) were omitted from the MSD analysis. Once tracked, any fluorophores localized to a bacterial cell with an end-of-trajectory, asymptotic MSD value greater than $0.008\ \mu m^2$ were retained ($MSD_{end\ of\ trajectory} > 0.008\ \mu m^2$), while those with a lower end-of-trajectory MSD value ($MSD_{end\ of\ trajectory} < 0.008\ \mu m^2$) were considered immobilized on the quartz surface and discarded. The confinement diameter ($d_{confine}$) was calculated for each track by averaging MSD data for a specific time window (time lag $\approx 0.5-0.8$ s) and inputting this average MSD ($MSD_{average}$) into the following equation (Kusumi et al, 1993): $MSD_{average} = (d_{confine}/2)^2/6$. The two-dimensional diffusion coefficient ($D_{2D}$) was calculated from the time-dependence of the MSD using linear regression of the first 4 time delays for confined particles or the first 13 time delays for freely diffusing particles.

## Data availability

Custom image analysis scripts are available on the GitHub repository: https://github.com/RosieLeaman/FRAP.

The source data of this paper are collected in the following database record: biostudies:S-SCDT-10_1038-S44318-026-00711-5.

## Peer review information

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

## Acknowledgements

We thank the University of York Bioscience Technology Facility (BTF) for access to microscopy facilities, and G Calder (BTF) and N Sergent (Zeiss) for expert assistance with dSTORM data collection and analysis. We thank the following researchers for providing bacterial strains or plasmid DNA constructs: D Browning and I Henderson (OAR *E. coli* DFB1655), M van der Woude (*S.* Typhimurium LT2 and *E. coli* UTI89), G Thomas (*P. aeruginosa* PA01), E Severi (*E. coli* BW25113 with *imp4213* mutation), P de Boer (pNP4 plasmid) and E Lemke (pEVOL-pylRS plasmid). We thank C Sharrock and B Tello Rubio for assistance with site-directed mutagenesis in LptD and OmpA, respectively. We thank P Fogg, C Hill, C Spicer, and G Thomas for comments on the manuscript. CGB thanks C Kleanthous and S Khalid (University of Oxford) for helpful research discussions and sharing unpublished data. This work was supported by The University of York, the BBSRC (19ALERT Mid-Range Equipment Initiative Award to the Department of Biology to purchase ZEISS Elyra 7 system, BB/T017589/1), the BBSRC White Rose Doctoral Training Partnership (PhD studentship award to JN, 2272649), the Wellcome Trust CIDCATS Interdisciplinary PhD Training Programme (PhD studentship award to RML, WT095024MA), a Horizon Europe Guarantee Award to MAF (selected by the ERC and funded by UKRI, EP/X023680/1) and a MRC Discovery Award to CGB and MCC (MC-PC-15073).

## Author contributions

**Joe Nabarro**: Conceptualization; Resources; Data curation; Software; Formal analysis; Validation; Investigation; Visualization; Methodology; Writing—original draft; Writing—review and editing. **Rosalyn M Leaman**: Conceptualization; Resources; Data curation; Software; Formal analysis; Validation; Investigation; Visualization; Methodology; Writing—review and editing. **Samuel Lenton**: Conceptualization; Data curation; Formal analysis; Validation; Investigation; Visualization; Methodology; Writing—review and editing. **Leonore Mantion**: Resources; Investigation; Methodology; Writing— review and editing. **Richard J Spears**: Resources; Writing—review and editing. **Mark C Coles**: Conceptualization; Supervision; Funding acquisition; Writing—review and editing. **Dmitri O Pushkin**: Conceptualization; Supervision; Writing—review and editing. **Martin A Fascione**: Conceptualization; Resources; Supervision; Funding acquisition; Visualization; Writing—original draft; Project administration; Writing—review and editing. **Christoph G Baumann**: Conceptualization; Resources; Data curation; Formal analysis; Supervision; Funding acquisition; Investigation; Visualization; Methodology; Writing—original draft; Project administration; Writing—review and editing.

Source data underlying figure panels in this paper may have individual authorship assigned. Where available, figure panel/source data authorship is listed in the following database record: biostudies:S-SCDT-10_1038-S44318-026-00711-5.

## Disclosure and competing interests statement

The authors declare no competing interests.

# Expanded View Figures

▶

**Figure EV1.   Efficient and specific fluorescent labeling of LPS via a 2-step bio-orthogonal approach enabled LPS lateral mobility to be assessed by in vivo fluorescence microscopy.**

(**A**) Fluorescent labeling of LPS at the cell surface. In situ metabolic labeling of the chemically conserved LPS inner core oligosaccharide was done with an azide-functionalized Kdo-analog. Azide handles within LPS inner core oligosaccharide domain enabled bio-orthogonal conjugation of alkyne-functionalized small, photostable organic fluorescence dyes via Cu(I)-catalyzed (CuAAC) or Cu(I)-free strain promoted azide-alkyne cycloaddition (SPAAC). Gal*f* Galactofuranose, Glc Glucose, Rha Rhamnose, GlcNAc *N*-Acetyl-glucosamine, Hep Heptose, Kdo 3-Deoxy-D-manno-oct-2-ulosonic acid, Gal Galactose, GlcN Glucosamine, IM Inner membrane, Lpp Braun's lipoprotein, OMP Outer membrane protein, Und-PP undecaprenyl pyrophosphate anchor. (**B**) Characterization of LPS lateral mobility by fluorescence microscopy. LPS lateral mobility was assessed in a range of Gram-negative bacterial strains under different conditions using (i) fluorescence recovery after photobleaching (FRAP) and (ii) single-particle tracking (SPT) after detection by total internal reflection fluorescence microscopy. The photobleached region of the cell in the schematic for the FRAP experiment has a dashed outline and violet shading. Both confined and free Brownian lateral diffusion could be detected in the SPT experiments, and the type of lateral diffusion observed was found to vary depending on the strain and treatment type.

**A**

**Step 1:** LPS inner core metabolic labeling with Kdo-N₃

**Step 2:** Kdo-N₃-containing LPS fluorescently labeling via CuAAC or SPAAC

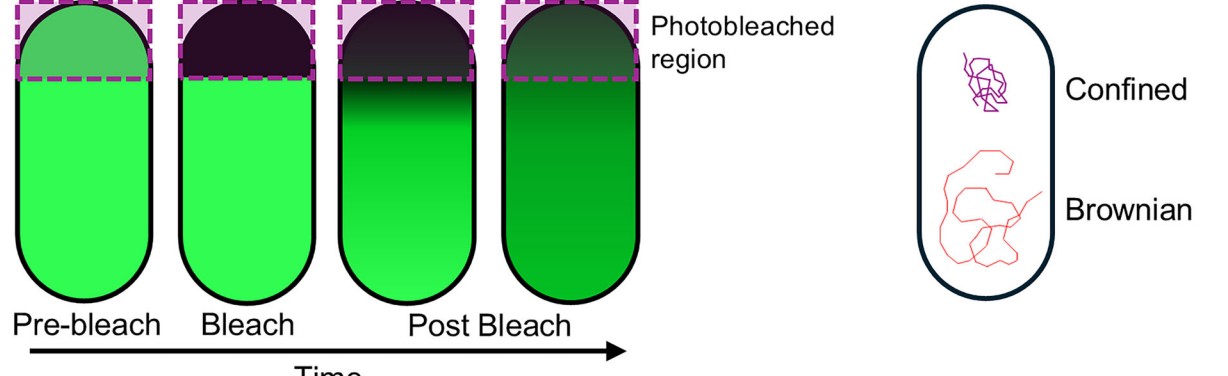

**B**

**i) FRAP:** Average LPS diffusion on cellular scale

Photobleached region

Pre-bleach  Bleach  Post Bleach

Time

**ii) SPT:** LPS diffusion on molecular scale

Confined

Brownian

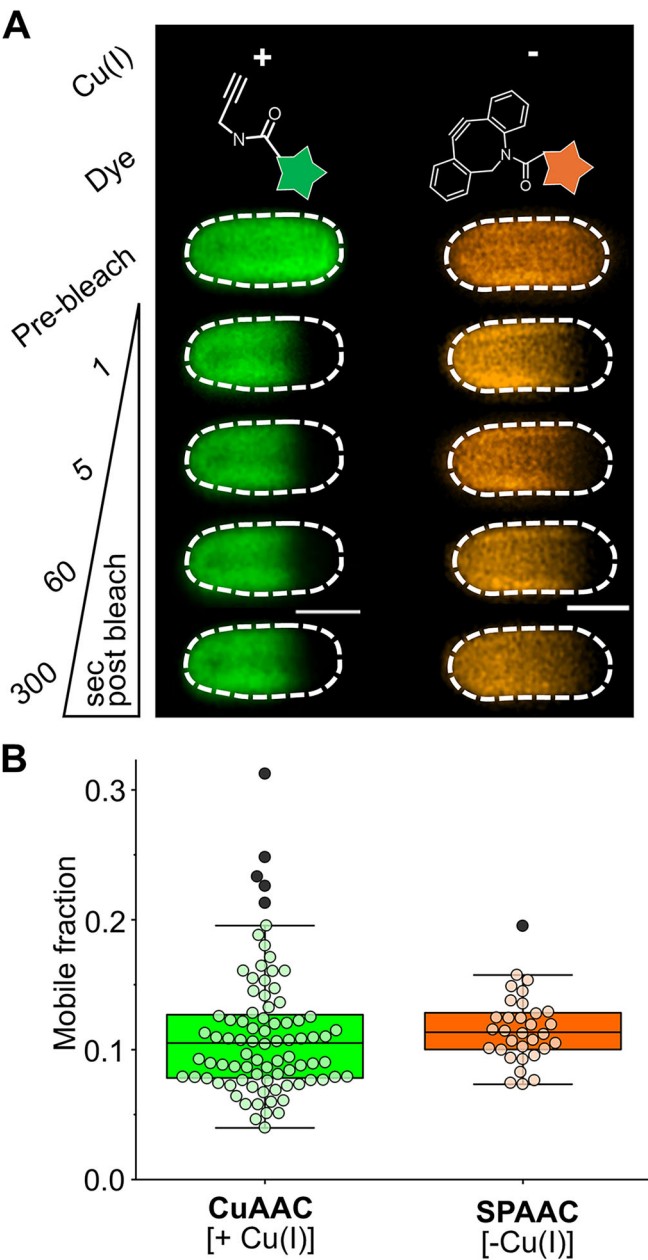

**Figure EV2.   Exposure to Cu(I) during fluorescent labeling of LPS does not significantly affect LPS lateral mobility as measured by FRAP.**

(A) Representative FRAP time-lapse images monitoring fluorescence recovery in photobleached regions of the OM for Δ*waaC* cells with LPS labeled using AF488-alkyne via CuAAC (left) or AZDye 568-DBCO via SPAAC (right). Dashed border denotes the outline of each bacterial cell. Scale bars: 1.0 μm. (B) No statistically significant difference (by Mann–Whitney test, $P = 0.11$) was observed in the LPS mobile fractions in the OM of *E. coli* Δ*waaC* cells irrespective of whether LPS was labeled via Cu(I)-catalyzed azide-alkyne cycloaddition (CuAAC, green box) or Cu(I)-free strain promoted azide-alkyne cycloaddition (SPAAC, orange box). CuAAC: median mobile fraction $= 0.11$ ($n = 84$). SPAAC: median mobile fraction $= 0.11$ ($n = 32$). All FRAP experiments were done in triplicate using independent biological samples. For each box plot, the center line is the median while the box defines the upper and lower quartiles of the data and the whiskers enclose the interquartile range (i.e., middle 50% of the data). Each symbol represents an individual measurement with black-filled symbols classified as outliers (i.e., outside the interquartile range). Source data are available online for this figure.

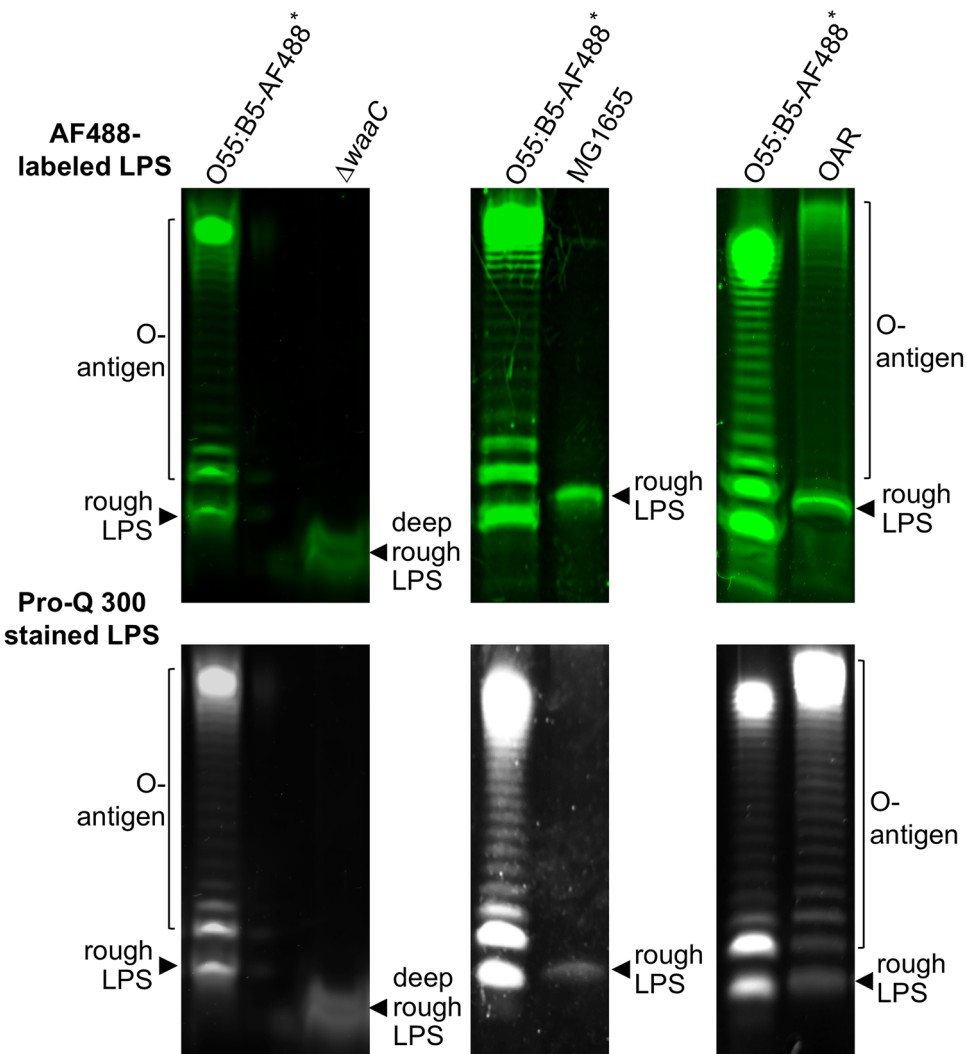

**Figure EV3. TSDS-PAGE analysis of LPS extracted from bacterial cells after two-step metabolic / bio-orthogonal labeling with Kdo-N₃ and AF488-alkyne.**

LPS was extracted from *E. coli* Δ*waaC* (left-hand gel images), *E. coli* MG1655 (middle gel images) and *E. coli* DFB1655 O-antigen restored (OAR) cells (right-hand gel images). The *E. coli* mutant strains (Δ*lpp*, Δ*ompA*) are in a BW25113 background. LPS extracted from the wild-type BW25113 strain yields the same TSDS-PAGE result as MG1655. AF488-labeled LPS was visualized with 488 nm laser excitation (top row of gel images) and total LPS was visualized using Pro-Q Emerald 300 LPS staining with UV illumination (bottom row of gel images). *Purchased AF488-labeled O55:B5 LPS standard (0.1 μg per lane). Source data are available online for this figure.

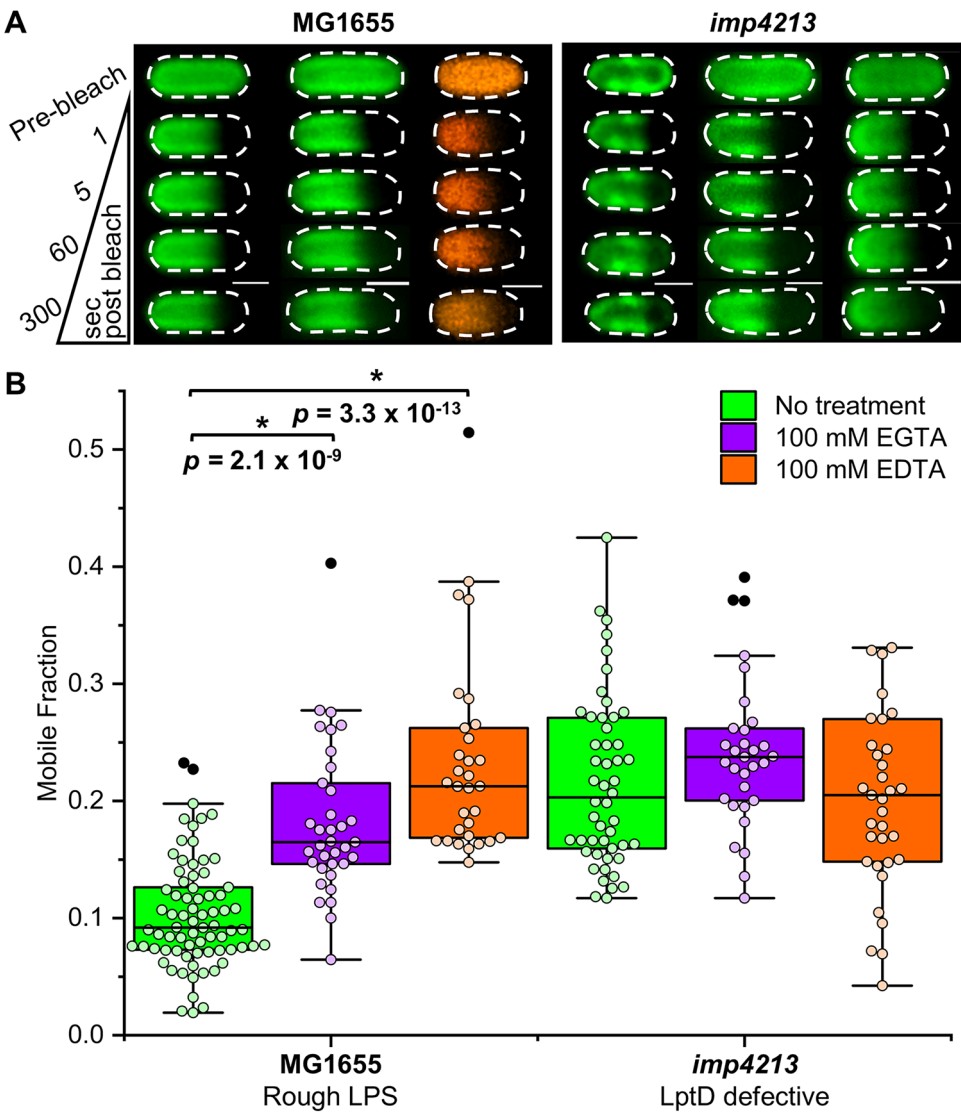

**Figure EV4. Observed differences in the sensitivity of LPS mobility to chelator treatments for *E. coli* MG1655 and *imp4213* cells suggests divalent cation-mediated LPS–LPS interactions contribute to LPS restriction in the OM.**

(A) Representative FRAP sequences for AF488-LPS (green) or AF568-LPS (orange) in the OM of *E. coli* MG1655 and *imp4213* cells without (left-hand vertical image sequence) or with 100 mM EGTA (middle vertical image sequence) and 100 mM EDTA (right-hand vertical image sequence) treatments. Dashed border denotes the outline of each bacterial cell. Scale bars: 1.0 μm. (B) Effect of 100 mM EGTA and 100 mM EDTA treatments on LPS mobile fraction distributions in the OM of *E. coli* MG1655 and *imp4213* cells measured via FRAP. The FRAP sequences and mobile fraction data for the *E. coli* MG1655 strain were reused from Fig. 4A,B. All FRAP experiments were done in triplicate using independent biological samples. For each box plot, the center line is the median while the box defines the upper and lower quartiles of the data and the whiskers enclose the interquartile range (i.e., middle 50% of the data). Each symbol represents an individual measurement with black-filled symbols classified as outliers (i.e., outside the interquartile range). MG1655: no treatment, median = 0.092 (*n* = 74); 100 mM EGTA, median = 0.165 (*n* = 35); 100 mM EDTA, median = 0.213 (*n* = 31). *imp4213*: no treatment, median = 0.207 (*n* = 49); 100 mM EGTA, median = 0.237 (*n* = 31); 100 mM EDTA, median = 0.205 (*n* = 35). The disruption of divalent-cation-mediated LPS-LPS interactions by chelator treatments is responsible for the observed significant increase (by Mann–Whitney test) in LPS mobile fractions for the MG1655 cells. In contrast, the disruption of OM asymmetry in the *imp4213* cells caused by phospholipid flipping into the outer leaflet of the OM appears to disrupt these divalent cation-mediated interactions yielding a much lower overall sensitivity to chelator treatment for this strain. Source data are available online for this figure.

