## [Peer Review File · The EMBO Journal]

Lipopolysaccharide confinement in the bacterial outer membrane is governed by interactions within the conserved Lipid A anchor

Joe Nabarro, Rosalyn Leaman, Samuel Lenton, Leonore Mantion, Richard Spears, Mark Coles, Dmitri Pushkin, Martin Fascione, and Christoph Baumann

Corresponding authors: Christoph Baumann (christoph.baumann@york.ac.uk) , Martin Fascione (martin.fascione@york.ac.uk)

Review Timeline:

Submission Date:	6th Jun 25
Editorial Decision:	4th Jul 25
Revision Received:	17th Dec 25
Editorial Decision:	16th Jan 26
Revision Received:	20th Jan 26
Accepted:	27th Jan 26

Editor: Ieva Gailite

Transaction Report:

Dear Christoph,

Thank you for submitting your manuscript for consideration by the EMBO Journal. We have now received comments from a full set of reviewers, which are included below for your information.

As you will see, all reviewers are generally positive in their evaluation and appreciate the contribution of the study to the research field. However, they also indicate a number of concerns that would need to be addressed before they can support publication here, including regarding more balanced presentation and interpretation of the data and previous literature. From my side, I find these requests generally reasonable. Therefore, based on these positive assessments, I invite you to submit a revised manuscript in response to the comments by all reviewers. I think that it could be useful to discuss the revision in more detail via email or phone/videoconferencing - please let me know which option you prefer.

We generally allow three months as standard revision time, which can be extended to six months in the case of major revisions. Should you foresee a problem in meeting this deadline, please let us know in advance to discuss an extension.

As a matter of policy, competing manuscripts published during this period will not negatively impact on our assessment of the conceptual advance presented by your study. However, please contact me as soon as possible upon publication of any related work to discuss the appropriate course of action.

When preparing your letter of response to the referees' comments, please bear in mind that this will form part of the Review Process File and will therefore be available online to the community. For more details on our Transparent Editorial Process, please visit our website: <https://www.embopress.org/page/journal/14602075/authorguide#transparentprocess>. Please also see the attached instructions for further guidelines on preparation of the revised manuscript.

Please feel free to contact me if you have any further questions regarding the revision. Thank you for the opportunity to consider your work for publication. I look forward to discussing your revision.

With best wishes,

Ieva

Ieva Gailite, PhD
Senior Scientific Editor
The EMBO Journal
Meyerohofstrasse 1
D-69117 Heidelberg
Tel: +4962218891309
i.gailite@embojournal.org

- a point-by-point response to the referees' comments, with a detailed description of the changes made (as a word file).
- a word file of the manuscript text.
- individual production quality figure files (one file per figure)
- a complete author checklist, which you can download from our author guidelines

(<https://www.embopress.org/page/journal/14602075/authorguide>).

- Expanded View files (replacing Supplementary Information)

- a Reagents and Tools Table as part of the Methods section, which can be downloaded from our author guidelines

(<https://www.embopress.org/page/journal/14602075/authorguide#structuredmethods>)

We realize that it is difficult to revise to a specific deadline. In the interest of protecting the conceptual advance provided by the work, we recommend a revision within 3 months (2nd Oct 2025). Please discuss the revision progress ahead of this time with the editor if you require more time to complete the revisions.

Referee #1:

This manuscript focuses on the lack of mobility of LPS in the outer membrane (OM) of *E. coli*. Its key findings are that:

- A. (Fig. 1) FRAP-measured LPS mobility in the OM for deep rough, rough and smooth LPS is not significantly different from the mobility of (static) cell wall components; that is, regardless of O-antigen, the LPS are essentially static in the OM to within the detection limit of the experiment.
- B. (Fig. 2) OmpA and LPS tend to cluster and this clustering anticorrelates, suggesting that there are OMP enriched and LPS-enriched domains in the OM.
- C. (Fig. 3) LPS is slightly more mobile when coupling to the cell wall is reduced and when the amount of LPS in the OM is reduced.
- D. (Fig. 4,5) In absence of divalent cations, known for enhancing LPS intermolecular interactions and binding, LPS is again slightly more mobile, where magnesium appears to be a (slightly) stronger binder than calcium.
- E. (Fig. 6) The immobility of LPS in the OM, and its dependence on divalent cation, is observed in different strains and species.

Of note, in all cases the immobile fraction remains $> \sim 70\%$, reiterating the main point that LPS has at best very low mobility in the OM

While findings A and B are not entirely novel (see comments below), and depending on one's perspective, findings C-E may not be hugely surprising (except possibly the stronger effect of magnesium over calcium), the experiments are technically well carried out and quantify these effects in a way that has not been done before. I therefore consider them a valuable addition to the literature.

Some more specific comments on the manuscript are as follows.

p. 4 "The lack of consensus in these landmark studies..."

This suggests that there remains a major controversy that is addressed in this study. But e.g. Ref. 20, Kumar et al., 2025, explicitly settles this point in favor of low LPS mobility, already demonstrating clearly that LPS is laterally confined in the OM. The manuscript here confirms those recent findings. The introduction and discussion should more explicitly acknowledge that the lateral confinement of LPS in the OM has already been demonstrated with state-of-the-art labeling methods elsewhere, and that the (only, but per se interesting) novel aspect of this manuscript is the further characterization of the molecular basis of this lateral confinement. And of note, I consider that also such a replication (of only very recent findings) has its place in the scientific literature, certainly when done at the high technical level as in this study.

p. 5, "we unequivocally show that LPS is laterally confined in the outer leaflet of the OM of the *E. coli* K12 cell envelope".

p. 15, start of the Discussion, "[this study] demonstrates unequivocally that this essential glycolipid is indeed tightly restricted in the native OM of *E. coli* (Ref. 20)"

Along the lines of the comment above, it would seem more appropriate to replace this with "we confirm that LPS is laterally confined in the OM, in agreement with previous work (Ref. 20)" or similar.

p. 5, "Despite pronounced differences in the predominating LPS oligosaccharide domain structures (Figure 1D), inter strain

variations in LPS mobile fraction distributions were statistically insignificant".

p. 15, start of the Discussion, "the observed confinement is not dependent on oligosaccharide headgroup size or structure"
This seems an overstatement of the result. The data show no significant difference here between different LPS oligosaccharide domain structure, but neither with presumed immobile PG. Hence a more careful interpretation would be that, if there are any changes in LPS mobility due to differences in oligosaccharide domain structures, these differences are undetectable here as in all cases, the mobility appeared at the bottom of the detection limit in these experiments.

p. 6, "LPS prefers self assembly into supramolecular structures or patches largely devoid of OmpA (Figure 2C), as recently hypothesised (ref. 30)".

(i) The authors write that LPS prefers self-assembly into supramolecular clusters largely devoid of OmpA, but do not appear to consider other, even more abundant OMPs, notably the trimeric OmpF and OmpC. This limitation should be clearly stated in the Results section and considered in the Discussion section.

(ii) Given the experimental evidence in Ref. 30, Benn et al., 2021, it would do better justice to past literature if the authors replace the "as recently hypothesized" with "as previously shown".

In the caption of Fig. 2, the authors interpret the "discrete clustering" of OmpA and LPS labels as indicative of LPS and OMPs assembling into their own spatially separated supramolecular islands.

This appears an over-interpretation, as it would need much stronger, quantitative evidence to demonstrate that OMPs and LPS are mutually exclusive in their respective clusters/islands. Of note, "islands" smaller than 50 nm are excluded from the analysis in B.

Here I have presumed that the 50 micrometer boundary in the caption of Fig. 2B is a typo that should read 50 nanometer!

In fact, a much more plausible interpretation is that there is local enrichment in LPS or OmpA, but no full segregation.

Also Fig. 2, as a control, could the authors show dSTORM data (with similar overall counts) on a generic membrane marker, to demonstrate that the behavior of the labeled LPS and labeled OmpA is clearly distinct from what is expected for homogenous (not-clustered) OM components?

Referee #2:

Summary

Nabarro et al. used state-of-the-art methods to label lipopolysaccharide (LPS) within the outer membrane and in combination with FRAP determined LPS diffusion rates. They not only show that LPS is highly confined (as has recently been shown by Kumar et al., 2025) but dissect the key interactions which lead to this lack of mobility - mainly the Lipid A anchor as opposed to the core oligosaccharide or O-antigen. This is demonstrated by E. coli containing deep rough, rough, or full length LPS having the same level of lateral mobility - which is again highly confined despite differences in LPS length. Utilizing dSTORM they demonstrated LPS patches can be observed on the outer membrane which are devoid of outer membrane protein, as recently described by Benn et al., 2021 utilizing atomic force microscopy. Then using FRAP they dissected the interactions within the outer membrane which govern lateral mobility, such as outer membrane asymmetry, outer membrane-to-cell wall linkages, and divalent cation mediated interactions. Furthermore, they demonstrate that it is the LPS-LPS acyl chain interactions which is the main interaction governing lateral mobility of LPS. Finally, by examining other Gram-negative bacteria they demonstrate LPS confinement is a conserved phenomenon in multiple species.

The paper is a very valuable work because it shows an important principle - that the outer membrane is a lipid A-magnesium gel - that most microbiologists take for granted by has not been directly visualized at the level of lipopolysaccharide diffusion. Great work overall.

We have several minor comments:

The manuscript was difficult to parse at times. For readability we would suggest breaking your paragraphs into smaller paragraphs.

Abstract.

- We do not think it will be surprising at all to microbiologists that magnesium has a stronger effect than calcium on outer membrane diffusion. It is common knowledge that bacteria use magnesium rather than calcium both in the cytoplasm and in the outer membrane. It is not surprising that LPS structure is evolutionarily evolved to bind magnesium.

Results

LPS is tightly confined in the OM irrespective of oligosaccharide domain size.

- We recently demonstrated that core oligosaccharide length has a strong effect on the load-bearing capacity of the outer membrane (Fitzmaurice et al., Biophys J, 2025). It is probably worth noting in the discussion that the emerging picture from your

work and ours is that lipid A composition inhibits LPS diffusion, which probably allows core oligosaccharides to interact and bear in-plane mechanical stresses.

LPS assembles into supramolecular structures in the OM.

- Benn et al., 2021 is referenced but it would be nice to know how your measurements of domain size compare to that of the AFM measurements in that paper. Furthermore, it was pretty clearly demonstrated in that work that there are patches without beta-barrels, not just "hypothesized." So, this language can change as well.

LPS lateral confinement is influenced by OM asymmetry and OM-PG covalent linkages.

- In regard to the Δ lpp strain showing increased lateral diffusion of LPS, can you speculate a molecular mechanism for this in the discussion. For instance, Lpp, is an example of a static phospholipid which should theoretically exert drag on neighboring mobile phospholipids in the IM.

Hydrophobic forces within the OM contribute to restricted LPS lateral mobility.

- It is stated that intermolecular H-bonds in the OM are "unlikely to be influenced by detergent treatment" but this is an incorrect statement, as detergents are well known to affect H-bonds. Rather in regard to the O-antigen, it presumably has many more H-bonds due to its increased length, as H-bonds tend to scale with the size of the core-oligosaccharide length (Fitzmaurice et al., 2025). There are likely so many H-bonds in the O-antigen, they can compensate for the detergent treatment.

- In our opinion, the results from the ClearColi strain are a key result demonstrating acyl chain number is the strongest molecular interaction governing LPS lateral diffusion, and this deserves to be highlighted in a main figure.

Dylan Fitzmaurice
Enrique Rojas

Referee #3:

Lipopolysaccharide confinement is governed by key interactions within the conserved Lipid A anchor

The article by Nabarro, Leaman et al., presents a conceptually insightful study of how lipopolysaccharide mobility is restricted in the bacterial outer membrane. Using combination of metabolic LPS labeling, FRAP, and single particle tracking, the authors investigate the roles of lipid A acylation, phosphate charge and their coordination by divalent cations, OM protein tethers, and membrane asymmetry in modulating lateral LPS dynamics. Their results reveal that LPS confinement is a robust feature of the OM, and that this organization can be disrupted under diverse structural perturbations.

The authors show that the lateral mobility of LPS, measured by FRAP, is equivalent to that of the peptidoglycan mesh, which is immobile. This non-mobility is maintained regardless of the nature of the LPS oligosaccharide domain. Upon perturbation of OM asymmetry—either by impairing Lpt function or removing the major lipoprotein Lpp or adding detergents—LPS mobility increases. In the case of Lpt malfunction, a subpopulation of LPS becomes hypermobile, as revealed by SPT. Interestingly, the mobility of outer membrane proteins is unaffected by these changes, suggesting a differential response of LPS-rich versus OMP-rich regions. The authors also show that Mg^{2+} exerts a stronger constraining effect on LPS mobility than Ca^{2+} and that LPS confinement is conserved in other Gram-negative bacteria, including pathogenic ones.

This is a carefully performed and technically well-designed study. The experimental design is comprehensive: the authors probe a wide range of genetic backgrounds (Δ waaC, Δ lpp, imp4213, CleaColi), OM states (deep rough, rough and smooth LPS), and chemical perturbations (EDTA, EGTA, urea, detergents).

The study addresses an important question about how LPS mobility is controlled in the outer membrane and what physical interactions underlie its confinement. The findings are conceptually relevant and contribute to our understanding of OM organization and robustness.

Comments:

1. The authors argue that their metabolically engineered LPS probe offers a more direct approach to labeling LPS, potentially avoiding limitations associated with earlier methods using physically large fluorescent probes. However, the labeling method they use may itself also alter the biophysical properties of LPS and thereby influence its mobility. Recently, WGA was also shown to label LPS rather than the previously assumed PG (Dubois et al., 2025). To support the robustness of their findings, the authors could consider testing whether LPS remains similarly restrained when labeled with WGA in FRAP experiments.

2. At page 5 and Supplementary Figure 2: "...the loss of cell viability associated with copper exposure during the CuAAC reaction does not disrupt LPS confinement." It is unclear what specific loss of viability the authors are referring to. This should be quantified or clarified. Moreover, the authors should justify their use of both CuAAC and SPAAC cycloaddition methods. If CuAAC reduces viability, why is it still used in some experiments (e.g., Fig. 1E,F)? Conversely, why was SPAAC chosen for

others (e.g., Fig. 3A-C)? A rationale for selecting one method over the other in each case would help readers understand how methodological differences might impact LPS behavior.

Additionally, SPAAC is sometimes referred to as "Cu(I)-free" and elsewhere as "copper-free 'click'-labelling"; standardizing the terminology throughout the manuscript would improve clarity. The same recommendation applies to CuAAC.

3. Page 6: The justification that the ompA* mutant complements ompA deletion based on resistance to ampicillin and chloramphenicol is not convincing, especially since the strain carries resistance genes for both antibiotics. While it is important to demonstrate that ompA* functionally complements a Δ ompA background, the rationale provided here is invalid. The authors should instead assess a phenotype that directly depends on ompA function. An established Δ ompA phenotype; such as OM permeability defects, vancomycin sensitivity, or altered envelope stability would provide a more appropriate and interpretable readout of complementation.

4. In the paragraph beginning by "This implies that the separate LPS-rich and OMP-rich regions of the OM may respond differently...", the authors state that "the majority of LPS remained immobile in these mutants (Figure 3D)," referring to both the Δ lpp and imp4213 strains. However, unless I missed it, SPT was not performed on the Δ lpp strain. Without SPT data, it is unclear whether the same subpopulation behavior (i.e., ~1/3 highly mobile and ~2/3 less mobile LPS molecules) observed in imp4213 also applies to Δ lpp. Have the authors performed this analysis, and if so, could they show whether the LPS mobility distribution in Δ lpp follows the same pattern?

A similar point applies to the statement on page 12 concerning low Mg²⁺ conditions. Unless SPT was also conducted under these conditions, the claim that most LPS remains immobile should be either experimentally supported or interpreted with more caution.

5. Page 7: The authors conclude that one-third of the LPS molecules diffuse freely and two-thirds show restricted diffusion in the imp4213 strain. However, even the "restricted" fraction in imp4213 displays a diffusion coefficient that is still approximately three times higher than that of LPS in the wild-type strain. This should be acknowledged, as it indicates that LPS mobility is globally increased in imp4213, not only due to the emergence of a freely diffusing population.

In the same context, the statement that EGTA or EDTA treatment "does not affect LPS mobility in the imp4213 strain" (Supplementary Figure 8) might be misleading. Since imp4213 already exhibits elevated LPS mobility, it would be more accurate to state that EDTA/EGTA treatment does not further exacerbate this increased mobility, rather than implying no effect in absolute terms.

6. Figure 4A: The authors do not explain why they use 10 mM of chelator for the Δ waaC strain but 100 mM for the rough or OAR strains. This discrepancy should be justified, especially since chelator concentration could influence the degree of LPS destabilization.

In addition, while EGTA treatment of Δ waaC significantly increases the mobile fraction compared to untreated cells, this increase is markedly smaller than that observed with EDTA treatment. Given that Δ waaC also lacks the two phosphate groups normally present on the heptose residues, could the authors comment on whether these phosphates contribute to LPS immobilization in wild-type cells, and if they do it via Mg²⁺ than Ca²⁺ cations?

7. In Figure 5B, the condition with 2 mM Ca²⁺ and 0.1 mM Mg²⁺ results in a higher LPS mobile fraction than the condition with 0.1 mM Ca²⁺ and 0.1 mM Mg²⁺. Could the authors explain this outcome?

8. Did the author checked whether the treatment with 0.025% (w/v) LDAO does not generate OMVs formation prior to FRAP analyses that might induce an increase of LPS mobility?

9. Page 8: The use of the term "demonstrating" in the sentence: "Rescued imp4213 cells producing functional LptD also showed reduced sensitivity to SDS, EDTA and bile salts (Supplementary Figure 6) demonstrating the links between OM asymmetry, LPS confinement and the barrier function of the OM" feels too strong, as the data presented are primarily correlative.

10. Although a minor point, it is unclear why the authors used the CirA receptor to assess OMP-confined diffusion in Supplementary Figure 9, given that they specifically generated the ompA* construct for this study.

11. "However" in MSD sentence: The use of "however" to introduce the higher confinement diameter in imp4213 is slightly awkward, as it doesn't truly contrast with the previous sentence.

Title:

Lipopolysaccharide confinement is governed by key interactions within the conserved Lipid A anchor

Authors:

Joe Nabarro, Rosalyn M. Leaman, Samuel Lenton, Leonore Manton, Richard J. Spears, Mark C. Coles, Dmitri O. Pushkin, Martin A. Fascione and Christoph G. Baumann

Point-by-Point Response to Comments from Referee #1:

This manuscript focuses on the lack of mobility of LPS in the outer membrane (OM) of *E. coli*. Its key findings are that:

- A. (Fig. 1) FRAP-measured LPS mobility in the OM for deep rough, rough and smooth LPS is not significantly different from the mobility of (static) cell wall components; that is, regardless of O-antigen, the LPS are essentially static in the OM to within the detection limit of the experiment.
- B. (Fig. 2) OmpA and LPS tend to cluster and this clustering anticorrelates, suggesting that there are OMP enriched and LPS-enriched domains in the OM.
- C. (Fig. 3) LPS is slightly more mobile when coupling to the cell wall is reduced and when the amount of LPS in the OM is reduced.
- D. (Fig. 4,5) In absence of divalent cations, known for enhancing LPS intermolecular interactions and binding, LPS is again slightly more mobile, where magnesium appears to be a (slightly) stronger binder than calcium.
- E. (Fig. 6) The immobility of LPS in the OM, and its dependence on divalent cation, is observed in different strains and species.

Of note, in all cases the immobile fraction remains $> \sim 70\%$, reiterating the main point that LPS has at best very low mobility in the OM

While findings A and B are not entirely novel (see comments below), and depending on one's perspective, findings C-E may not be hugely surprising (except possibly the stronger effect of magnesium over calcium), the experiments are technically well carried out and quantify these effects in a way that has not been done before. I therefore consider them a valuable addition to the literature.

Response 1: Thank you for your appreciation of our work.

Some more specific comments on the manuscript are as follows.

p. 4 "The lack of consensus in these landmark studies..."

This suggests that there remains a major controversy that is addressed in this study. But e.g. Ref. 20, Kumar et al., 2025, explicitly settles this point in favor of low LPS mobility, already demonstrating clearly that LPS is laterally confined in the OM. The manuscript here confirms those recent findings. The introduction and discussion should more explicitly acknowledge that the lateral confinement of LPS in the OM has already been demonstrated with state-of-the-art labeling methods elsewhere, and that the (only, but per se interesting) novel aspect of this manuscript is the further characterization of the molecular basis of this lateral confinement. And of note, I consider that also such a replication (of only very recent findings) has its place in the scientific literature, certainly when done at the high technical level as in this study.

p. 5, "we unequivocally show that LPS is laterally confined in the outer leaflet of the OM of the E. coli K12 cell envelope".

p. 15, start of the Discussion, "[this study] demonstrates unequivocally that this essential glycolipid is indeed tightly restricted in the native OM of E. coli (Ref. 20)"

Along the lines of the comment above, it would seem more appropriate to replace this with "we confirm that LPS is laterally confined in the OM, in agreement with previous work (Ref. 20)" or similar.

Response 2: We have rewritten the sections on pages 4, 5 and 15 to acknowledge that rough LPS confinement in the native *E. coli* K12 cell envelope has been shown previously by Kumar *et al.* (2025). In the Introduction (page 3), we have retained a brief summary of the prior work done to assess LPS mobility in the OM of intact cells because two of these landmark studies also presented data consistent with smooth LPS having low mobility.

p. 5, "Despite pronounced differences in the predominating LPS oligosaccharide domain structures (Figure 1D), inter strain variations in LPS mobile fraction distributions were statistically insignificant".

p. 15, start of the Discussion, "the observed confinement is not dependent on oligosaccharide headgroup size or structure"

This seems an overstatement of the result. The data show no significant difference here between different LPS oligosaccharide domain structure, but neither with presumed immobile PG. Hence a more careful interpretation would be that, if there are any changes in LPS mobility due to differences in oligosaccharide domain structures, these differences are undetectable here as in all cases, the mobility appeared at the bottom of the detection limit in these experiments.

Response 3: As suggested by the referee, we have rewritten this section of the text (pages 5-6) to acknowledge that any small differences in the lateral mobility of deep rough, rough and smooth LPS could not be distinguished from the tightly confined cell wall in our FRAP experiments. We did not alter the wording at the start of the Discussion (page 15) because it succinctly describes one of the main findings of this work, *i.e.* LPS confinement is not dependent on oligosaccharide headgroup size or structure.

p. 6, "LPS prefers self assembly into supramolecular structures or patches largely devoid of OmpA (Figure 2C), as recently hypothesised (ref. 30)".

(i) The authors write that LPS prefers self-assembly into supramolecular clusters largely devoid of OmpA, but do not appear to consider other, even more abundant OMPs, notably the trimeric OmpF and OmpC. This limitation should be clearly stated in the Results section and considered in the Discussion section.

Response 4: Monomeric OmpA is as abundant (1×10^5 copies per cell) as the porins OmpF or OmpC (2.5×10^5 copies per cell) if you consider the oligomeric state of these trimeric beta-barrel outer membrane proteins. Furthermore, OmpA has been used previously as a marker for OMP-rich regions in the outer membrane (Benn *et al.*, 2021), and it functions in mediating OM-PG interactions in these regions via its periplasmic PG-binding domain (Benn *et al.*, 2024). We have addressed this referee's comment by explicitly stating in the text on page 6 that OmpA is used here as a marker for these porin-rich regions.

(ii) Given the experimental evidence in Ref. 30, Benn et al., 2021, it would do better justice to past literature if the authors replace the "as recently hypothesized" with "as previously shown".

Response 5: We have edited the text (page 6) to indicate that this local enrichment of LPS has been observed previously as recommended by this referee.

In the caption of Fig. 2, the authors interpret the "discrete clustering" of OmpA and LPS labels as indicative of LPS and OMPs assembling into their own spatially separated supramolecular islands.

This appears an over-interpretation, as it would need much stronger, quantitative evidence to demonstrate that OMPs and LPS are mutually exclusive in their respective clusters/islands. Of note, "islands" smaller than 50 nm are excluded from the analysis in B.

Response 6: Thank you for this recommendation. We have edited the text in the legend to Figure 2 to address the referee's comments. Our use of the phrase "spatially separated" was not meant to imply OMPs and LPS are mutually exclusive in their respective islands. We now refer to the obvious discrete clustering of LPS and OmpA observed in the composite dSTORM images as a result which "indicates that local enrichment of LPS molecules and OMPs can occur in the OM, resulting in their own respective supramolecular islands".

Here I have presumed that the 50 micrometer boundary in the caption of Fig. 2B is a typo that should read 50 nanometer!

Response 7: This typo has been corrected in the text. Thank you!

In fact, a much more plausible interpretation is that there is local enrichment in LPS or OmpA, but no full segregation.

Response 8: Thank you for this recommendation. We have edited the wording in the legend to Figure 2 as recommended by this referee.

Also Fig. 2, as a control, could the authors show dSTORM data (with similar overall counts) on a generic membrane marker, to demonstrate that the behavior of the labeled LPS and labeled OmpA is clearly distinct from what is expected for homogenous (not-clustered) OM components?

Response 9: We agree that this would be an ideal control; however, a homogeneously distributed OM component that can be labeled with an exogenously added fluorophore in intact cells does not exist. A lipophilic membrane dye could be used to stain the outer membrane, but this would not work well for dSTORM due to dye quenching during cell fixation and dye mobility even post-fixation. We have also noticed that fluorescent dyes utilized for staining eukaryotic plasma membranes do not exclusively stain the outer membrane in the Gram-negative bacterial cell envelope. Unfortunately, we cannot satisfactorily address this referee's comment with the suggested experimental control. Reassuringly, the clustering of LPS and OmpA observed by two-color dSTORM is in excellent agreement with previous results obtained using high-resolution atomic force microscopy (Benn *et al.*, 2021).

Point-by-Point Response to Comments from Referee #2:

Summary

Nabarro et al. used state-of-the-art methods to label lipopolysaccharide (LPS) within the outer membrane and in combination with FRAP determined LPS diffusion rates. They not only show that LPS is highly confined (as has recently been shown by Kumar et al., 2025) but dissect the key interactions which lead to this lack of mobility - mainly the Lipid A anchor as opposed to the core oligosaccharide or O-antigen. This is demonstrated by E. coli containing deep rough, rough, or full length LPS having the same level of lateral mobility - which is again highly confined despite differences in LPS length. Utilizing dSTORM they demonstrated LPS patches can be observed on the outer membrane which are devoid of outer membrane protein, as recently described by Benn et al., 2021 utilizing atomic force microscopy. Then using FRAP they dissected the interactions within the outer membrane which govern lateral mobility, such as outer membrane asymmetry, outer membrane-to-cell wall linkages, and divalent cation mediated interactions. Furthermore, they demonstrate that it is the LPS-LPS acyl chain interactions which is the main interaction governing lateral mobility of LPS. Finally, by examining other Gram-negative bacteria they demonstrate LPS confinement is a conserved phenomenon in multiple species.

The paper is a very valuable work because it shows an important principle - that the outer membrane is a lipid A-magnesium gel - that most microbiologists take for granted but has not been directly visualized at the level of lipopolysaccharide diffusion. Great work overall.

Response 10: Thank you for your supportive comments and your appreciation of our work.

We have several minor comments:

The manuscript was difficult to parse at times. For readability we would suggest breaking your paragraphs into smaller paragraphs.

Response 11: We have addressed this comment throughout the Results and Discussion sections wherever it was possible to do so without disrupting the flow of the narrative.

Abstract.

- We do not think it will be surprising at all to microbiologists that magnesium has a stronger effect than calcium on outer membrane diffusion. It is common knowledge that bacteria use magnesium rather than calcium both in the cytoplasm and in the outer membrane. It is not surprising that LPS structure is evolutionarily evolved to bind magnesium.

Response 12: We have removed the word 'surprisingly' from the abstract text, thus allowing the reader to decide whether or not our result is surprising to them.

Results

LPS is tightly confined in the OM irrespective of oligosaccharide domain size.

- We recently demonstrated that core oligosaccharide length has a strong effect on the load-bearing capacity of the outer membrane (Fitzmaurice et al., Biophys J, 2025). It is probably worth noting in the discussion that the emerging picture from your work and ours is that lipid A composition inhibits LPS diffusion, which probably allows core oligosaccharides to interact and bear in-plane mechanical stresses.

Response 13: Thank you for the recommendation. We have now integrated this article into our discussion of how core oligosaccharide length or size influences the properties of the OM (see pages 8 and 13).

LPS assembles into supramolecular structures in the OM.

- Benn et al., 2021 is referenced but it would be nice to know how your measurements of domain size compare to that of the AFM measurements in that paper. Furthermore, it was pretty clearly demonstrated in that work that there are patches without beta-barrels, not just "hypothesized." So, this language can change as well.

Response 14: Thank you for this recommendation. We have edited the text as recommended by this referee (and referee #1) to indicate that this local enrichment of LPS has been observed previously. Benn *et al.* (2021) observed "sparse, pore-free, smooth patches" in the OM by AFM which they attributed to LPS-rich regions. The size of these patches ranged from 25-225 nm in width (mean diameter \approx 55 nm). The LPS-rich patches we observed by dSTORM ranged from 50-600 nm in width (Figure 2, mean diameter \approx 260 nm). We did not include patches with a diameter < 50 nm. These independent measurements of LPS-rich patch size in the OM are reassuringly similar considering they were done using very different imaging techniques. It is unsurprising that our measurements of patch size are larger given an auto threshold algorithm was utilized in ImageJ to define discrete patches in the dSTORM images. This unbiased thresholding method may slightly overestimate patch size by joining neighbouring patches.

LPS lateral confinement is influenced by OM asymmetry and OM-PG covalent linkages.

- In regard to the Δ lpp strain showing increased lateral diffusion of LPS, can you speculate a molecular mechanism for this in the discussion. For instance, Lpp, is an example of a static phospholipid which should theoretically exert drag on neighboring mobile phospholipids in the IM.

Response 15: Thank you for this thoughtful suggestion, and for encouraging us to speculate on the origin of the increased lateral LPS diffusion observed in the Δ lpp strain. We agree with your suggested mechanism and thought that Braun's lipoprotein might be acting in this way. We have now speculated on the origin of the increased LPS lateral diffusion in the Δ lpp strain citing the potential drag of Braun's lipoprotein on neighbouring mobile membrane lipids in the native OM (page 16). The Δ lpp strain also has a hypervesiculation phenotype. Therefore, we have also speculated on how vesicle formation might also contribute to the observed changes in LPS mobility (page 8).

Hydrophobic forces within the OM contribute to restricted LPS lateral mobility.

- It is stated that intermolecular H-bonds in the OM are "unlikely to be influenced by detergent treatment" but this is an incorrect statement, as detergents are well known to affect H-bonds.

Rather in regard to the O-antigen, it presumably has many more H-bonds due to its increased length, as H-bonds tend to scale with the size of the core-oligosaccharide length (Fitzmaurice et al., 2025). There are likely so many H-bonds in the O-antigen, they can compensate for the detergent treatment.

Response 16: Thank you for these comments. We have edited the text as recommended by this referee and cited the prior work investigating how core oligosaccharide length in LPS affects OM mechanics (page 13).

- In our opinion, the results from the ClearColi strain are a key result demonstrating acyl chain number is the strongest molecular interaction governing LPS lateral diffusion, and this deserves to be highlighted in a main figure.

Response 17: Thank you for these comments about the importance of the results from the ClearColi strain. We have now incorporated these supplementary data into the new Figure 6 in the main text.

Dylan Fitzmaurice / Enrique Rojas

Point-by-Point Response to Comments from Referee #3:

Lipopolysaccharide confinement is governed by key interactions within the conserved Lipid A anchor

The article by Nabarro, Leaman et al., presents a conceptually insightful study of how lipopolysaccharide mobility is restricted in the bacterial outer membrane. Using combination of metabolic LPS labeling, FRAP, and single particle tracking, the authors investigate the roles of lipid A acylation, phosphate charge and their coordination by divalent cations, OM protein tethers, and membrane asymmetry in modulating lateral LPS dynamics. Their results reveal that LPS confinement is a robust feature of the OM, and that this organization can be disrupted under diverse structural perturbations.

The authors show that the lateral mobility of LPS, measured by FRAP, is equivalent to that of the peptidoglycan mesh, which is immobile. This non-mobility is maintained regardless of the nature of the LPS oligosaccharide domain. Upon perturbation of OM asymmetry-either by impairing Lpt function or removing the major lipoprotein Lpp or adding detergents-LPS mobility increases. In the case of Lpt malfunction, a subpopulation of LPS becomes hypermobile, as revealed by SPT. Interestingly, the mobility of outer membrane proteins is unaffected by these changes, suggesting a differential response of LPS-rich versus OMP-rich regions. The authors also show that Mg^{2+} exerts a stronger constraining effect on LPS mobility than Ca^{2+} and that LPS confinement is conserved in other Gram-negative bacteria, including pathogenic ones. This is a carefully performed and technically well-designed study. The experimental design is comprehensive: the authors probe a wide range of genetic backgrounds ($\Delta waaC$, Δlpp , $imp4213$, CleaColi), OM states (deep rough, rough and smooth LPS), and chemical perturbations (EDTA, EGTA, urea, detergents).

The study addresses an important question about how LPS mobility is controlled in the outer membrane and what physical interactions underlie its confinement. The findings are conceptually relevant and contribute to our understanding of OM organization and robustness.

Response 18: Thank you for your supportive comments and your appreciation of our work.

Comments:

1. The authors argue that their metabolically engineered LPS probe offers a more direct approach to labeling LPS, potentially avoiding limitations associated with earlier methods using physically large fluorescent probes. However, the labeling method they use may itself also alter the biophysical properties of LPS and thereby influence its mobility. Recently, WGA was also shown to label LPS rather than the previously assumed PG (Dubois et al., 2025). To support the robustness of their findings, the authors could consider testing whether LPS remains similarly restrained when labeled with WGA in FRAP experiments.

Response 19: We have already cited Ghosh *et al.* (2005) where fluorescently labeled ConA (a lectin) was used to label LPS for FRAP microscopy in a manner which is like that used for wheat germ agglutinin (WGA, a lectin) by Dubois *et al.* (2025). Therefore, it has been established in prior work that these types of multivalent lectin-based LPS labels can be used to probe LPS mobility. We did not feel it was necessary to repeat this work again using a different lectin-based probe. We note here, as stated in the introductory section of the manuscript, that the multivalent interactions mediated by lectin-based probes could themselves influence the mobility of LPS. It was for this reason that we used a bio-orthogonal fluorescent labeling approach for LPS.

2. At page 5 and Supplementary Figure 2: "...the loss of cell viability associated with copper exposure during the CuAAC reaction does not disrupt LPS confinement." It is unclear what specific loss of viability the authors are referring to. This should be quantified or clarified. Moreover, the authors should justify their use of both CuAAC and SPAAC cycloaddition methods. If CuAAC reduces viability, why is it still used in some experiments (e.g., Fig. 1E,F)? Conversely, why was SPAAC chosen for others (e.g., Fig. 3A-C)? A rationale for selecting one method over the other in each case would help readers understand how methodological differences might impact LPS behavior.

Additionally, SPAAC is sometimes referred to as "Cu(I)-free" and elsewhere as "copper-free 'click'-labelling"; standardizing the terminology throughout the manuscript would improve clarity. The same recommendation applies to CuAAC.

Response 20: Thank you for these comments on the CuAAC and SPAAC reactions. It is well-known and well-documented that the Cu(I) exposure associated with the CuAAC approach renders the bacterial cells non-viable after 'click'-labeling. However, the higher level of fluorescent labeling achieved with the CuAAC approach is very useful for FRAP, super-resolution imaging and TSDS-PAGE studies. We have added a statement on pages 4-5 which clearly defines when the CuAAC approach is used. An existing statement on page 7 clearly explains why the SPAAC approach was used for the single-particle tracking experiments. We also now consistently refer to the two types of 'click'-labeling used in this work as 'the CuAAC approach' (copper-catalyzed) and 'the SPAAC approach' (copper-free) to avoid any confusion.

3. Page 6: The justification that the *ompA** mutant complements *ompA* deletion based on resistance to ampicillin and chloramphenicol is not convincing, especially since the strain carries resistance genes for both antibiotics. While it is important to demonstrate that *ompA** functionally complements a $\Delta ompA$ background, the rationale provided here is invalid. The authors should instead assess a phenotype that directly depends on *ompA* function. An established $\Delta ompA$ phenotype; such as OM permeability defects, vancomycin sensitivity, or altered envelope stability would provide a more appropriate and interpretable readout of complementation.

Response 21: Thank you for the critical feedback on our evidence for complementation in the $\Delta ompA$ strain by recombinant OmpA labeled with a non-canonical amino acid (OmpA*). We agree that our original evidence for complementation was sub-optimal though we don't accept that our rationale for using resistance to ampicillin and chloramphenicol was invalid. Regardless, we tested different selective conditions and found that both the $\Delta ompA$ strain and the $\Delta ompA$ transformed with the pBAD-*ompA** and pEVOL-pyIRS plasmids do not grow on Lysogeny Broth (Miller formulation) agar plates supplemented with 0.5% (w/v) SDS and 1 mM EDTA. However, the $\Delta ompA$ strain transformed with the pBAD-*ompA** and pEVOL-pyIRS plasmids did grow on these selective agar plates if they were supplemented with 1 mM propargyl-L-lysine, the non-canonical amino acid substituted for E89 in OmpA*. Therefore, we have confirmed that production of recombinant OmpA* due to 'leaky' expression of the *ompA** gene can functionally complement the $\Delta ompA$ background (described on page 6). The Methods section has been updated to include a description of how this complementation work was done (new sub-section entitled: *Rescuing wild-type phenotype in the E. coli BW25113 $\Delta ompA$ strain by production of functional ncAA-containing OmpA**) and images of representative plates are provided in Appendix Figure S3.

4. In the paragraph beginning by "This implies that the separate LPS-rich and OMP-rich regions

of the OM may respond differently...", the authors state that "the majority of LPS remained immobile in these mutants (Figure 3D)," referring to both the Δ lpp and imp4213 strains. However, unless I missed it, SPT was not performed on the Δ lpp strain. Without SPT data, it is unclear whether the same subpopulation behavior (i.e., ~1/3 highly mobile and ~2/3 less mobile LPS molecules) observed in imp4213 also applies to Δ lpp. Have the authors performed this analysis, and if so, could they show whether the LPS mobility distribution in Δ lpp follows the same pattern?

A similar point applies to the statement on page 12 concerning low Mg^{2+} conditions. Unless SPT was also conducted under these conditions, the claim that most LPS remains immobile should be either experimentally supported or interpreted with more caution.

Response 22: Thank you for this critical feedback. We have revised the text to ensure that it is clear to the reader when our claims are based on either a combination of FRAP and SPT data or only FRAP data. If we state in the text that "the majority of LPS molecules remain immobile" then we clearly state if this claim is only supported by FRAP experimental data. We have now stated in the legend to Figure 3 that the high background fluorescence routinely observed with the Δ lpp strain prevented its study by SPT.

5. Page 7: The authors conclude that one-third of the LPS molecules diffuse freely and two-thirds show restricted diffusion in the imp4213 strain. However, even the "restricted" fraction in imp4213 displays a diffusion coefficient that is still approximately three times higher than that of LPS in the wild-type strain. This should be acknowledged, as it indicates that LPS mobility is globally increased in imp4213, not only due to the emergence of a freely diffusing population.

Response 23: Thank you for this critical feedback. We did not describe this observation in the text as we were relying on the reader to notice this change in D_{2D} for the restricted LPS fraction as reported in the text. On page 7, we have acknowledged that LPS mobility is globally increased in the imp4213 strain due to a reduction in LPS confinement and an increase in the lateral diffusion rate.

In the same context, the statement that EGTA or EDTA treatment "does not affect LPS mobility in the imp4213 strain" (Supplementary Figure 8) might be misleading. Since imp4213 already exhibits elevated LPS mobility, it would be more accurate to state that EDTA/EGTA treatment does not further exacerbate this increased mobility, rather than implying no effect in absolute terms.

Response 24: Thank you for this critical feedback. On page 10, we have adopted the wording suggested by the referee when describing how the chelator treatment does not further change LPS mobility in imp4213 cells.

6. Figure 4A: The authors do not explain why they use 10 mM of chelator for the Δ waaC strain but 100 mM for the rough or OAR strains. This discrepancy should be justified, especially since chelator concentration could influence the degree of LPS destabilization.

Response 25: Thank you for this comment because it is important for the reader to understand how the final chelator concentrations were selected. An explanation of how the final chelator concentration was selected for each strain is provided on page 9. We have also added details of the final chelator concentrations used for each strain at this location in the text.

In addition, while EGTA treatment of $\Delta waaC$ significantly increases the mobile fraction compared to untreated cells, this increase is markedly smaller than that observed with EDTA treatment. Given that $\Delta waaC$ also lacks the two phosphate groups normally present on the heptose residues, could the authors comment on whether these phosphates contribute to LPS immobilization in wild-type cells, and if they do it via Mg^{2+} than Ca^{2+} cations?

Response 26: Thank you for the comments on the observed effects of these chelator treatments. The increase in LPS mobility observed by FRAP for the $\Delta waaC$ (deep rough LPS) strain after treatment with EGTA (small change in LPS mobility) and EDTA (large change in LPS mobility) is good evidence that Mg^{2+} ions are primarily involved in bridging anionic headgroups in neighbouring LPS molecules. The increase in LPS mobility observed for the MG1655 (rough LPS) strain after treatment with these chelators is larger for EGTA when compared with the results for the $\Delta waaC$ (deep rough LPS) strain after EGTA treatment. We agree with the referee that divalent cation binding to the phosphate groups in the inner core oligosaccharide domain would occur and that removal of these cations by chelator treatment would contribute to an increase in LPS mobility. We also noticed that EGTA treatment had a more pronounced effect on LPS mobility for the MG1655 (rough LPS) strain though we chose not to speculate on the origin of this effect. We want to remind the referee that higher chelator concentrations were required for the rough LPS strain (100 mM) compared to the deep rough LPS strain (10 mM) to observe a change in LPS mobility by FRAP. In the revised manuscript (page 10), we have noted that the larger change in LPS mobility observed after EGTA treatment suggests Ca^{2+} may have a more significant role in bridging phosphate groups in the inner core oligosaccharide domain of neighbouring rough LPS molecules.

7. In Figure 5B, the condition with 2 mM Ca^{2+} and 0.1 mM Mg^{2+} results in a higher LPS mobile fraction than the condition with 0.1 mM Ca^{2+} and 0.1 mM Mg^{2+} . Could the authors explain this outcome?

Response 27: Thank you for these comments. The most straightforward explanation here is that Mg^{2+} is bound more tightly than Ca^{2+} . A higher concentration of Ca^{2+} can displace Mg^{2+} from the binding site bridging anionic headgroups in neighbouring LPS molecules. When these two divalent metal ions are present at identical concentrations of 0.1 mM, Mg^{2+} binding is favoured over Ca^{2+} binding at these bridging sites and LPS mobility is more confined. We have added a brief statement which explains this outcome (page 12). We have already speculated in the manuscript that the smaller radius of the Mg^{2+} ion allows tighter packing of adjacent LPS molecules.

8. Did the author checked whether the treatment with 0.025% (w/v) LDAO does not generate OMVs formation prior to FRAP analyses that might induce an increase of LPS mobility?

Response 28: Thank you for this comment on the LDAO treatment. We did not test whether the treatment with 0.025% (w/v) LDAO causes an increase in OMV formation. However, we did demonstrate that this LDAO concentration does not alter the growth characteristics or morphology of *E. coli* BW25113 cells (Supplementary Figure 11, now Appendix Figure S8). OMV formation removes a significant patch of the OM (*i.e.* both the inner and outer leaflets). Therefore, we would expect a higher LDAO concentration to be required to generate OMVs which would disrupt the OM and alter the cell morphology. Furthermore, the Δlpp strain has a known hypervesiculation phenotype but it has only a slightly increased LPS mobility (Figure 3)

when compared to the MG1655 strain after treatment with 0.025% (w/v) LDAO (Figure 4). We have now noted in the text (page 8) that the Δlpp strain is hypervesiculating and commented on how this process might lead to OM perturbations.

9. Page 8: The use of the term "demonstrating" in the sentence: "Rescued imp4213 cells producing functional LptD also showed reduced sensitivity to SDS, EDTA and bile salts (Supplementary Figure 6) demonstrating the links between OM asymmetry, LPS confinement and the barrier function of the OM" feels too strong, as the data presented are primarily correlative.

Response 29: Thank you for this comment. We have changed '*demonstrating the links between*' to '*implying links exist between*' as suggested by the referee (page 8).

10. Although a minor point, it is unclear why the authors used the CirA receptor to assess OMP-confined diffusion in Supplementary Figure 9, given that they specifically generated the ompA* construct for this study.

Response 30: Thank you for this comment on OMP selection. We utilized the CirA receptor to assess OMP-confined diffusion for two reasons: 1) we could verify our current SPT results by comparing to what we had published previously using this approach [Rassam *et al.* (2015) *Nature* 523: 333-336], and 2) the signal:noise in the SPT experiments was better with this CirA-specific labeling strategy which was particularly important for the chelator treatments where a higher background fluorescence was observed (Supplementary Figure 9, now Appendix Figure S6). Furthermore, OmpA can be anchored to the peptidoglycan through its periplasmic domain, therefore it is not the best protein target for assessing whether OMP diffusion is confined or not.

11. "However" in MSD sentence: The use of "however" to introduce the higher confinement diameter in imp4213 is slightly awkward, as it doesn't truly contrast with the previous sentence.

Response 31: The word 'however' has been removed from the text in this section.

Dear Christoph,

Thank you for submitting the revised version of your manuscript to The EMBO Journal. The study has now been seen by one of the original referees, who now finds that their main concerns have been addressed satisfactorily and recommends acceptance of the manuscript.

There now remain only a few editorial and formatting points that need to be addressed before I can extend official acceptance of the manuscript:

1. As part of the EMBO Press transparent editorial process, The EMBO Journal will publish online a Peer Review File to accompany accepted manuscripts. This file will be published in conjunction with your paper and will include the anonymous referee reports, your point-by-point response and all pertinent correspondence relating to the manuscript, including decision letters. Please note that the Author Checklist will be published at the end of the Peer Review File. Please let us know if you want to remove or not any figures or data from the Peer Review File prior to publication. Please note that retaining unpublished data in the Peer Review File means that these count as published and that the Peer Review File would need to be referenced in future publications.
2. There is an author name discrepancy between the manuscript and our online system: Mark Coles vs Mark C. Coles. Please check.
3. Please include corresponding authors' email addresses on the title page of the manuscript.
4. Figure 5A, Figure 7A-C and Appendix Figure S11 are not mentioned in the manuscript text. Please add the corresponding callouts.
5. Currently, Table EV1 is provided both in the manuscript text file and uploaded separately. Please remove the copy included in the manuscript file.
6. During our standard image and source data check, we noticed possible figure panel reuse between the following figures:
 - Figure 1E and Figure 7B
 - Figure 4A and Figure 5A
 - Figure 4A and Figure 7B
 - Figure 4A and Figure EV4A
 - Figure 5A and Figure EV4AFurthermore, the numerical source data appear identical for figures 1F, 4B, 5B and 7C. Please check and correct if needed. If this was intentional, please clearly indicate the reuse in the figure legends.
7. During our standard numerical data check, we noticed a number of numerical repetitions in the source data for FRAP experiments. In all tables, two of the columns are identical - I appreciate that this could be the case due to the calculations used. However, there are cases where the overlap is not complete (folder "Source data numerical values shifted"). Furthermore, in several tables some rows are duplicated (folder "Source data duplicated rows"). I have attached the corresponding files with the detected duplications labelled in colour. Please take a look and correct if needed. Please provide a brief explanation, which would be very helpful.
8. Our data editors have flagged the following issues in figure legends that need correcting:
 - Please define the box plots in terms of minima, maxima, centre, bounds of box and whiskers, and percentile in the legends of figures 1C, F; 2C, 3B, D; 4B, 5B, 6C, 7C, EV4 B.
 - Please provide the information on the number and nature of replicates in the legends of figures 1C, F; 3B.
 - Please define the error bars in the legends of figures 3C, 4D.
 - Please define the dashed borders in the legends of figures 1B, E; 2A, 3A, 4A, 5A, 6B, 7B.
9. I would like to propose some edits in the manuscript title, abstract and synopsis - please see below and in the attached file. I have also written a short blurb that will accompany the title of your manuscript in our online system. Please take a look and let me know if any corrections are needed.

With best wishes,

Ieva

Ieva Gailite, PhD
Senior Scientific Editor
The EMBO Journal
Meyerhofstrasse 1

D-69117 Heidelberg
Tel: +4962218891309
i.gailite@embojournal.org

Please remember: Digital image enhancement is acceptable practice, as long as it accurately represents the original data and conforms to community standards. If a figure has been subjected to significant electronic manipulation, this must be noted in the figure legend or in the 'Methods' section. The editors reserve the right to request original versions of figures and the original images that were used to assemble the figure.

We realize that it is difficult to revise to a specific deadline. In the interest of protecting the conceptual advance provided by the work, we recommend a revision within 3 months (16th Apr 2026). Please discuss the revision progress ahead of this time with the editor if you require more time to complete the revisions.

Referee #3:

I have read the authors' response and the updated version of the paper, and I find that my main concerns have been addressed satisfactorily. I have no further comments and support publication of the manuscript in its present form.

The authors addressed the remaining editorial issues.

Dear Christoph,

Thank you very much for addressing the final editorial requests. I am now pleased to inform you that your manuscript has been accepted for publication in the EMBO Journal. Congratulations with a nice study!

You may qualify for financial assistance for your publication charges - either via a Springer Nature fully open access agreement or an EMBO initiative. Check your eligibility: <https://link.springer.com/journal/44318/how-to-publish-with-us>

If you have any questions, please do not hesitate to contact the Editorial Office or me directly. Thank you for your contribution to The EMBO Journal!

With best wishes,

Ieva

Please note that it is The EMBO Journal policy for the transcript of the editorial process (containing referee reports and your response letters) to be published as an online supplement to each paper. If you should prefer removal of any referee-only figures included in the point-by-point response(s), e.g. because they may still be used for future publication or because they have been reproduced from published work by others, please do let us know immediately via response email.
More information is available here: <https://link.springer.com/partners/embo-press/editorial-policies#Peer%20review>
